# Broad-spectrum capture of hundreds of per- and polyfluoroalkyl substances from fluorochemical wastewater

Yiyang Liang [1,2,5], Lihui Yang [1,2,5], Caiming Tang [1,2,5], Ying Yang[1,2], Shangtao Liang [3], Anqi Wang [1,2], Jiale Xu [4], Qingguo Huang [3] & Hui Lin [1,2] ✉

Hundreds of per- and polyfluoroalkyl substances (PFAS) are present in fluorochemical production effluents, and existing adsorption devices are inadequate to address this PFAS challenge given their extreme structural diversity. Here, we achieve the broad-spectrum capture of 107 PFAS from fluorochemical effluents using a treatment-train strategy that combines Zn-based electrocoagulation (EC) with anion-exchange resin (AER) beds. The "zero-carbon" adsorbent, zinc hydroxide flocs generated insitu by Zn-based EC, bulk removes PFAS with log $K_{ow} > 4$ through a semi-micellar adsorption mechanism similar to mineral flotation and achieves adsorption capacities at the optimal level of all reported adsorbents. Technical-economic analysis and life-cycle environmental impact show that coupling Zn-based EC reduces the cost by an order-of-magnitude and the carbon-footprint by 70% compared to AER beds alone. It is also observed that iodinated PFAS, with some fluorine atoms are replaced by iodine atoms, exhibit significantly improved adsorption selectivity, which may shed light on designing environmentally-friendly fluorochemicals.

A study indicates that almost all newborn possesses detectable levels of per- and polyfluoroalkyl substances (PFAS) in their bloodstream[1,2]. Regrettably, PFAS are linked to severe health issues, including cancer and reproductive toxicity. In response to this public health crisis, the United States Environmental Protection Agency (U.S EPA) issued the legally binding U.S. National Drinking Water Standard for PFAS to ensure that everyone has access to clean and safe drinking water, which will ultimately reduce PFAS exposure for more than 100 million Americans[3,4]. Numerous studies have demonstrated that these anthropogenic "forever chemicals" are widely distributed in the waters[5], soils[6,7], atmosphere[8], and living organisms in various venues[9,10], including mountains[11] and deep oceans[12], and from the Antarctic[13] to the Arctic[14]. Industrial wastewater, particularly from fluorochemical-related industries that extensively use PFAS as emulsifiers, is regarded as one primary source of PFAS entering the environment[5,15]. For instance, Feng et al.[16] estimated that a mega fluorochemical industrial park (FIP) located in Shandong, China, had emitted a maximum of 9450 kg of perfluorooctanoic acid (PFOA) and 6066 kg of hexafluoropropylene oxides (HFPO) into the air and water in 2021. These fluorochemical effluents, rich in PFAS, cause serious threats to contaminate water sources in China's urban areas. A study covering an area of 400 million people suggests that fluorochemical production significantly contributes to excessive PFAS concentrations in China's urban drinking water[17].

Eliminating PFAS from waste streams poses a significant challenge due to their highly stable spiral structure and strong carbon-fluorine

[1]Research Center for Eco-Environmental Engineering, Dongguan University of Technology, Dongguan, PR China. [2]College of Eco-Environment and Architectural Engineering, Dongguan University of Technology, Dongguan, PR China. [3]College of Agricultural and Environmental Sciences, Department of Crop and Soil Sciences, University of Georgia, Griffin, GA, US. [4]Department of Civil, Construction and Environmental Engineering, North Dakota State University, Fargo, North Dakota, US. [5]These authors contributed equally: Yiyang Liang, Lihui Yang, Caiming Tang. ✉e-mail: linhui@dgut.edu.cn

bonds (531.5 kJ mol⁻¹). Advanced redox technologies often suffer from harsh conditions, high energy consumption, and difficulty in fully mineralizing PFAS[18–20]. Microbial treatment, such as an anaerobic biochemical process, has shown potential for addressing certain polyfluorinated chemicals. For instance, a recent study demonstrated the capability of this process is capable of defluorinating chlorinated polyfluorocarboxylic acids (Cl-PFCA)[21]. Currently, adsorption is the most practiced technology for the treatment of PFAS-containing waters[22]. Conventional activated carbon (AC) and anion-exchange resins (AER) remain the economically viable adsorbents for PFAS effluent treatment[22], despite the screening of a large variety of adsorbent types and the development of many novel efficient adsorbents such as $\beta$-cyclodextrin polymer[23], metal-organic frameworks[24], and covalent-organic frameworks[25]. However, the effectiveness of AC and AER in treating PFAS in complex effluents is limited[26,27]. Competing constituents, such as dissolved organic matter (DOMs) and various anions, can significantly reduce their adsorption selectivity to PFAS[28]. As a result, adsorption studies and engineering treatments on PFAS are mainly limited to relatively clean water bodies with minimal background matrix, such as drinking water[29,30] and groundwater[31,32]. On the other hand, real-world industrial wastewater typically contains a multitude of PFAS. For instance, Tang et al. identified 175 formulae of PFAS with over 350 congeners in fluorochemical effluents[33]. In a recent study, Zhang and colleagues[32] examined the mass flows of 48 PFAS, ranging from 14.7 to 5200 µg L⁻¹, in the effluents of 10 FIPs in China and showed that the current wastewater treatment processes are ineffective at removing these PFAS from discharges by fluorochemical manufacturers, resulting in significant PFAS contamination of the receiving waters. Specifically, the mass flows of PFAS increased from − 20% to 233% with the activated sludge system but decreased by only 0–13% after the AC filtration. Removing a wide range of PFAS with diverse structures is a challenging task that requires the synergy of multiple interactions. To our knowledge, no study has yet addressed the broad-spectrum adsorption removal of these compounds from complex real wastewater, and most previous reports have focused on removing a few regulated and well-known PFAS, such as PFOA and perfluorooctane sulfonate (PFOS)[34].

One of the defining characteristics of PFAS is its exceptional surface activity, which allows it to be adsorbed in multiple layers through semi-micellar or micellar adsorption. This significantly boosts its adsorption capacity. An example of this is mineral flotation, where trapping agents are adsorbed onto hydrophobic mineral particles through semi-micellar adsorption, enabling selective mineral capture. Inspired by this, we propose a reverse mineral flotation process using hydrophobic "mineral particles" as adsorbents to selectively extract PFAS from water via semi-micellar adsorption. We found that zinc hydroxide flocs, formed in situ by electrolysis, exhibit properties of hydrophobic "mineral particles," thus enabling them to rapidly adsorb hydrophobic long-chain PFAS from water with high adsorption capacities[35,36]. For example, these zinc hydroxide flocs achieved an equilibrium adsorbed amount ($q_e$) of up to 5.74/7.69 mmol g⁻¹ (Zn) for PFOA/PFOS within minutes at an initial concentration of 0.5 mM[34]. We therefore hypothesize that Zn-based electrocoagulation (EC) can effectively capture long-chain PFAS in the fluorochemical wastewater, while the existing AC and/or AER adsorption devices can then remove the remaining low concentrations of short-chain PFAS. This approach is founded on two primary observations: (1) Despite the presence of hundreds of PFAS types in fluorochemical wastewater, long-chain PFAS, particularly PFOA, dominate the total PFAS concentration, and (2) Zn-based EC also removes DOMs and traps colloidal particles from wastewater, significantly mitigating the impact of these competing constituents on subsequent adsorption processes. This strategy represents an attempt to achieve broad-spectrum removal of hundreds of PFAS from real fluorochemical wastewaters, potentially offering options for tackling PFAS in complex industrial waste streams.

Here, we report the extensive capture of 107 PFAS (Fig. 1, ranging from C2 to C16, including 82 carboxylic acids and 25 sulfonic acids), comprising 60 long-chain PFAS (C ≥ 7) ad 47 short-chain PFAS, from polymer fluoropolymer production effluents using a treatment-train strategy that combines Zn-based EC with existing AC and/or AER devices. The effectiveness, cost, and environmental impacts of the treatment-train process were compared to those of a single adsorption process in a systematic evaluation. A mechanism similar to mineral flotation is proposed to explain the selective adsorption of hydrophobic PFAS by the Zn-based EC. Furthermore, the analysis examined the impact of structural features, such as the ratio of F/C and F/H, as well as the number of −O− and C−X (H, Cl, and I), on the adsorption selectivity of PFAS in the Zn-based EC process. It was observed that substituting iodine for fluorine significantly alters the properties of PFAS, that favors their adsorptive removal. This study provides options for addressing the challenge of severe PFAS pollution in fluorochemical production effluents: treatment trains for broad-spectrum, effective PFAS capture as well as treatment-facile fluorochemical design.

## Results and discussion

### Capturing 107 PFAS in complex fluorochemical wastewaters

The study assessed a total of 107 PFAS (Supplementary Table 1), which were classified according to their structural properties into 5 categories (15 classes) (Fig. 1a), including: I) 9 perfluorocarboxylic acids (PFCA, C2 - C10), II) 32 hydrogenated polyfluoroalkyl acids (H-PFAA, C2 - C16), III) 52 poly- and perfluoropolyether acids (Ether-PFAA, C3 - C16), IV) 9 chlorinated polyfluoroalkyl acids (Cl-PFAA, C2 - C9), and V) 5 iodinated polyfluoroalkyl acids (I-PFAA, C2 - C8). Figure 1b presented the concentrations of individual PFAS identified in the fluorochemical wastewater by targeted liquid chromatography Orbitrap mass spectrometry (LC-Orbitrap-MS) quantitative analysis with authentic standards. For PFAS without reference standards, semi-quantification was performed by comparing the MS signal intensities of their quasi-molecular ions with those of similar PFAS that were quantitatively analyzed, e.g., hydrogenated PFOA and PFOA. The profile of quantified PFAS was uniquely dominated by PFCA (63.9%). Significantly, PFOA displayed the highest concentration, reaching up to 58 µM (23.8 mg L⁻¹), which accounted for 48.7% of the concentration of ∑PFAS assessed (117.8 µM or 36 mg L⁻¹) and was considerably higher than any other reported samples by at least an order of magnitude. On the other hand, Ether-PFAA and H-PFAA had the highest prevalence with molar concentrations of 6% and 28% of all assessed PFAS, respectively. We also discovered several specific PFAS that underwent substitution by other halogen atoms, including Cl-PFAA and I-PFAA. The concentrations of these Cl/I-PFAA were all less than 1 µM, and their total molar concentration only accounted for 2.2% of all assessed PFAS. The assessed 107 PFAS are mainly carboxylic acids (82 species), with fewer sulfonic acids (25 species) (Fig. 1c). The carbon chain length of the 107 PFAS assessed ranges from C2 to C16, encompassing 47 short-chain PFAS (C < 7) with 9 being ultra-short-chain PFAS (C < 4) and 60 long-chain PFAS (C ≥ 7). In addition, the combustion-ion chromatography (CIC) test showed a total organic fluorine (TOF) concentration of 49.05 ± 2.51 mg L⁻¹, surpassing the combined concentrations of a total of 107 PFAS examined (∑PFAS = 36 mg L⁻¹). This implies the likelihood of other unobserved PFAS in the fluorochemical wastewater or underestimation of certain PFAS that lack authentic standards. The measured fluorochemical wastewater pH was neutral (pH = 7.3), and the presence of large amounts of background constituents included 35.86 ± 0.43 mg L⁻¹ total organic carbon (TOC), 26.22 ± 0.44 mg L⁻¹ total inorganic carbon (TIC), 388.42 ± 2.34 mg L⁻¹ chloride, 34.51 ± 0.92 mg L⁻¹ nitrate, 181.07 ± 4.18 mg L⁻¹ sulfate, and 44.52 ± 1.22 mg L⁻¹ fluoride (Supplementary Table 2).

Given that PFOA is the most significant PFAS in fluorochemical wastewater, we first examined the kinetics of PFOA removal from

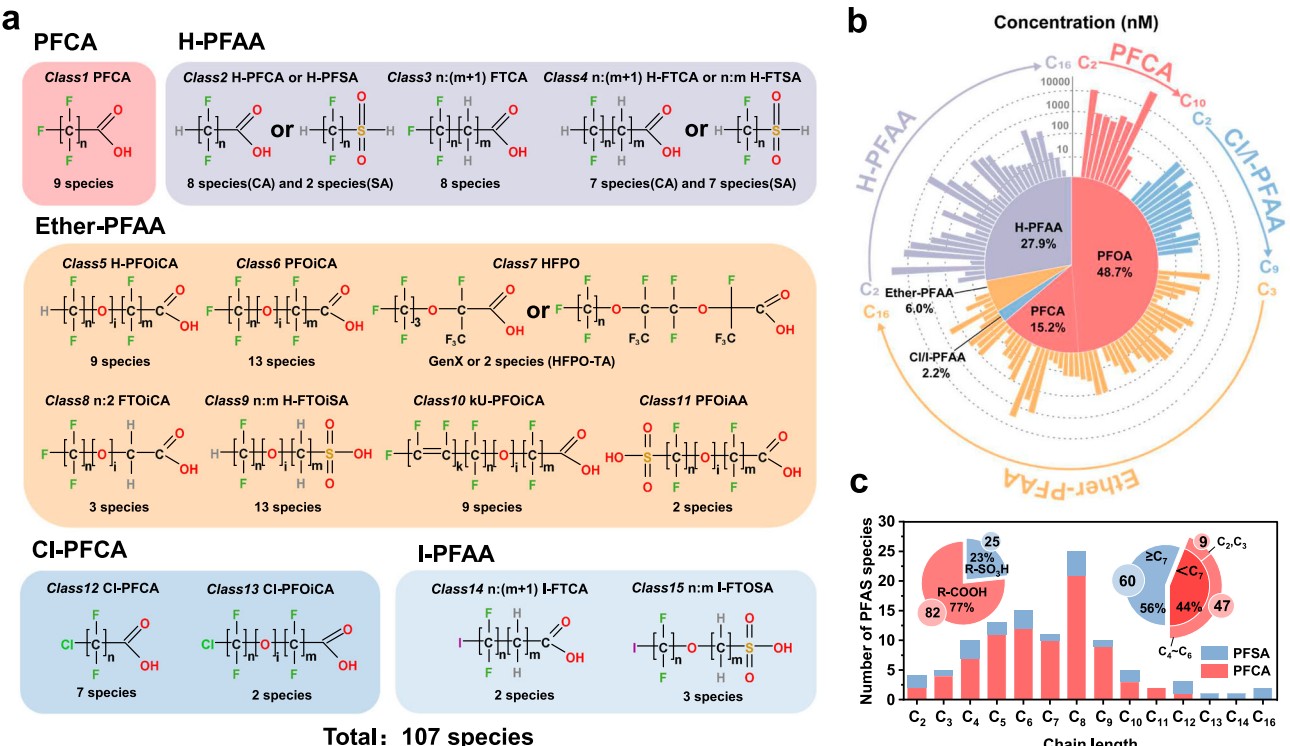

**Fig. 1 | Distribution of 107 PFAS species in fluorochemical wastewater.**
**a–c** Classification (**a**), concentrations (**b**), and chain length (**c**) distribution of the assessed 107 PFAS. The inset in (**c**) illustrates the numbers of sulfonic PFAS (R-$SO_3H$), carboxylic PFAS (R-COOH), long-chain PFAS (≥ C7), and short-chain PFAS (<C7). Wastewater samples were collected from the reverse osmosis concentrate of mixed effluents from polymer fluoropolymers production plants of a mega fluorochemical industrial park in China. The 107 PFAS included 9 PFCA, 32 H-PFAA, 52 Ether-PFAA, 9 Cl-PFAA, and 5 I-PFAA. The quantification of PFAS concentration were performed using liquid chromatography Orbitrap mass spectrometry, and the total concentration of 107 PFAS was determined to be 36 mg L$^{-1}$ or 117.8 μM.

fluorochemical wastewater using coconut shell AC and PFA694E AER as adsorbents, as well as EC system using Zn, Fe, and/or Al electrodes. As shown in Fig. 2a, the traditional Al-based and Fe-based EC systems were ineffective in removing PFOA from the fluorochemical wastewater with a removal rate of less than 20%; while sustained and rapid reduction of PFOA concentration was seen in the Zn-based EC system and achieved a 92 ± 1.5% removal after 30-min treatment. Although AC and PFA694E used here have high theoretical adsorption capacities (by Langmuir model, Supplementary Fig. 2) of 0.81 and 1.87 mmol PFOA g$^{-1}$, respectively; only a small amount of PFOA was removed, namely 16.5 ± 5.9% for AC and 26.9 ± 1.5% for PFA694E, after 60 min of sorption treatment at a high adsorbent dosage of 330 mg L$^{-1}$. It was noted that the quantity of Cl$^-$ (1416.4 ± 246.3 μmol g$^{-1}$) exchanged from the PFA694E significantly surpassed the amount of PFOA adsorbed (46.4 ± 2.6 μmol g$^{-1}$) (Supplementary Fig. 7a). This observation suggests that the elevated concentration of background inorganic anions in fluorochemical wastewater markedly competes with PFAS for adsorption sites. Wahman et al.[37] have previously reported that the exchange adsorption of PFAS by strong base AERs is affected by the presence of nitrite, sulfate, and bicarbonate. In this study, NO$_3^-$ (382.4 ± 9.9 μmol g$^{-1}$ adsorbed) and SO$_4^{2-}$ (389.4 ± 40.7 μmol g$^{-1}$ adsorbed) emerged as the predominant competing anions (Supplementary Fig. 7a), accounting for over 89% of the chloride ions exchanged out of PFA694E. Consequently, it is clear that effective pretreatment strategies are essential to reduce the interference of competing ions when utilizing AERs in the treatment of PFAS-laden industrial wastewater. The quantity of PFOA adsorbed ($q_t$) over time was calculated and depicted in Fig. 2b. Zinc hydroxide flocs generated in situ by Zn-based EC exhibited the highest sorption of PFOA with a magnitude of 76.1 ± 4.2 mg g$^{-1}$ (zinc hydroxide flocs), -14.1 and 5.8 times greater than that of AC (5.4 ± 0.4 mg g$^{-1}$) and PFA694E (13.2 ± 0.1 mg g$^{-1}$), respectively.

To assess the efficacy of eliminating all PFAS from fluorochemical wastewater utilizing AC, PFA694E, and Zn-based EC, changes in the concentrations of TOF (Fig. 2c) and 107 PFAS (Fig. 2d) were analyzed. Compared to AC (12.1 ± 4.9%) and PFA694E (19.5 ± 1.9%), the Zn-based EC system achieved a significantly higher TOF reduction (51.6 ± 7%). The bubble plots in Fig. 2d and Supplementary Figs. 3–6 displayed the PFAS arranged by m/z (X-axis) and carbon-chain length (Y-axis), with bubble diameters being proportional to their concentrations. The Zn-based EC system was greatly effective at reducing long-chain PFAS but less effective at removing short-chain PFAS. The PFA694E appeared to be able to remove all kinds of PFAS, but most PFAS had limited removal; AC was the least effective, failing to significantly remove any of the 107 PFAS. This is consistent with the results of a recent survey of 10 FIPs in China, which demonstrated that the existing AC adsorption processes removed only 0–13% of PFAS[38]. Further, we counted the removal rate of 51 PFAS with concentrations higher than 10 μg L$^{-1}$, as depicted in Fig. 2e. Specifically, 43 PFAS were removed at less than 30% after AC adsorption treatment, and the other 8 PFAS ranged from 30 to 50%; PFA694E adsorption achieved removal of 2 long-chain PFAS above 70%, but the vast majority of PFAS were removed at less than 50%. Impressively, as many as 21 (or 15, or 10) PFAS, mainly the long-chain PFAS, were removed greater than 70% (or 80%, or 90%) by the Zn-based EC system.

It is well known that the traditional Al/Fe-based (electro)coagulation process is frequently used as a pre-treatment process for adsorption processes because of its ability to efficiently remove DOMs and certain inorganic ions, as well as trapping colloidal particles from wastewater[39], and thereby reducing the adverse effects of these competing constituents on subsequent adsorption processes[40,41]. Our observations showed that the Zn-based EC process also significantly reduced TOC as well as F$^-$, SO$_4^{2-}$ and NO$_3^-$ (almost complete removal) in the fluorochemical wastewater (Supplementary Fig. 7c). Results from XRD analysis (Supplementary Fig. 8) showed that SO$_4^{2-}$ and NO$_3^-$

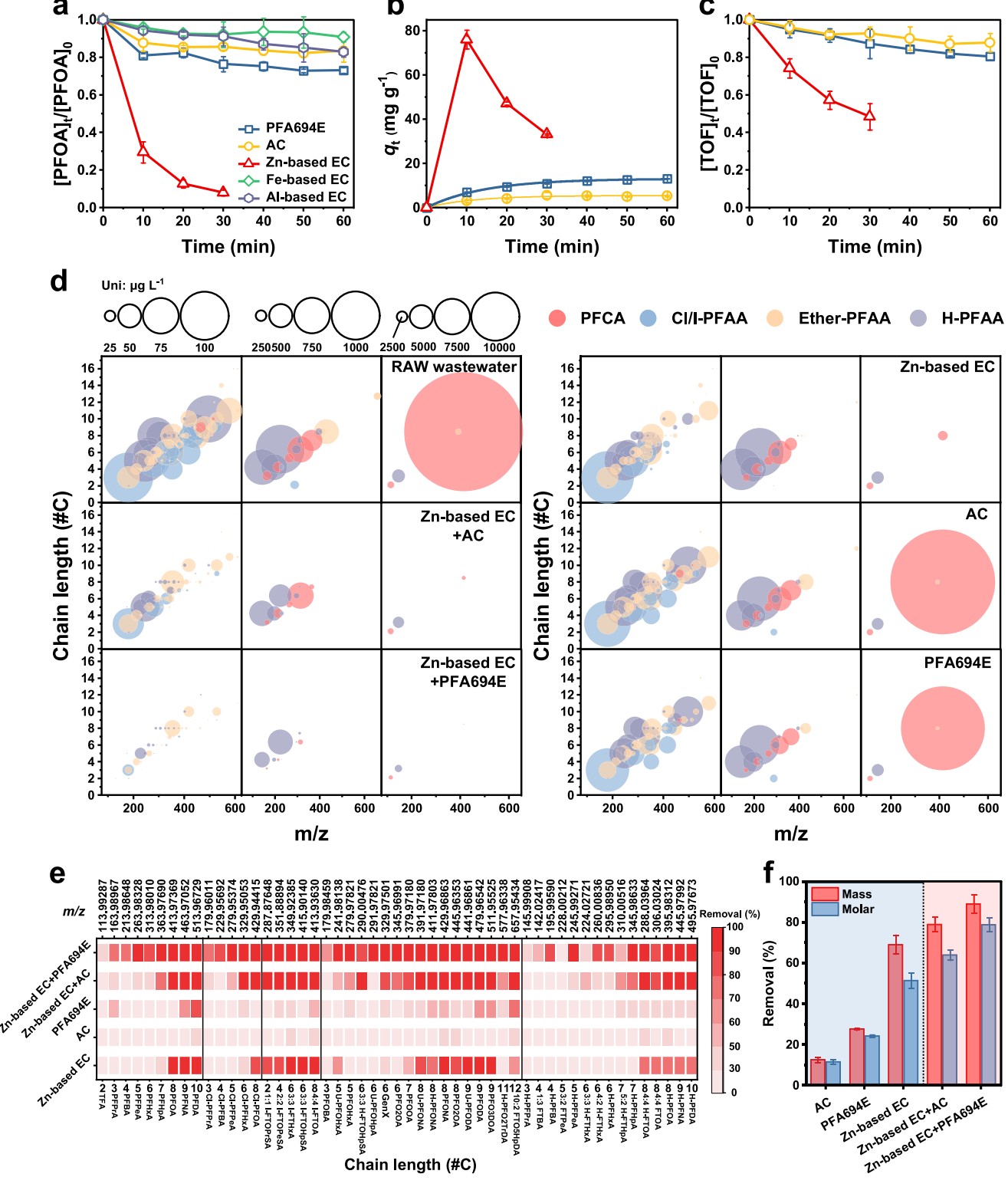

**Fig. 2 | Performance of different systems for the removal of PFAS from fluorochemical wastewater. a–d** Changes in the concentration of PFOA (**a**), the adsorbed amount ($q_t$) of PFOA (**b**), TOF (**c**), and 107 PFAS (**d**) vs. time. **e** Removal efficiencies to 51 PFAS with initial concentration > 10 $\mu$g L$^{-1}$. **f** The total concentration removal of 107 PFAS. Experiment conditions for AC and PFA694E adsorption: 100 mg AC or PFA694E, 300 mL fluorochemical wastewater. Experiment conditions for Al/Fe or Zn-based EC: 100 cm$^2$ of Al/Fe or Zn electrode, 300 mL fluorochemical wastewater, 3 mA cm$^{-2}$ of applied current density. All the error bars in this figure represent the standard deviation of the data from duplicate tests.

participate in the Zn$^{2+}$ hydrolysis reaction, leading to their removal as zinc hydroxide salts, i.e., zinc hydroxide nitrate (Zn$_5$(NO$_3$)$_2$(OH)$_8$, ZnHN) and zinc hydroxide sulfate (Zn$_4$SO$_4$(OH)$_6$, ZnHS). This is quite significant given that few technologies have been reported to be

capable of adsorbing large quantities of PFAS as well as simultaneously removing competing constituents from complex waste streams, which would greatly benefit the subsequent tandem conventional adsorption processes. It is encouraging that a proof-of-concept test a treatment-

train process involving Zn-based EC-coupled PFA694E adsorption achieved remarkable removal of all 107 PFAS (Fig. 2d). Out of the 51 PFAS of high concentrations, up to 35 PFAS were reduced in concentration by more than an order of magnitude (Fig. 2e). The total molar (mass) removal of 107 PFAS was $79.4 \pm 3.2\%$ ($89.4 \pm 3.9\%$), significantly higher than that of sorely PFA694E adsorption, which had a value of $24.4 \pm 3.0\%$ ($27.8 \pm 1.3\%$) (Fig. 2f). A similar synergistic effect, albeit slightly less effective, was also noted when coupling the Zn-based EC process with generic AC sorption.

## Zn-based EC for Selective adsorption of hydrophobic PFAS

**Mechanistic insights.** To examine the adsorption behavior of PFAS with different structures, the selective adsorption coefficient ($K_d$)[42] was introduced as a measure of PFAS adsorbability. The $K_d$ value represents the tendency of the interface to favor the adsorption of one specific PFAS over others, and a $K_d$ value greater than 1 for PFAS indicates that it will be preferentially adsorbed. The coefficient is calculated using the following Eq. 1:

$$K_d = \frac{\omega_b}{\omega_a} / \frac{(1 - \omega_b)}{(1 - \omega_a)} \qquad (1)$$

where $\omega_b$ and $\omega_a$ represent the molar concentration fraction of a specific PFAS to the total 107 PFAS in the fluorochemical wastewater before and after adsorption, respectively. Figure 3a illustrates the $K_d$ values of PFA694E for all 107 PFAS measured in the fluorochemical wastewater, which are located around 1 despite their wide variation in chemical structure and concentration. This finding is consistent with previous studies suggesting AER can adsorb a variety of ionizable PFAS[43,44]. However, the lack of strong specific affinity between AER and PFAS also makes it highly susceptible to interference from coexisting competing constituents, thereby reducing its effectiveness in removing PFAS in practical complex wastewater matrices. It is interesting to note that the Zn-based EC selectively adsorbs highly hydrophobic PFAS and ignores hydrophilic PFAS (Fig. 3b). Quantitative structure-activity relationship (QSAR) model fitting revealed a robust correlation between $K_d$ values and log of the $n$-octanol/water partition coefficient ($K_{ow}$) values (log $K_{ow} > 4$) of PFAS, with the equation $K_d = 9.5 \times \log K_{ow} - 38.6$ ($R^2 = 0.849$). It should be highlighted that all short-chain I-PFAA (C2 ~ C6) were preferentially adsorbed with $K_d > 3$, suggesting that all of them should be highly hydrophobic. However, the log $K_{ow}$ values (blue circle) estimated by the EPI Suite software for all short-chain I-PFAA are less than 4. Due to the lack of training set, current software

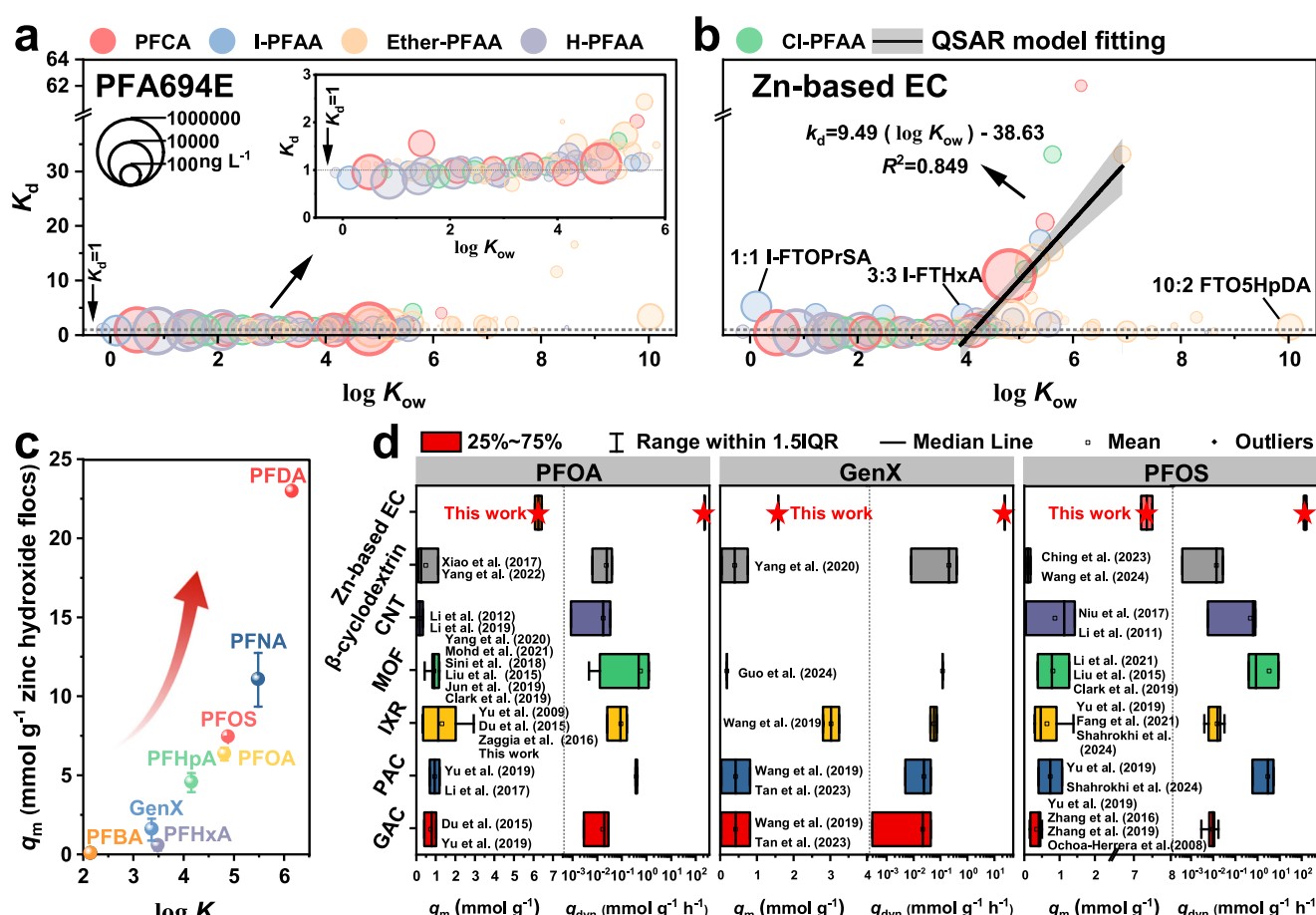

**Fig. 3 | Selective adsorption of PFAS. a, b** Relationship between log $K_{ow}$ and $K_d$ of 107 PFAS: PFA694E (**a**) vs. Zn-based EC (**b**). Inset in (**a**) is the enlarged relationship between log $K_{ow}$ and $K_d$ of PFAS (log $K_{ow} < 6.0$). Shaded areas in (**b**) are 95% confidence intervals for the quantitative structure-activity relationship (QSAR) model estimates. Experiment conditions for PFA694E adsorption: 100 mg PFA694E, 300 mL fluorochemical wastewater, 60 min of treatment time. Experiment conditions for Zn-based EC: 100 cm² of Zn electrode, 300 mL fluorochemical wastewater, 3 mA cm⁻² of applied current density, 20 min of treatment time. **c** The achieved maximum adsorption amount ($q_m$, mmol g⁻¹) of 8 PFAS (C4 - C10) in simulated solution by Zn-based EC vs. their log $K_{ow}$ values. Experiment conditions: 100 cm² of

zinc electrode, 300 mL simulated solution with 1.2 mM PFAS, 20 mM NaCl as supporting electrolyte, 1 mA cm⁻² of applied current density, pH = 7. **d** Comparison of the $q_m$ and dynamic adsorption capacity ($q_{dyn}$, mmol g⁻¹ h⁻¹) values for PFOA ($n = 23$), PFOS ($n = 25$) and GenX ($n = 10$) by Zn-based EC with various adsorbents in reported literature (listed in Supplementary Tables 4–6). The Centerline, upper limits and the lower limits of the box, upper whiskers and the lower whiskers, and the point in (**c**) represent the median, 25% of the maximum and minimum, the maximum and minimum, and the average of $q_m$ or $q_{dyn}$ in different technologies. All the error bars in this figure represent the standard deviation of the data from duplicate tests.

has poor accuracy in predicting the physicochemical properties of the PFAS with novel structural, e.g., Ether-PFAA and I-PFAA. In other words, these results suggest that the introduction of other atoms or structures, such as iodine atoms or ether groups, significantly alters the physicochemical properties of parent PFAS. This will be discussed in more detail later.

To elucidate the selective adsorption mechanism in the Zn-based EC process, we further investigated the adsorption kinetics of eight PFAS (for which reliable log $K_{ow}$ values were available) with varying chain lengths and structural characteristics in a simulated solution. Zinc hydroxide flocs presented rapid adsorption of PFAS with equilibrium time ($t_{eq}$) less than 2 min (Supplementary Fig. 9), whereas AC and AER widely used in current applications had $t_{eq}$ of tens of hours or more (Supplementary Tables 4–6). The observed maximum adsorption amount ($q_m$, mmol PFAS g$^{-1}$ zinc hydroxide flocs) was monotonically correlated with their hydrophobicity and chain lengths (Fig. 3c, Supplementary Table 3 and Supplementary Fig. 9), suggesting a pivotal role of hydrophobic interaction. The weakly hydrophobic perfluorobutanoic acid (PFBA, C4, log $K_{ow}$ = 2.14) had a $q_m$ value of < 0.1 mmol g$^{-1}$, the moderately hydrophobic perfluorohexanoic acid (PFHxA, C6, log $K_{ow}$ = 3.48) had an elevated $q_m$ of 0.6 ± 0.3 mmol g$^{-1}$, and the highly hydrophobic perfluorodecanoic acid (PFDA, C10, log $K_{ow}$ = 6.15) achieved an ultra-high $q_m$ of > 23 mmol g$^{-1}$ (>10 g g$^{-1}$). For the hexafluoropropylene oxide dimer acid (HFPO-DA or GenX, C6, log $K_{ow}$ = 3.36) with less hydrophobicity, the achieved $q_m$ was 1.6 ± 0.7 mmol g$^{-1}$, lower than that of AER but also several times higher than that of the AC and other reported adsorbents (Fig. 3d). However, the most discussed long-chain PFAS, i.e., PFOA (C8, log $K_{ow}$ = 4.81) and

PFOS (C8, log $K_{ow}$ = 4.88), their $q_m$ values were estimated to be as high as 6.4 ± 0.4 mmol g$^{-1}$ (2.6 g g$^{-1}$) and 7.5 ± 0.04 mmol g$^{-1}$ (3.8 g g$^{-1}$), respectively. To the best of our knowledge, these achieved $q_m$ for hydrophobic PFAS such as PFOA and PFOS are the highest of all values reported in the literature, which are over an order of magnitude higher than the theoretical maximum adsorption capacity derived from the adsorption model fitting that of the data for the benchmark AC and several times higher than that of the AER (Fig. 3d and Supplementary Tables 4–6). Furthermore, the dynamic adsorption capacity ($q_{dyn}$, mmol g$^{-1}$ h$^{-1}$) was more than 1 ~ 4 orders of magnitude higher than that of the benchmark AC and AER and other state-of-art adsorbents reported in the literature (Fig. 3d). The $q_{dyn}$ (mmol g$^{-1}$ h$^{-1}$) was calculated according to the equation, $q_{dyn} = q_m/t_{eq}$, in which the $t_{eq}$ represents the adsorption equilibration time[45]. The kinetics of PFAS adsorption by conventional porous adsorbents are primarily constrained by the diffusion process within the particles. In contrast, the adsorption of PFAS on the zinc hydroxide flocs predominantly occurs through a particle-surface adsorption process, as depicted in Fig. 4c, resulting in significantly faster adsorption rates.

The scanning electron microscopy (SEM) characterizations in Fig. 4a and Supplementary Fig. 10c showed that the presence of PFAS affects the morphology of zinc hydroxide flocs generated in situ by Zn-based EC. The fresh zinc hydroxide flocs were dispersed nanoflakes, while the PFAS-adsorbed zinc hydroxide flocs became dense. PFAS seemed to act as a binder to tightly aggregate and completely cover the dispersed Zn hydroxide flocs. As shown in Fig. 4b, the particle size of PFOA-adsorbed zinc hydroxide flocs had a much greater median volume diameter (DV$_{50}$) than that of the fresh zinc hydroxide flocs, i.e.,

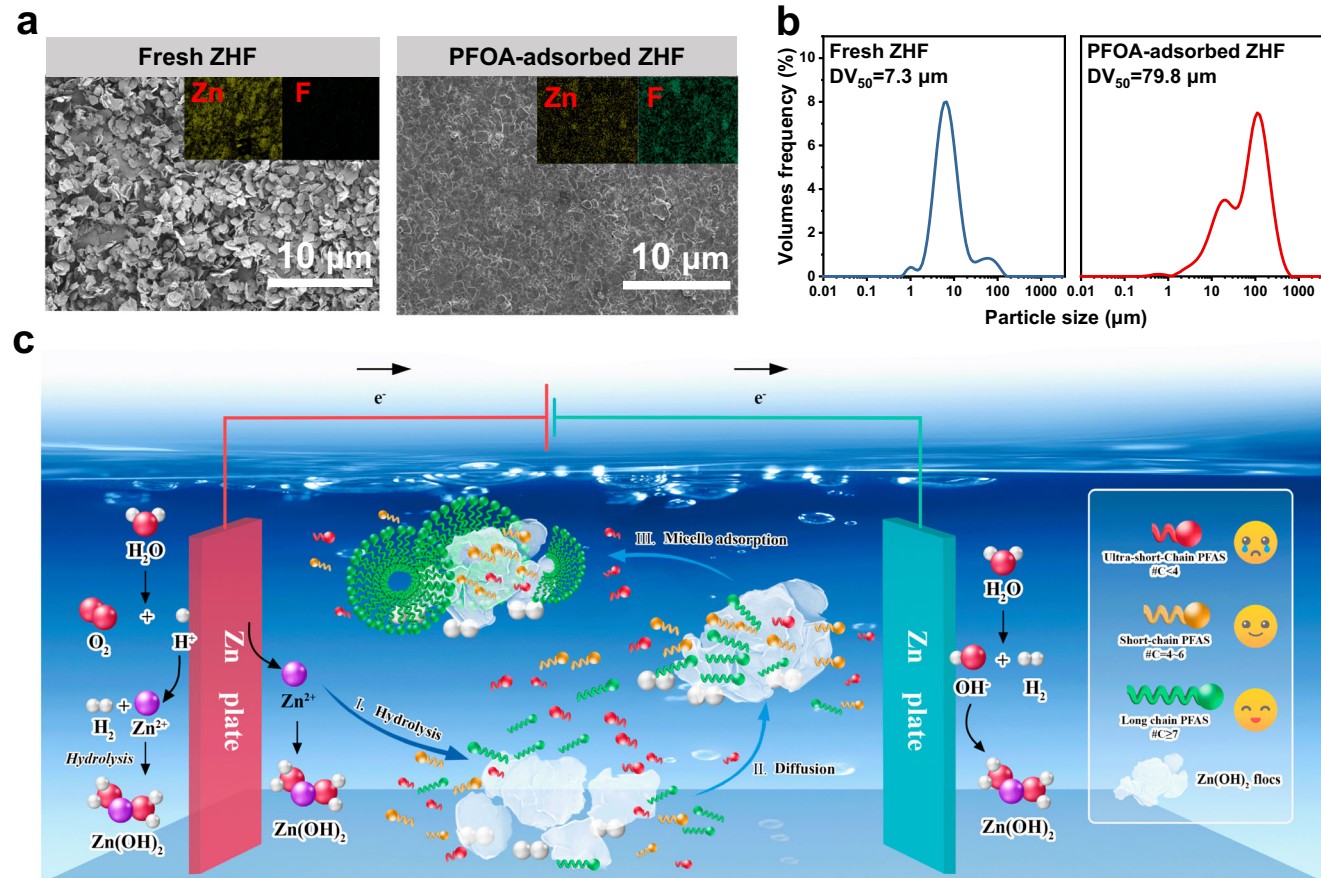

**Fig. 4 | Mechanisms for PFAS adsorption by Zn-based EC. a, b** SEM-EDX characterizations (**a**) and particle size distribution (**b**) of the fresh zinc hydroxide flocs (ZHF) and PFOA-adsorbed ZHF. Inset in (**a**) is the elements mapping of Zn (yellow) and F (green). ZHF in (**a, b**) represents the zinc hydroxide flocs. Experiment conditions for Zn-based EC: 100 cm$^2$ of Zn electrode, 300 mL simulated solution with or without 1.2 mM PFOA, 20 mM NaCl as supporting electrolyte, 1 mA cm$^{-2}$ of applied current density, pH = 7. **c** Schematic diagram of the proposed mechanisms for PFAS adsorption by Zn-based EC.

7.3 µm *vs.* 79.8 µm. This phenomenon is interesting, as it suggests that the adsorption of PFOA and the aggregation and growth of zinc hydroxide flocs occur simultaneously. In mineral flotation, the non-polar ends of long hydrocarbon chain traps adsorbed on the surface of hydrophobic mineral particles associate with each other to form semi-micelles, i.e., semi-micellar adsorption, by van der Waals forces[46]. Previous studies have shown that aluminum hydroxide flocs are mainly composed of hydrophilic colloidal aluminum hydroxide, ferric hydroxide species are also hydrophilic[47]. However, the flocs produced by Zn-based EC were observed to exhibit notable hydrophobic characteristics. The characterization results indicated that the zinc hydroxide flocs have a contact angle of ~ 90° (Supplementary Fig. 11), which is significantly greater than the contact angles of the aluminum (~ 60°) and ferric hydroxide flocs (~ 33°). Apparently, the zinc hydroxide flocs generated in situ by electrocoagulation were similar to the hydrophobic mineral particles. Inspired by the mineral flotation process, we propose a mechanism, i.e., hydrophobic force-driven semi-micellar adsorption, to clarify how Zn-based EC selectively absorbs hydrophobic PFAS (Fig. 4c). Initially, benefiting from the highly dispersed and high specific surface area of zinc hydroxide flocs (minerals), hydrophobic PFAS (trapping agent) can quickly move to their surface via hydrophobic force. Then, van der Waals forces induce high surface activity PFAS (e.g., long-chain PFAS) to create semi-micelles or micelles on their own, which leads to a significant improvement in their adsorption capabilities and ultimately results in ultra-high adsorption capacities. The adsorption of hydrophobic PFAS by adsorbents such as AC and AER also involves hydrophobic, and van der Waals interactions, so hydrophobic PFAS could also form semi-micelles or micelles on the surfaces of these adsorbents. However, intraparticle diffusion determines their adsorption of PFAS, the presence of semi-micelles or micelles on the external surface would impede the diffusion of PFAS into the internal micropores of these adsorbents and may even lead to a decrease in overall adsorption capacity[48].

Results from physicochemical characterizations of the PFAS-adsorbed zinc hydroxide flocs and theoretical calculations provided evidence for the proposed mechanism. Hydrophobic PFAS molecules would be adsorbed flat on the surface of zinc hydroxide flocs to minimize water-fluorine interactions[49]. Based on the molecular size of PFAS optimized by the Gaussian 09 (Supplementary Fig. 12), the spatial maximum number of PFOA ($11.61 \times 4.05 \times 3.98$ Å) and PFDA ($13.72 \times 4.05 \times 3.96$ Å) molecules per unit surface area for a monolayer of coverage was estimated to be less than 2.1 and 1.8 molecules per nm$^2$, respectively, assuming that the long axis (C-C chain) of the molecule is parallel to the surface and no space exists between molecules. Conversion of the molar mass of PFAS adsorbed per unit surface area results in approximately 18 PFOA molecules and >65 PFDA molecules per nm$^2$ of the zinc hydroxide flocs (BET = $213.1 \pm 10.6$ m$^2$ g$^{-1}$, Supplementary Fig. 13) according to their $q_m$ values, which is over an order-of-magnitude higher than the maximum number of molecules for a monolayer of coverage. The X-ray photoelectron spectroscopy (XPS) and scanning electron microscope-energy dispersive X-ray spectroscopy (SEM-EDX) characterizations further confirmed that the surface of zinc hydroxide flocs was tightly covered by the adsorbed-PFAS, as demonstrated by the intense F-element signals and significantly reduced Zn-element signals in Supplementary Fig. 10. It is evident that highly hydrophobic PFAS were adsorbed by multilayers, as demonstrated by the measured F/Zn atomic ratios of the PFOA/ perfluorononanoic acid (PFNA)/PFDA-adsorbed zinc hydroxide flocs, which were determined to be up to 6.35 ~ 7.54/ 7.35 ~ 7.9/ 8.85 ~ 9.96 (Supplementary Table 7). Weakly hydrophobic PFAS, usually the short-chain PFAS, have low surface activity and are unable to form semi-micelles or micelles on the surface of zinc hydroxide flocs. Furthermore, the electrostatic adsorption can also be neglected because of the zinc hydroxide flocs had a negative or weakly positive zeta potential (Supplementary Fig. 14). As a result, their adsorption capacity

is significantly lower compared to hydrophobic PFAS. In principle, this unique mechanism would also enable the Zn-based EC to avoid adverse effects from other coexisting contaminants and DOMs in solution, as these competing constituents tend not to be highly hydrophobic. Therefore, the Zn-based EC could be an effective technique for treating highly complex waste streams such as aqueous film-forming foams solution, landfill leachate, and still bottom liquid waste containing high concentrations of PFAS from adsorbent regeneration.

**Effect of the structure of PFAS on their adsorbability.** As the use of legacy perfluorinated $C_nF_{2n+1}$–X (X = COO$^-$ or SO$_3^-$) compounds has been restricted, many alternative PFAS have been created and extensively applied for fluorochemical production. The main substitution strategy involves inserting –H, –Cl, –OH or –O– into perfluorinated molecules, which reduces the "effective length" of fluorinated chain segments to reduce their persistence. Here, we sought to explore the adsorbability of various PFAS in terms of their chemical structure, which may provide critical guidance for the design of alternative PFAS that are easier to eliminate. Figure 5a illustrated the relationship between the $K_d$ value of PFAS and the ratio of F/C as well as F/H in their chemical structure. The preferentially adsorbed PFAS were mainly concentrated in the upper-right quadrant region with high F/C and F/H ratios. This suggests that for a PFAS to be preferentially adsorbed, it must satisfy two structural conditions: (1) a high degree of fluorination (e.g., F/C > 1.6), and (2) a large number of fluorine atoms (e.g., >8). It is evident that the physical and chemical properties of PFAS, as well as their environmental behavior, are profoundly determined by the number and distribution of [CF$_n$] ($n = 1 - 3$) and [CH$_n$] ($n = 1 - 3$) in their chemical structure. Figure 5b showed how the length of the [CF$_n$] chain and the number of hydrogen atoms attached to the carbon in the PFAS molecule affect its $K_d$. Typically, PFAS with 6 or more [CF$_n$] units would be preferentially adsorbed. However, [CH$_n$] groups may significantly reduce their adsorbability. For instance, although some PFAS have 7 or 8 [CF$_n$] units in their chemical structure, but the existence of multiple [CH$_n$] groups (e.g., >5) can offset the benefits of the [CF$_n$] units. Ether-PFAA are the most abundant class of PFAS in the fluorochemical wastewater. Figure 5c illustrates the impact of the number of carbon-ether bonds (C–O–C) on the $K_d$ value of Ether-PFAA (with PFCA and iodi-nated PFPESA as controls). In general, the existence of C–O–C bonds is likely to result in lower adsorbability regardless of the structure of the Ether-PFAA. For instance, the $K_d$ value of PFPECA (green circle) was consistently lower than that of the corresponding chain length of perfluoroalkyl acid (blue circle), and a plurality of C–O–C bonds further reduced its $K_d$ value. These results clearly indicated that PFAS alternatives with –H and/or –O– introduced into the perfluorinated structure ($C_nF_{2n+1}$–) tend to be less adsorbable.

Notably, the behavior of the novel I-PFAA was markedly different. Their chemical structure contains 1 to 3 [CH$_n$] units and no more than 4 [CF$_n$] groups (Fig. 5b), placing them in the lower-left quadrant region with low F/C and F/H ratios in Fig. 5a. In principle, they should not be preferentially adsorbed. However, all measured I-PFAA had $K_d$ values much greater than 1. For example, 4:4 I-FTOA (CH$_2$I[CF$_2$]$_4$[CH$_2$]$_2$COOH) has a composition of 3[CH$_2$] groups and only 4[CF$_2$] units, yet it had a high $K_d$ value of 17.39. Similarly, 3:3 I-FTHxA (CF$_2$I[CF$_2$]$_2$[CH$_2$]$_2$COOH) with even less [CF$_2$] also had a $K_d$ value of 3.75. Both of them had higher $K_d$ values than the corresponding chain-length perfluorocarboxylic acids, i.e., PFOA ($K_d = 10.9$) and PFHxA ($K_d = 0.53$), respectively. Furthermore, we observed that the short-chain and ultra-short-chain iodinated ether-FTSAs (I-FTPESAs) were also preferentially adsorbed (Fig. 5d). For example, 1:1 I-FTOPrSA (CH$_2$I-O-CF$_2$SO$_3$H) and 2:2 I-FTOPeSA (CF$_2$ICH$_2$-O-CH$_2$CF$_2$SO$_3$H) achieved $K_d$ values of 5.28 and 3.68, respectively. It is widely recognized that chlorinated PFAS (Cl-PFAS), as a class of alternative PFAS, have been developed and used extensively in commercial products and industrial materials for

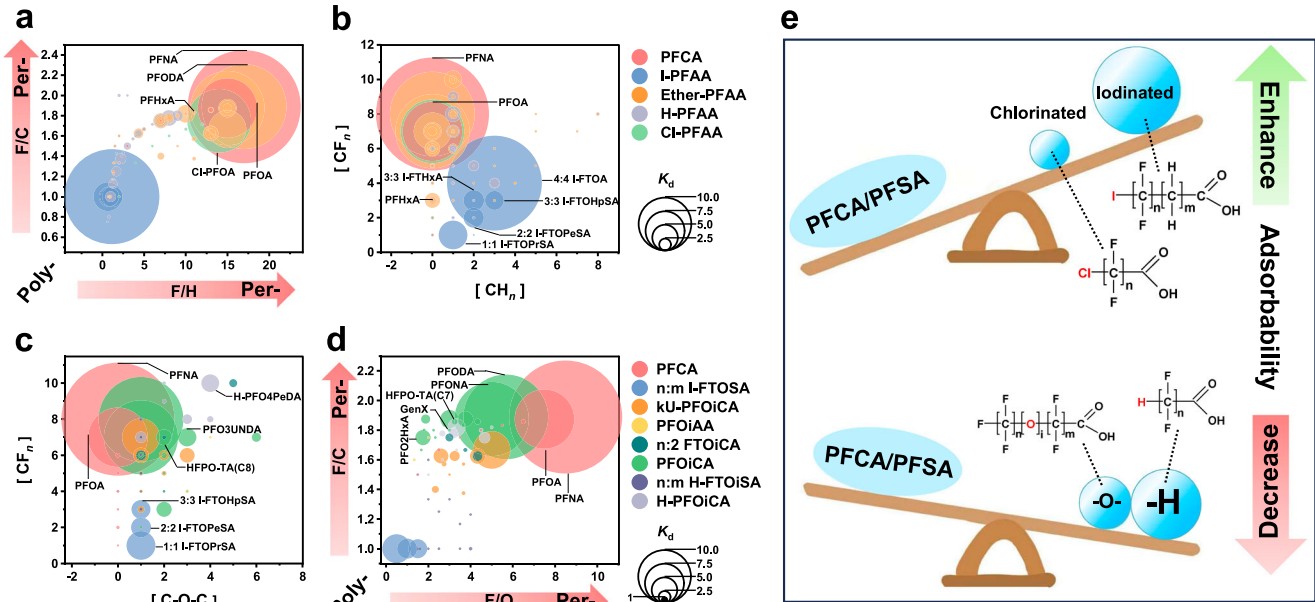

**Fig. 5 | Effect of the chemical structure of PFAS on their adsorbability. a, b** F/C ratio, F/H ratio of 107 PFAS $vs.$ their $K_d$ values (**a**), and [$CF_n$] ($n=1$ - 3), [$CH_n$] ($n=1$ - 3) numbers of 107 PFAS $vs.$ their $K_d$ values (**b**). **c, d** [$CF_n$] ($n=1$ - 3), [C–O–C] numbers of 52 Ether-PFAA $vs.$ their $K_d$ values (**c**), and F/C ratio, F/O ratio of 52 Ether-PFAA $vs.$ their $K_d$ values (**d**). Experiment conditions for Zn-based EC: 100 cm² of Zn electrode, 300 mL fluorochemical wastewater, 3 mA cm⁻² of applied current density, 20 min of treatment time. Poly- and Per- in (**a, d**) represent polyfluorinated and perfluorinated, respectively. **e** Schematic diagram of the potential effect of inserting –H, –Cl, –I or –O– into the structure of PFAS molecules on their adsorbability.

decades[50]. Ten Cl-PFAS (Supplementary Table 1) including 8 Cl-PFCA (ranging from C2 to C9) and 2 chloroperfluoropolyether carboxylates (Cl-PFPECA) found in the fluorochemical wastewater. The study found that Cl-PFAS and non-chlorinated PFAS have similar or slightly higher $K_d$ values, e.g., Cl-PFOA (CF₂Cl[CF₂]₆COOH, $K_d = 11.65$) $vs.$ PFOA ($K_d = 10.9$), but none of the short-chain Cl-PFCAs showed preferential adsorption (Fig. 5a). These results suggested that the Cl substitution (Cl→F) has little effect on the adsorbability of PFAS, but substitution of even one fluorine atom in PFAS with an iodine atom (I → F) causes a dramatic shift in their chemical properties. This finding is significant because it suggests that the potential environmental impacts of I-PFAA may differ significantly from those of traditional non-iodinated PFAS. A schematic diagram (Fig. 5e) was drawn to depict the potential effect of inserting –H, –Cl, –I, or –O– into the structure of PFAS molecules on their adsorbability. The novel structural feature and important environmental relevance (e.g., several hundred µg L⁻¹ in the fluorochemical wastewater) of iodinated PFAS require an adequate understanding of their environmental behavior and fate. Alternatively, a recent study by Jin et al.[21] showed that the substitution of F with Cl significantly improved the biodegradability and reduced the toxicity of PFAS. Replacing F with I could further enhance this effect due to the larger radius of the iodine atom. The experimental results of electro-oxidative treatment of the fluorochemical wastewater demonstrated that the degradation rate of PFAS with the same chain length followed: I-PFAA >> Cl-PFAA > Ether-PFAA, H-PFAA and PFCA (Supplementary Fig. 15). In addition, it was observed that the total I-PFAA concentrations in a sealed white polyethylene plastic drum stored at ambient temperature decreased by 52% over a 13-month period[51], suggesting that I-PFAA is susceptible to degradation under natural conditions. The presence of I-PFAA in the fluorochemical wastewater may be attributed to the unintentional by-products of iodine transfer polymerization/copolymerization processes during the synthesis of fluoropolymers, where iodofluoroalkanes are used as chain transfer agents[51]. However, their specialized properties provide insights into the design of alternative iodinated PFAS that are readily degradable.

## Full-scale simulations of Zn-based EC-coupled PFA694E Bed

To confirm the efficacy of the treatment-train strategy of Zn-based EC-coupled existing full-scale conventional adsorption units, a rapid small-scale column test (RSSCT) breakthrough experiment with fluorochemical wastewater was conducted as a proof-of-concept experiment. A scale-up test of the Zn-based EC was conducted with a volume of 20 L (Supplementary Fig. 16), yielding results that were largely consistent with those of the bench-scale test. A constant diffusivity design was chosen for the RSSCTs. A recent study[52] has shown that, with proper interpretation, this approach can lead to useful approximations of pilot- or full-scale system performance. The crushed PFA694E exhibited a comparable total anion exchange capacity to the as-received (6.6 ± 0.4 $vs.$ 5.9 ± 0.4 mmol Cl⁻ per mL resin, Supplementary Table 8). The RSSCT design parameters and the simulated full-scale PFA694E bed parameters are presented in the Method section and in the Supplementary Table 9. First, we examined the breakthrough curve of the PFA694E bed fed by a simulated solution containing 25 mg L⁻¹ of PFOA, at a concentration consistent with fluorochemical wastewater. The results showed that the values of bed volume (BV) to 80% (BV₈₀) breakthrough were estimated to be $15 \times 10^3$ BV (Fig. 6a). Unexpectedly, the PFA694E bed fed with untreated fluorochemical wastewater was breached in a very short time, with the values of bed volume to 10% (BV₁₀), 50% (BV₅₀), and 80% (BV₈₀) breakthrough being 0.39 ×, 0.54 × and 0.66 × $10^3$ BV, respectively (Fig. 6a).

The concentration of TOF and other 106 PFAS in the effluent of the PFA694E bed was also monitored. As shown in Fig. 6b, the TOF profile of the PFA694E bed was also rapidly breached, suggesting that a quick breakthrough occurred for all PFAS. When fed only $1.03 \times 10^3$ BV of untreated fluorochemical wastewater, the PFA694E bed showed almost complete breakthrough of the monitored 107 PFAS (Fig. 6c and Supplementary Fig. 17), and the effluent TOF removal was less than 10% (Fig. 6b). In contrast, Zn-based EC-coupled PFA694E bed adsorption treatment-train strategy achieved at least 50% removal of TOF throughout the treatment. At the point of $1.03 \times 10^3$ BV, the treatment-train strategy was still able to remove most of the monitored 107 PFAS

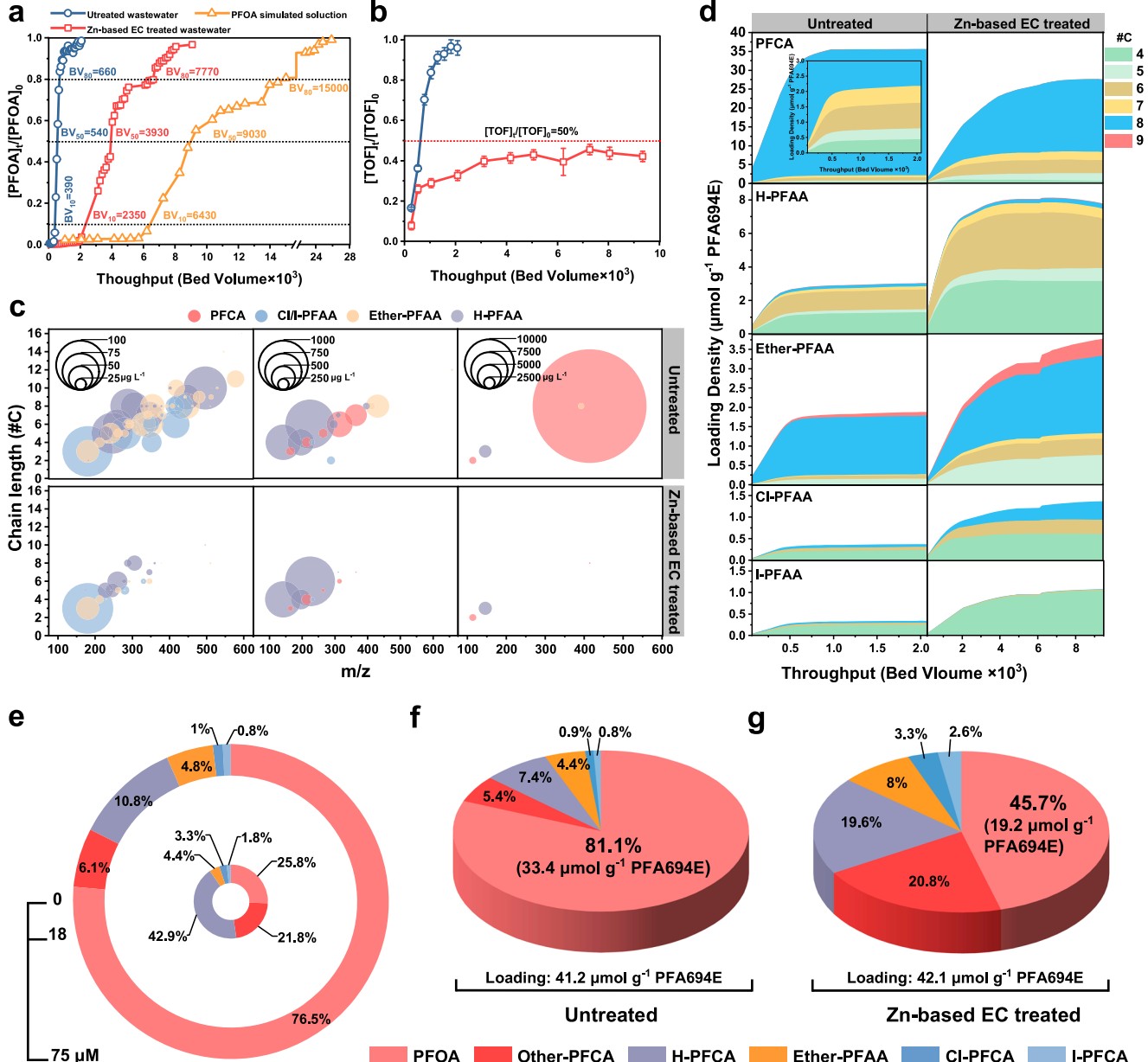

**Fig. 6 | Full-scale system simulations for fluorochemical wastewater purification. a, b** RSSCT breakthrough curves of PFOA (**a**) and TOF (**b**) for the PFA694E adsorption bed fed by untreated or Zn-based EC-treated fluorochemical wastewater. **c** Concentrations of 107 PFAS in the influent at $1.03 \times 10^3$ BV for the PFA694E adsorption bed fed by untreated or Zn-based EC-treated fluorochemical wastewater. **d** Molar adsorption density profiles of 23 representative PFAS on the PFA694E adsorption bed. Inset in (**d**) is the enlarged molar adsorption density profiles of PFCA (except PFOA). **e** Total molar concentrations of 23 representative PFAS and their percentage contribution in fluorochemical wastewater: untreated (outer ring) *vs.* Zn-based EC treated (inter ring). **f, g** Finite loading and percentage of different categories of PFAS onto the PFA694E bed fed with untreated (**f**) or Zn-based EC treated (**g**) fluorochemical wastewater. The 23 representatives of PFAS included 6 PFCA, 6 H-PFAA, 5 Ether-PFAA, 3 Cl-PFAA, and 3 I-PFAA. All the error bars in this figure represent the standard deviation of the data from duplicate tests.

(Fig. 6c and Supplementary Fig. 17) and achieved 70% removal of TOF (Fig. 6b). For example, nearly 99% of PFOA was removed by the treatment-train strategy with an effluent concentration of only 0.3 mg L$^{-1}$, compared to 22.3 mg L$^{-1}$ in the effluent from the PFA694E bed fed with untreated fluorochemical wastewater. As a result, the front-end Zn-based EC treatment resulted in a 12.3-fold increase in BV$_{80}$ for PFOA, with a value of $7.77 \times 10^3$ BV (Fig. 6a).

In addition to PFOA, we also monitored the breakthrough curves of 22 other representative PFAS with carbon-chain lengths ranging from 4 to 9 in fluorochemical wastewater at relatively high concentrations. As expected, the front-end Zn-based EC treatment significantly delayed the full breakthrough of the PFA694E bed

(Supplementary Fig. 18), resulting in a 2.5-fold (PFBA) to 13.6-fold (Cl-PFOA) increase in BV$_{80}$ values (Supplementary Fig. 19). Overall, PFAS with high log $K_{ow}$ values are more likely to achieve higher enhancement folds. A comparison of PFAS mass loading on a molar concentration basis identified distinct adsorption behaviors on the PFAE694E bed when combined with the front-end Zn-based EC treatment. As shown in Fig. 6d, the cumulative adsorption mass profiles of all PFAS flattened, indicating that the PFA694E bed had reached its maximum PFAS adsorption capacity during the RSSCT test. The PFA694E bed retained a final PFAS loading of 41.2 µmol g$^{-1}$ adsorbent fed with untreated fluorochemical wastewater, consisting of 81.1% of PFOA (compared to 76.5% PFOA in feed, Figs. 6e) and 18.9% of other 22

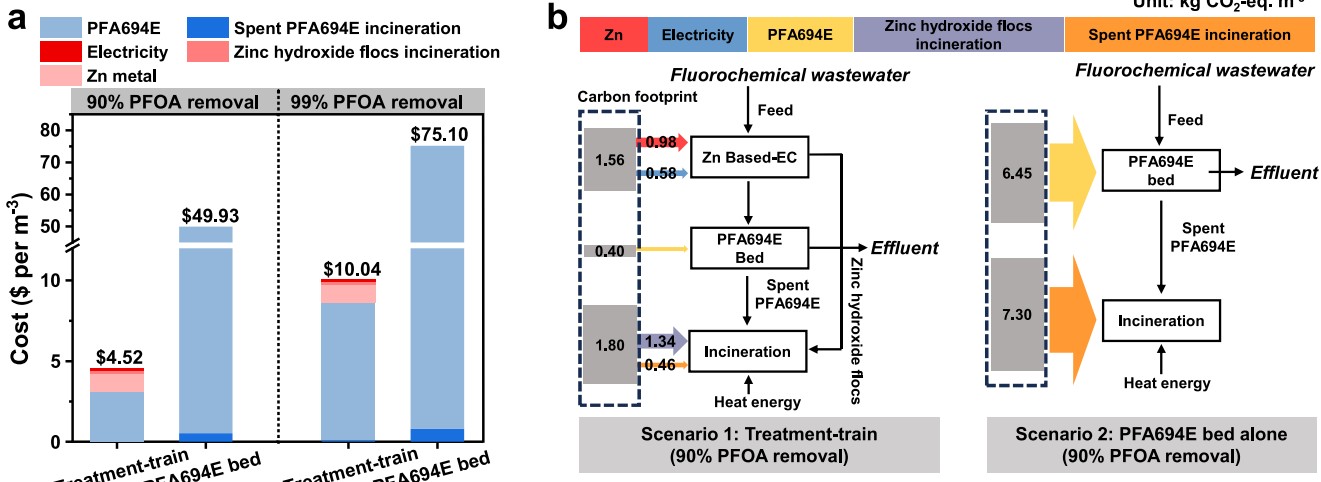

**Fig. 7 | Cost and environmental impact analysis. a, b** Techno-economic analysis (**a**) and life-cycle environmental impact analysis (**b**) for the treatment of fluorochemical wastewater by the PFA694E adsorption bed or the treatment-train processes. Treatment-train represents the Zn-based EC-coupled PFA694E adsorption bed process. The PFA694E adsorption bed replacement criterion set at >90% or 99% PFOA removal.

PFAS. Although the front-end Zn-based EC treatment resulted in a 76% reduction in the total concentration of the 23 PFAS in the fluorochemical wastewater, from 75 μM to 18 μM (Fig. 6e), the final loading of the 23 PFAS on the PFA694E bed, was even higher than that of the feed of untreated fluorochemical wastewater, i.e., 42.1 vs. 41.2 μmol g⁻¹ (Fig. 6f, g). More importantly, the total loading of the other 22 PFAS was dramatically increased by over 3 times to 22.9 μmol g⁻¹, accounting for 54.3% of the final loading of the PFA694E bed. The final loading of each PFAS is shown in Supplementary Fig. 20. It can be observed that significant increases in loadings for most of the 23 PFAS, except for several highly hydrophobic PFAS (such as PFOA, PFNA, 3:3 I-FTHxA and 4:4 I-FTOA) whose concentrations were dramatically or nearly completely removed by the front-end Zn-based EC treatment. These results highlighted that to maximize the usable adsorbent bed service life and achieve broad-spectrum removal of dozens or hundreds of PFAS in real-world complex scenarios, a potential treatment configuration could use a combination of Zn-based EC treatment and adsorbent in series.

Furthermore, the concentrations of residual zinc in the effluent of the Zn-based EC and RSSCT column were also determined. The residue zinc concentration in the treated fluorochemical wastewater was 0.84 mg L⁻¹ after Zn-based EC and less than 0.5 mg L⁻¹ after the RSSCT column (Supplementary Fig. 21). Given that zinc is an essential semi-trace element and the drinking water limits set by the World Health Organization (WHO) is 3 mg L⁻¹ [53], using Zn-based EC for water treatment is considered safe.

### Techno-economic analysis and life-cycle environmental impact

The application prospects of the proof-of-concept treatment-train strategy were also evaluated based on carbon footprint and techno-economic analysis. In this study, the PFA694E bed was operated as a single-use adsorbent, and the spent PFA694E and zinc hydroxide flocs with adsorbed PFAS would be incinerated. As shown in Fig. 7a and Supplementary Table 10, the operational cost of the Zn-based EC process was approximately $1.43 per m³ treated under a treatment time of 20 min, consisting of $1.14 for zinc metal cost, $0.1 for the electricity, and $0.17 for the incineration of zinc hydroxide flocs (Moisture content not considered). Assuming a base case scenario where adsorption bed change-out criteria are dictated by PFOA removal less than 90%, the predicted operating cost of the treatment-train strategy was $4.52 per m³ treated. In the absence of front-end Zn-

based EC treatment, the estimated operating cost would greatly increase to $49.94 per m³ treated for the PFA694E bed system, over an order-of-magnitude higher than those of the treatment-train strategy. A cradle-to-grave life-cycle assessment was employed to compute the carbon footprint of both the Zn-based EC and the PFA694E adsorption bed systems. The PFA694E adsorption bed system had a carbon footprint of 13.75 KgCO₂ per m³ treated under the change-out criteria of >90% of PFOA removal (Fig. 7b and Supplementary Table 10). Setting the same change-out criteria, the carbon footprint of the treatment-train strategy, i.e., 3.76 KgCO₂ per m³ treated, was only 27.3% of that of the PFA694E adsorption bed alone. Unlike PFA694E, which is considered a "high-carbon" adsorbent due to the large amounts of CO₂ emitted during its incineration (1.77 ± 0.02 KgCO₂ per KgPFA694E), the inorganic zinc hydroxide flocs generated in situ by Zn-based EC are essentially a "zero-carbon" adsorbent. Additionally, CIC and TOC/N analyzer tests showed that the PFA694E contains 0.6 ± 0.01 mmol S, 2.42 ± 0.1 mmol N, and 0.39 ± 0.01 mol Cl per gram, suggesting that incineration treatment of the spent PFA694E will produce significant amounts of other pollutants, such as SO₂, ozone, smoke particles, carcinogenics, and NOx. Significantly, Zn-based EC has a much lower environmental impact compared to the reported adsorbents mainly carbon material-based adsorbents[54]. We would also like to highlight that the amount of dissolved metallic zinc determines the cost and environmental impact of the Zn-based EC treatment. Therefore, developing methods to produce zinc hydroxide flocs with enhanced PFAS adsorption capabilities is an important area for future research. Furthermore, the resourceful reuse of waste zinc hydroxide flocs to reduce the cost of Zn-based EC treatment should be considered.

### Critical implications for PFAS research

The importance of addressing PFAS emissions from industrial production should be of great concern, as it is still the most important source of PFAS entering the environment in many countries and regions, such as China. However, the high concentration and diversity of PFAS in industrial wastewater, as well as the complex background matrix, pose significant challenges to existing adsorption technologies. This study is the critical initial step in developing a treatment-train strategy that couples a Zn-based EC process with existing adsorption devices (e.g., AER and AC) to achieve the efficient and broad-spectrum capture of hundreds of PFAS with diverse properties from a fluorochemical

industrial park effluent. The zinc hydroxide flocs generated in situ by Zn-based EC can selectively and rapidly ($t_{eq} < 2$ min) adsorb hydrophobic PFAS (log $K_{ow} > 4$) via a semi-micellar adsorption mechanism similar to that of mineral flotation. This unique multilayered adsorption mechanism outweighs the conventional adsorbents that are based on their limited adsorption sites to remove PFAS, and results in Zn-based EC with the highest adsorption capacities (e.g., 6.4 mmol PFOA per gram of zinc hydroxide flocs) to hydrophobic PFAS among all adsorbents reported. Meanwhile, the Zn-based EC is also capable of substantially removing the coexisting competing constituents such as DOMs and $NO_3^-$ to the conventional adsorbents. These features are quite important for the subsequent tandem adsorption processes, as they greatly extend their lifetime, enhance their adsorption selectivity and adsorption capacity for short-chain PFAS, and reduce operating costs. Furthermore, the zinc hydroxide flocs are essentially inorganic "zero-carbon" adsorbents that significantly reduce the environmental impact of the treatment-train strategy. Therefore, the treatment-train methodology could be a potential upgrade of existing adsorption devices.

Electrocoagulation is a mature and simple pretreatment process that has been widely used in water treatment for decades. The Zn-based EC process is notably versatile, allowing for integration with various other techniques to remove PFAS from complex wastewater streams. Its most promising application involves coupling with existing conventional adsorption processes and membrane separation technologies. This not only efficiently removes hydrophobic PFAS from wastewater but also reduces competing constituents in tandem adsorption processes or mitigates membrane fouling. Because zinc hydroxide flocs are readily soluble in acidic or alkaline solutions, the Zn-based EC process, therefore, could also be employed as a pre-enrichment approach for PFAS. This approach can be combined with various destruction technologies, such as advanced redox processes and pyrolysis, to achieve cost-effective results. For example, aqueous film-forming foams contain substantial PFAS; however, the efficacy of direct treatment by advanced redox processes is limited due to the coexistence of numerous additional components, including sodium dodecyl sulfate (SDS), a hydrocarbon surfactant, and diethylene glycol butyl ether (DGBE), an organic solvent. Given the poor hydrophobicity of SDS (log $K_{ow} = 1.69$) and DGBE (log $K_{ow} = 0.29$), our preliminary results indicate that the Zn-based EC can achieve selective enrichment of PFAS from aqueous film-forming foams, which can then be effectively degraded by advanced redox processes. It is important to note that zinc is a toxic heavy metal element. The safe disposal and utilization of zinc hydroxide floc sludge are vital considerations for the potential application of the Zn-based EC process. Pyrolysis, including incineration and low-temperature alkaline hydrothermal processes, may be applicable to recover Zn. In addition, our studies show that zinc hydroxide flocs generated in a simulated solution with NaCl as the supporting electrolyte exhibited superior PFAS adsorption capabilities. Therefore, a comprehensive investigation into the hydrolysis behavior of $Zn^{2+}$ and the identification of the key hydrolysis products that are responsible for the adsorption of PFAS is warranted, which is crucial for optimizing the Zn-based EC process.

On the other hand, fluorochemical-related industries have been crucial to modern socio-economics. As the use of traditional PFAS is gradually restricted, more and more new alternatives are being developed and widely used. However, many of them have come under global scrutiny of new concerns, e.g., GenX, for exhibiting similar persistence and biotoxicity as traditional PFAS. For the future design of specialty PFAS products, we need to maximize their eliminability and reduce their biotoxicity and persistence while maintaining desirable properties. Our experimental results evidence the inclusion of −H and −O− into perfluorinated structure decreases the adsorbability compared to the parent PFAS. A recent study by Jin et al.[21] highlighted that replacing one or more F atoms with Cl atoms in PFAS structures, known as Cl-PFAS, could be an effective strategy to improve their

biological and chemical degradability without increasing toxicity. In this study, Cl-PFAS does not show a significant improvement in adsorbability; rather, the substitution of even a single iodine atom impressively alters the nature of the parent PFAS, greatly enhancing its hydrophobicity and allowing iodinated PFAS to be readily adsorbed. In addition, we also observed the degradation of iodinated PFAS under natural conditions and significantly faster degradation kinetics than Cl-PFAS under electrooxidation. These findings provide not only critical fundamental knowledge into the assessment of the environmental fate of iodinated PFAS, but can also help to design environmentally-friendly PFAS and achieve sustainable management of fluorochemicals.

## Methods
### Samples and chemicals
Wastewater samples with a concentration of ∑PFAS of 36 mg L$^{-1}$ were taken from the reverse osmosis (RO) concentrate of mixed effluents from multiple production plants of a mega fluorochemical industrial park (FIP) in northern China, where various PFAS were extensively used during the production of polymer fluoropolymers, including polyper-fluoroethylene propylene (FEP), polytetrafluoroethylene (PTFE), and polyvinylidene fluoride (PVDF). Ammonium acetate (NH$_4$AC, 5 mol L$^{-1}$ in 100 mL H$_2$O), sodium chloride (NaCl, 99.9%), and all PFAS chemicals such as hexafluoropropylene oxide dimer acid (HFPO-DA or GenX, >98%), perfluorobutanoic acid (PFBA, >98%), perfluorohexanoic acid (PFHxA, >97%), perfluoroheptanoate (PFHpA, >97%), perfluorooctanoic acid (PFOA, >95%), perfluorononanoic acid (PFNA, >97%), per-fluorodecanoic acid (PFDA, >98%), and perfluorooctane sulfonate (PFOS, >98%) used in the experiments were purchased from Sigma-Aldrich Chemical Co., Ltd. The mass-labeled perfluorinated compounds EISs solution (MPFAC-C-ES, 13 $^3$C-labeled PFAS) were purchased from Wellington Laboratories Inc. Details of these internal standards are provided in Supplementary Table 11. Methanol (MeOH) and acetonitrile (ACN) were chromatographic grade and purchased from Merck Corp. Granular activated carbon (GAC, 20 - 40 mesh) was purchased from Macklin Biochemical Co., Ltd; PFA694E anion exchange resin (AER) was obtained from Purolite® (Bala Cynwyd, PA, USA).

### Electrocoagulation, batchadsorption and RSSCT experiments
The electrocoagulation (EC) reactor was composed of a cylindrical EC cell (8 cm diameter and 15 cm height, Supplementary Fig. 1a) with a 500 mL volume, and was equipped with a gas stirring device (0.3 L N$_2$ per min), as shown in Supplementary Fig. 1a. A metal (Zn, Al and/or Fe) sheet of 100 cm$^2$ surface area was used as the anode, while a 304 stainless steel rod of 0.3 cm diameter was used as the cathode, with a distance of 3 cm between the electrodes. In each experimental run, 300 mL of fluorochemical wastewater or simulated solution was added and operated in a batch mode with a DC power supply (DH1718G-4, Dahua, China) under a constant current mode. Prior to the EC treatment, the pH of the fluorochemical wastewater was adjusted to 3.5. During the EC treatment, the solution pH value was not adjusted or controlled. During the experimental procedure, once a specific time has been reached, the aeration and electrolysis will be terminated. The solution is left to stand for five minutes to ensure that the residual PFAS is evenly distributed.

Prior to use, AC (50 - 60 mesh) and PFA694E (50 - 60 mesh) were repeatedly washed with DI water, followed by drying at 150 °C for 4 h and stored in a dryer. Adsorption isotherm tests were conducted by AC and PFA694E with PFOA solution of a wide range from 10 to 400 mg L$^{-1}$ (AC) or 100 to 600 mg L$^{-1}$ (PFA694E) for 24 h under 25 °C. All isotherm data were fitted by the Langmuir model. The adsorption treatment of fluorochemical wastewater experiments was also conducted. The fluorochemical wastewater pH was not adjusted. All adsorption experiments were performed in the polypropylene centrifuge tubes (150 mL) at room temperature on a vortex plate at 500 rpm. The adsorbent dose was 330 mg L$^{-1}$.

Rapid small-scale column test (RSSCT) experiments (Supplementary Fig. 1b) were designed based on the constant diffusivity mode. Columns were organic glass made (6.4 mm inner diameter and 50 mm height) with a maximum volume of 1.66 mL. The PFA694E resins were crushed with a ball mill (Nanbei Instrument Limited, China) and sieved to 0.282 ~ 0.25 mm, which allowed the column diameter/PFA694E particle diameter ratio to be >8 ~ 10 to eliminate wall or channel effects. The volume of the ball-milling jar was 50 mL, and the diameter of the balls was 20 mm. The ball milling process was performed at a speed of 50 Hz with a 30 s pause after 30 seconds of milling, for a total of five cycles. Filters (200-mesh) on both sides of the column are to distribute water flow and prevent the leading of the PFA694E particles during the column testing. The columns were operated in an up-flow configuration mode of a rate of 2 mL min$^{-1}$ using a peristaltic pump. The column was thoroughly cleaned using DI water for 24 h before the experiment. The specific design formula was expressed as Eq. 2 and parameters can be found in Supplementary Table 9:

$$\frac{EBCT_{SC}}{EBCT_{LC}} = \left(\frac{d_{SC}}{d_{LC}}\right)^2 \qquad (2)$$

where EBCT refers to the empty bed contact time (min); SC and LC refer to the RSSCT column and full-scale adsorber, respectively; $d$ represents the diameter of the adsorbent (mm).

In all cases, triplicate experiments were conducted, and samples were collected and filtered by a 0.22 μm cellulose membrane (>95% PFAS recovery). In addition, solid phase loading (μmol PFAS per adsorbent) of PFAS was calculated as follows:

$$Solid\ phase\ loading = \frac{\int (C_0 - C_{effluent})dV}{m} \qquad (3)$$

where $C_0$ and $C_{effluent}$ refer to the PFAS concentrations in the influent and effluent, respectively; $V$ is the wastewater treated; $m$ is the mass of adsorbent. The dissolved zinc dosage (mg L$^{-1}$) was calculated as a function of electrocoagulation time using the following equation:

$$Zinc\ dosage = \frac{1000}{V} \times \frac{I \times t_{EC}}{nF} \times M \times \eta \qquad (4)$$

where $I$ and $t_{EC}$ refer to the applied current and time during electrocoagulation, respectively; $F$ is the Faraday's constant; $n$ is the number of electrons in Zn - $2e \rightarrow Zn^{2+}$; $M$ (65 g mol$^{-1}$) refers to the relative molar mass of zinc; $\eta$ refers the current efficiency of Zn - $2e \rightarrow Zn^{2+}$, which is determined to 0.91 in the fluorochemical wastewater used in this study. The adsorbed amount ($q_t$, mmol g$^{-1}$) of PFAS on zinc hydroxide flocs was calculated using the following equation:

$$q_t = 1000 \times \frac{C_0 - C_t}{Zinc\ dosage} \qquad (5)$$

where $C_0$ and $C_t$ are the PFAS concentration in solution at initial and reaction time, respectively.

## Flocs characterization

The zinc hydroxide flocs were collected and freeze-dried. The Brunauer-Emmett-Teller (BET) surface areas of the dried flocs were measured using a Micromeritics ASAP 2460. Field-emission scanning electron microscopy (FESEM, ZEISS Sigma 300) coupled with energy dispersive X-ray spectroscopy (EDX) was used for the morphology and elemental analyses. X-ray photoelectron spectroscopy (XPS) spectra of the flocs were measured by an ESCALAB 250Xi XPS system with a monochromatic Al Kα source. The zeta potentials of zinc hydroxide flocs during EC were measured by a Malvern zeta potential analyzer (Zetasizer Nano ZS90). A laser particle-size analyzer (Mastersizer 3000, Malvern) was used to determine the size distribution of the zinc hydroxide flocs. A drop shape analyzer (DSA 100, KRüSS) was employed for the characterization of the contact angle of the flocs.

## PFAS and Wastewater analysis

Concentrations of 107 PFAS were analyzed using an LC-Q-Orbitrap-HRMS system comprised of a Dionex ultraperformance liquid chromatograph (UPLC) and a Q-Extractive Plus mass spectrometer equipped with a heated-electrospray ionization (HESI) source (Thermo-Fisher Scientific, USA). The UPLC separation was carried out by an Acquity UPLC® BEH C18 column (2.1 100 mm, 1.7, Waters) using a gradient composition of solvent A (ACN) and solvent B (2 mM NH$_4$AC in DI water) at a flow rate of 0.25 mL min$^{-1}$. The gradient expressed as the concentration of solvent A was as follows: 0 – 0.2 min, hold at 20% A; 0.2 – 8 min, a liner increase from 20% A to 80% A; 8 – 10 min, a liner increase from 80% A to 95% A; 10 – 12 min, hold at 95% A; 12 – 12.1 min, a liner decrease from 95% A to 20% A; 12.1 – 15 min, and hold at 20% A. The sample volume injected was 5 μL. The HESI source was operated in negative ionization mode with the spray potential at 3.2 kV, the capillary temperature at 320 °C, the aux gas heater temperature at 350 °C, and the sheath and auxiliary gas flow rate of 45 arb and 10 arb, respectively. Each sample was spiked with EISs solution (5 μg L$^{-1}$ of each $^{13}$C-labeled PFAS, Supplementary Table 11) as the internal standard for analysis. The mass transitions and spectrometry conditions of the monitored PFAS for multiple reaction monitoring (MRM) are specified in Supplementary Table 1 and details can be also found in our previous study[33].

Concentrations of total organic fluorine (TOF) in wastewater, and sulfur and chlorine content in PFA694E were detected using a combustion-ion chromatography (CIC) equipped with an automatic quick furnace (AQF-2100H), a combustion monitor (CM-210), and an ion chromatography system (Dionex ICS-5000). The Dionex ICS-5000 was also employed to measure the anions in wastewater including $SO_4^{2-}$, $Cl^-$, $F^-$ and $NO_3^-$. The mobile phase was 30 mM NaOH, and the flow rate was set at 1.2 mL min$^{-1}$. The ASRS300 suppressor current was set at 90 mA. Concentrations of total organic carbon (TOC) and total inorganic carbon (TIC) in wastewater, and carbon and nitrogen content in PFA694E were measured by a multi N/C UV TOC analyzer (Analytic Jena, Germany) using a catalytic combustion method at 800 °C. Inductively coupled plasma mass spectrometry (ICP-MS, Thermo Fisher iCAP RQ) was employed to measure the zinc concentration. Samples were collected and acidified by HNO$_3$ and then analyzed. The total anion exchange capacities of crushed and as-received PFA694E resins were measured using the GB/T 11992-2008 National Standard of China in polypropylene columns.

## Techno-economic analysis (TEA) and life-cycle environment impact assessment (LCEIA)

The electric energy ($E$, kWh m$^{-3}$) of the Zn-based EC was calculated by Eq. 6:

$$E = \frac{U \times I}{V} \times t_{EC} \qquad (6)$$

where $U$ refers to the average voltage during electrocoagulation. The operating costs include the direct cost of electricity, adsorbents, and spent adsorbents incineration treatment. For detailed calculations of operating costs for the Zn-based EC, PFA694E bed, and treatment-train process, see Supplementary Text 1. A cradle-to-grave life-cycle assessment was employed to compute the carbon footprint of both the Zn-based EC and the PFA694E adsorption bed processes. In this study, we only consider the carbon footprints (KgCO$_2$ eq.). For detailed calculations, see Supplementary Text 2.

## Data availability

The data supporting the findings of this work are available within the articles and its Supplementary Information files. Source Data file has been deposited in Figshare under accession code DOI link https://doi.org/10.6084/m9.figshare.28468565[55]. Source data for all graphs are provided in this paper.

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

## Acknowledgements

This study was financially supported by the National Natural Science Foundation of China, grant No. 51878170 (H.L.), the Guangdong Basic and Applied Basic Research Foundation, grant No. 2023A1515140067 (H.L.), and the National Natural Science Foundation of China, grant No. U23A2056 (C.T.).

## Author contributions

Y.L. and L.Y. performed the experiments, analyzed the data, and drafted the manuscript. C.T. contributed to the instrumental analysis. Y.Y. and A.W. contributed to data interpretation. C.T., S.L., J.X., and Q.H. contributed to the revision of the manuscript. H.L. designed and supervised the research, provided resources, and revised the manuscript. All authors participated in the manuscript reviewing and editing.

## Competing interests

The authors declare no competing interests.
