## [Transparent Peer Review file · Nature Communications]

Broad-spectrum capture of hundreds of per- and polyfluoroalkyl substances from fluorochemical wastewater

Corresponding Author: Professor Hui Lin

Version 0:

Reviewer comments:

Reviewer #1

(Remarks to the Author)

This study examined a two-stage PFAS treatment process train to treat a wide range of PFAS. While PFAS control for discharging water is an imminent issue, this work is based on very known processes (electrocoagulation, activated carbon, and anion exchange resin) that were already well studied. It is not surprising to see greater PFAS removal efficiencies with the two-stage process than those with each single unit process, and the combined process does not show any new direction or significant scientific merits. The biggest concern is the results from electrocoagulation with a Zn sacrificing anode (Zn-based EC). Most of the Zn-based EC results were already reported in previous EC studies, but none of the previous Zn-based EC studies were mentioned or cited in the manuscript, no rationale or background was stated as to why Zn-based EC was chosen for this work. Various PFAS were analyzed but their treatment based on their characteristics by the proposed process was not well addressed. Due to the significant overlaps with previous studies and lack of novelty, this work is not suitable for publication. Please see specific comments below.

Major comments:

1. No rationale was provided why Zn-based EC was chosen for this study. What were the hypotheses or reasons for employing an electrocoagulation process among various PFAS treatment processes? There are previous studies that examined Zn-based EC for PFAS treatment, and none of them were mentioned or cited in the manuscript.
2. Regarding Zn-based EC related results and conclusions, most of the results here are not new, previous EC studies already reported very similar results. The first Zn-based EC study was reported in 2015 (<https://doi.org/10.1021/acs.est.5b02092>). In that study, better removal efficiencies were reported in EC with a Zn electrode with longer PFAS due to the hydrophobic interactions between a hydrophobic tail, tested PFAS with different C-F lengths, and reported better long-chain PFAS removal with a Zn electrode versus Al and Mg electrodes. That previous study systemically analyzed the Zn hydroxide floc to identify the main removal mechanism in the Zn-based EC system for long-chain PFAS removal. The produced Zn hydroxide flocs were characterized and their adsorption capacities were also examined in that study. Most of the results in this work regarding Zn-based EC were overlapped with the results in previous studies and they were not mentioned at all.
3. So, both Zn-based EC and anion exchange resin processes are well-known strategies to control either short-chain or long-chain PFAS. Then what are the new perspectives on implementing them together in one system? The rationale for using the two-stage process is very weak and no scientific merits were found. This work is close to the optimization of two existing processes rather than exploring new significant findings or suggesting new directions for this research field.
4. Descriptions of methods should be elaborated. For example, there is limited information on Zn-based EC reactor design and how Zn-based EC reactors were operated (semi-batch? Continuous flow? With aeration? Applied voltages? Etc.). In previous Zn-based EC studies, they employed aeration in the EC system to save electrical energy consumption (<https://doi.org/10.1021/acs.est.5b02092>) and one recent study also reported that aeration can be very critical for PFAS removal.
5. To assess the environmental impact of the proposed process, potential Zn ion concentration in the effluent should be analyzed. Since Zn (heavy metal) is released from the Zn-based EC reactor, it could potentially cause significant

environmental issues in the environment.

6. L26-27, please clarify how these adsorption capacities were assessed. Were they measured in a separate reactor with Zn hydroxide flocs only, or in the Zn-based EC reactor with aeration?

7. It was mentioned that 107 PFAS were detected in the wastewater sample but their concentrations and impacts on the removal process are not clear. PFOA was found in the highest concentration among those PFAS, and treatment results were mostly focused on PFOA control and potential mechanism, not other species with different physicochemical properties. Also, the impacts of the wastewater complexity (due to other components) were not well addressed in the manuscript. The impacts of other components are often very significant to PFAS treatment in the Zn-based EC as published in the previous study (<https://doi.org/10.1016/j.scitotenv.2016.03.114>).

8. L306, for the full scale simulation, only anion exchange resin was examined, nothing was conducted for Zn-based EC for the full scale simulation. This study examined those processes in one single process train but only one part of the process was assessed for the pilot-scale operation. It seems not appropriate.

Minor comments:

9. L63, the cited reference was published in 2007, now we have many more studies regarding treating PFAS in complex water, it is not limited.

10. L76, no rationale was provided to justify the use of Zn-based EC for this study. How did you know the hydrophobicity of Zn hydroxide flocs?

11. L135, what does mean the bubble diameter here? Were there any aeration bubbles during EC?

12. L178-L179, this is not interesting since the same result was already reported in the previous study published in 2015 (<https://doi.org/10.1021/acs.est.5b02092>).

13. L191, the kinetic study depending on PFAS chain lengths also reported by the previous study (<https://doi.org/10.1021/acs.est.5b02092>).

14. L305, however, PFAS removal by EC can be significantly hindered by other cations or anions in the water. See (<https://doi.org/10.1016/j.scitotenv.2016.03.114>).

Reviewer #2

(Remarks to the Author)

In this manuscript, the authors report a bench-scale application of zinc-based electrocoagulation and anion exchange resin to remove PFAS from concentrated industrial wastewater. The combination approach was found to be much more efficient at removing hydrophobic PFAS than anion exchange treatment alone. This type of work is timely and important; needs for efficient treatment methods for PFAS-laden industrial discharges are growing as PFAS become increasingly regulated. The data and methodology supporting this finding appear sound. However, the quality of the manuscript could be significantly improved if some minor technical and presentation issues were addressed.

The focus of this study is clearly on PFAS-rich industrial effluents, but readers might be interested in applying the treatment train in other contexts. Consider adding some discussion of the feasibility of this treatment train to applications such as site remediation or landfill leachate treatment.

The Zn-based EC process appears in the introduction at L76 but isn't given a proper introduction. Where has this technique been used before? Why was it selected for this study over other metal-based EC processes? Consider including a citation for SI ref 14 and other relevant literature here. Adding references or reviews in the introduction will help readers more quickly understand the novelty and importance of this paper.

The constant diffusivity scheme for the RSSCT was selected without commentary. Some discussion as to whether this approach can be reasonably expected to produce correspondence for pilot or full scale-systems would be beneficial. Some recent work on the subject might be worth discussion or citation: Liu et al., 2024 <https://doi.org/10.1016/j.watres.2024.121661> and Cheng et al, 2024 <https://doi.org/10.1016/j.watres.2023.120956> seem relevant.

L27: The statement "resulting in the highest 26 adsorption capacities among all reported adsorbents" might need some clarification or qualification because, for instance, the adsorption capacity for PFBA is about zero.

L55: It is not clear from the references or from the text what is meant by "special helical structure" and how it is relevant to this work. Are any of the treated PFAS helical?

L58: delete "necessary." Some membrane techniques are used to concentration PFAS solutions for destructive techniques.

L123: The choice of the Langmuir isotherm model needs discussion. At least some discussion of traditional ion exchange

isotherms would be helpful here. The empirical application single-component Langmuir model here does make some sense because the wastewater matrix is so complex and more rigorous treatment of the IX isotherms (Wahman et al., 2023 <https://doi.org/10.1021/acsestwater.3c00396>; Smith et al., 2023 <https://doi.org/10.1021/acsestwater.2c00572>) probably aren't feasible, but there are still opportunities to learn more about the mechanisms by at least thinking about them. For instance, how does the estimated maximum adsorption capacity estimated in this work compare to the anion exchange capacity of the resin? What does that imply about the mechanism of PFAS removal from the wastewater in this case?

Line 170: What is that selective adsorption coefficient? It almost looks like a separation factor. But it's just all the PFAS? [What is in this fluorochemical wastewater? Are most of the ions PFAS?] Interesting that most PFAS on the IX resins have K_d of about 1. What does that mean? Would it make more sense to use an equilibrium separation factor with a reference ion (even if it's one of the PFAS?).

Line 193: What is the particle size of the Zn EC flocs? This might matter for discussions of process kinetics. On a related note, what does a full-scale Zn EC system look like? Would the floc particle size be similar to this work?

Line 206: What is the physical significance of q_{dyn} (Called q_{uni} in SI Table S3)? Is it an empirical parameter for comparing kinetics across processes? If that is the case, some remarks on the generalizability of this comparison are needed. Kinetics of adsorption on ion exchange resins and activated carbons are usually determined by diffusive processes and depend on particle size and adsorbent dose---among other things---while Zn-EC likely follows a different mechanism. Thus, q_{dyn} may depend differently on changes in conditions across techniques.

L346 – 349: Argument is hard to follow from the petal plots. Consider changing to more traditional bar plots to make these points. Was the adsorbed PFOA concentration “dramatically” different?

L399—401: It is not clear how the evaluations in 6c and Table S6* [actually S7] were determined. The categories and scores seem arbitrary. Is this part of some more formal decision-making framework or developed ad hoc for this manuscript?

L431: (And other locations) There is an off-by-one error in the numbering of supplementary tables.
Line 445 mentions isotherms fits using the Freundlich model, but Freundlich fits do not appear to have been done.

L451. How was the resin crushed? This could be important because it is not yet obvious if or how crushing method affects ion exchange RSSCT correspondence.

Ref 1 needs to also cite federal register for the PFAS drinking water rule. USEPA, (2024). PFAS National Primary Drinking Water Regulation Rulemaking Fed. Regist., 89(82): p. 32532-32757

Figure 2: Check that panel letter labels in the caption are correct.

Figures 2d, S3--6: What are the subplots in the two panels? Specifically, what are the 3 columns?

Figure 3d: Panels are hard to see/interpret --- especially the spectra.

Figure 5b: Legend on (b)? Markers and line styles are different from (a).

Figure 5 e,f,g: The nested petal plots and ring plots are difficult to read and interpret. Consider breaking out separate pie and bar charts to highlight important trends more clearly.

Figure S9: Rightmost panels are too condensed to be interpretable.

Figure S13 (and others): Text is too small and compressed to read easily, even zoomed in digitally.

SI Title: “realities” should be “realizes.”

Reviewer #3

(Remarks to the Author)

This manuscript makes a major contribution to advancing treatment for industrial wastewaters. The authors tested very high PFAS concentrations (mg/L), as opposed to lower ng/L levels that drinking water standards require, but the authors did excellent work on their system and proposed mechanisms to explain findings. They studied a much wider range of PFAS than most studies, and the graphics are phenomenal. The following are minor suggestions to improve clarity.

Line 23: “PFASs” does not need the second s because PFAS is plural as defined in line 20

Line 30: “iodinated PFAS, in which the fluorine atom is replaced by an iodine atom”... if the F is replaced is it still PFAS? This line is confusing. Please edit for clarity. It sounds like PFAS are becoming PIAS.

Line 36: “a few days ago, the Biden-Harris administration...” I recommend not being so specific on timing because it can make the paper look outdated quickly.

Introduction: Studying 107 PFAS is impressive. The authors could add a bit more information on the range of chemical properties such as Know, solubility, MW or other to help show how broad the PFAS were that they studied. For example, did they study short-chain and long chain? Cationic, anionic? Sulfonates and carboxylic acids? It would be worth mentioning

these key groups PFAS are classified into somewhere in the introduction around line 74.

Results

Line 87: I disagree that 107 is the same as "hundreds". Hundreds would imply at least two one-hundreds, and the authors did not measure 200 PFAS.

In general, the figures are spectacular. Very well done!

The mechanism discussion on line 218 is very interesting.

Line 240 uses "PFASs" and "PFAS", both should be PFAS

Line 305: Do authors have a reference to a toxicity study that shows iodinated PFAS are indeed less toxic or is this speculation?

Implications: do authors think this treatment strategy could work on more dilute streams such as municipal wastewater or drinking water? The opening line in the paper is about drinking water standards. Could this work for a drinking water utility, or is it more applicable to the higher mg/L levels tested?

Methods:

Somewhere in the first section can authors note the total concentration of PFAS in the industrial wastewater?

Reviewer #4

(Remarks to the Author)

This article presents an innovative treatment strategy that achieves efficient and broad-spectrum capture of 107 PFASs by combining Zn-based electrocoagulation (EC) with anion-exchange resin (AER) beds. The study demonstrates that the "zero-carbon" adsorbent, zinc hydroxide flocs generated by Zn-based EC, significantly enhances adsorption capacity through a semi-micellar adsorption mechanism similar to mineral flotation. The technical-economic analysis and life-cycle environmental impact assessment reveal that this method not only reduces costs by an order of magnitude but also decreases the carbon footprint by 70%. Additionally, the study observes significantly improved adsorption selectivity for iodinated PFAS, providing new insights into designing environmentally friendly fluorochemicals. Overall, this research offers a valuable solution to the PFAS challenge. However, the following issues need to be appropriately addressed.

1. The abstract mentions that the PFAS removed by Zn-based electrocoagulation achieved the highest adsorption capacities among all reported adsorbents. On what basis was this result generated? The authors need to clarify whether this result applies to all PFAS or specific PFAS.
2. In the first paragraph of the introduction, it is recommended to specify the hazards of PFAS.
3. In introduction part, the authors need to summarize the current PFASs removal method in FIP. How about the removal efficiencies of these 107 PFASs? What is the biggest challenge for the current used method?
4. Lines 53-55 suggest providing examples of chemical and biological processes.
5. Line 74: why do the authors choose these 107 PFASs, are cation and zwitter PFASs included?
6. In lines 74-76, the description of the Zn-based electrocoagulation process is brief. It is recommended to elaborate on its advantages compared to traditional methods.
7. The concept of Zn-based electrocoagulation is not proposed for the first time. Please summarize previous reports on this topic in the introduction.
8. Line 123: Why was the Langmuir model used to study adsorption capacity instead of other models?
9. Line 192: Please explain the reason for the selection of these 6 PFCAs? no other PFASs?
10. In line 276, it is stated: "Notably, the behavior of the novel Ix-PFAAs was markedly different." It is recommended to further explore the reasons behind this phenomenon. Consider using theoretical calculations (e.g., DFT) or other methods to elucidate the mechanism.
11. Lines 293-295 mention that iodine substitution significantly affects the chemical properties of PFAS, while chlorine substitution does not. It is recommended to expand this discussion and compare the environmental behavior (e.g., degradability) of iodinated and chlorinated PFAS in detail.
12. Is the software-predicted log Kow of PFAS reasonable?
13. Line 417: The degradation results of iodinated PFAS under natural conditions should be presented in detail.
14. Line 428: Is PFOS/PFOA linear, branched, or an isomer?
15. Line 431: Please double check the table and figure across the whole manuscript. In this sentence, Table S7 should be replaced by Table S8. There are other similar mistakes in the MS.
16. Line 443-444: Why is the isotherm only for PFOA? Besides, the concentrations were also quite high?"
17. Line 450: Is there any possibility that a certain amount of PFASs were adsorbed on the surface of the glass? Did you check it?
18. Line 455: What do you mean "of a rate a mL·min⁻¹"
19. What was the unadjusted pH range during the EC treatment process?
20. There are errors in the shapes of Figures 2b and 2c. Please check carefully.
21. Considering the manuscript, it is recommended that the authors expand on the practical application prospects and potential challenges of combining Zn-based electrochemical and adsorption processes.

Reviewer #5

(Remarks to the Author)

This manuscript describes a comprehensive study of PFAS treatment comparing established and novel approaches. The

study advances the current options available for PFAS treatment for industrial waste streams. The authors are encouraged to consider the following comments in a revised version of the manuscript.

1. For the results in Figure 2, the authors should provide more discussion on the performance of AC and AER given that adsorption is considered the best available treatment for PFAS. For instance, was the relatively low removal of PFAS by AC and AER due to high concentration of PFAS, diverse PFAS structures, or high concentration of TOC or inorganic chemicals?

2. Given that Zn EC effectively removed diverse PFAS, TOC, F⁻, NO₃⁻, and SO₄²⁻, additional discussion on the mechanisms of Zn EC are needed to generalize the results to other waste streams and operating conditions. For example, the authors developed relationship between PFAS K_{ow} and K_d for Zn EC removal, but what were the molecular interactions and what accounts for the removal of TOC and inorganic chemicals? What is unique about Zn hydroxide flocs relative to Al or Fe hydroxide flocs? Also, what is unique about the adsorption of PFAS to Zn hydroxide floc relative to AC and AER? PFAS removal by AC is due to hydrophobic + van der Waals (vdW) interactions, and AER is combination of electrostatic and hydrophobic + vdW interactions, so the authors should be able to say something more specific about the interactions between PFAS/Zn hydroxide floc relative to AC and AER.

3. Following from comment 2, the authors describe unique F/Zn interaction, but this description is still general. What is unique about F/Zn interaction relative to other metals.

4. Following from comment 3, what is the expected nature of the PFAS/Zn hydroxide floc? Will substantial de-watering be required before the floc can be treated or disposed of?

5. The results for I-PFAA were interesting. Do the authors have additional insights on the production or use of I-PFAAs? Is the higher adsorption of I-PFAAs simply explained by more hydrophobic chemical or are other size or shape factors part of the explanation?

6. The coupled system of Zn EC followed by AER makes sense, especially for high-strength industrial PFAS wastewater. Coagulation followed by AER is usually more effective than either process alone for TOC removal from natural water or leachate.

7. For economic and environmental impact, the authors should qualify/support the 10% water content for the PFAS/Zn floc. Given the proof of concept nature of the research, the authors should also highlight other assumptions that are likely to have a strong impact on the economic and environmental results. These assumptions (sensitive inputs) provide direction for future research.

8. Do the authors expect the results for PFAS removal by Zn EC to be applicable to lower PFAS concentrations encountered in surface water or groundwater? Additional insights on factors that support or inhibit PFAS removal by Zn EC would benefit the PFAS research community in prioritizing future research.

Trevor Boyer
Professor, Arizona State University

Version 1:

Reviewer comments:

Reviewer #2

(Remarks to the Author)

The manuscript has been significantly improved by the revisions the authors made in response to reviewer comments. However, there are still some minor technical revisions which would be useful to make before publication.

The justification for selecting the constant diffusivity approach for the RSSCT (revised lines 381-383) is still not clear in the text. I suggest deleting the text "The breakthrough of PFAS in RSSCT is dependent³⁸² on the particle size of the adsorbent, and hence the constant diffusion model is appropriate for designing RSSCT experiments" and replacing it with something like "A constant diffusivity design was chosen for the RSSCTs. A recent study has shown that, with proper interpretation, this approach can lead to useful approximations of pilot- or full-scale system performance.[50]"

L563: Please state what kind of grinder was used to crush the resin. List any relevant operational settings.

Reviewer #3

(Remarks to the Author)

The authors have satisfactorily addressed my remaining comments to make an already strong manuscript even stronger. Very well done.

Reviewer #4

(Remarks to the Author)

The authors have made a commendable effort to address and revise the majority of the comments. However, there remain two minor points of clarification:

1. In response to Comment 1, the authors have not yet clearly specified whether the results pertain to all PFAS or are limited to certain specific compounds. While hydrophobic PFAS are mentioned in the authors' reply, the definition of hydrophobic versus hydrophilic PFAS is still unclear. Are all PFAS with a log Kow > 4 considered hydrophobic?

2. Regarding Comment 20, what do the red squares in Figure 2c represent? Is it possible that the red squares should be replaced with red triangles?

Reviewer #5

(Remarks to the Author)

The authors have addressed the reviewers' comments and improved the submission. It is suitable for publication.

Detailed Response to the Reviewers' Comments

Nov. 28, 2024

Journal: Nature Commun. (Manuscript ID: NCOMMS-24-25626)

Title: "Treatment-train strategy realizes broad-spectrum capture of hundreds of per- and polyfluoroalkyl substances from fluorochemical wastewater"

Yiyang Liang^{†#}, Lihui Yang^{†#}, Caiming Tang^{†#}, Ying Yang[†], Shangtao Liang[§], Anqi Wang[†], Jiale Xu[†], Qingguo Huang[§], Hui Lin^{†*}

We sincerely thank all reviewers for their valuable comments and suggestions, which are certainly helpful in improving the quality of our work. We have carefully and systematically responded to all the points raised. We have also highlighted the revised text in **red** in the main text. Provided below are our detailed responses to each point.

Reviewer #1

This study examined a two-stage PFAS treatment process train to treat a wide range of PFAS. While PFAS control for discharging water is an imminent issue, this work is based on very known processes (electrocoagulation, activated carbon, and anion exchange resin) that were already well studied. It is not surprising to see greater PFAS removal efficiencies with the two-stage process than those with each single unit process, and the combined process does not show any new direction or significant scientific merits. The biggest concern is the results from electrocoagulation with a Zn sacrificing anode (Zn-based EC). Most of the Zn-based EC results were already reported in previous EC studies, but none of the previous Zn-based EC studies were mentioned or cited in the manuscript, no rationale or background was stated as to why Zn-based EC was chosen for this work. Various PFAS were analyzed but their treatment based on their characteristics by the proposed process was not well addressed. Due to the significant overlaps with previous studies and lack of novelty, this work is not suitable for publication. Please see specific comments below.

Response: We thank the reviewer for commenting and giving constructive suggestions on our manuscript, and we thus have the opportunity to clarify and improve the manuscript. We have carefully revised the manuscript according to your comments, and we would like to clarify the novelty of this research work.

First, we would like to thank you for your careful reading of the manuscript. The revised manuscript now includes relevant content related to the progress of Zn-based EC (see Comment 1 below).

As you indicated, it was us, Dr. Lin and Dr. Huang, who first proposed the utilization of Zn-based EC for the removal of PFAS in 2015 (<https://doi.org/10.1021/acs.est.5b02092>), and dozens of research groups followed us with further studies. Nevertheless, none of these studies explored the performance on complex actual PFAS wastewater. We agreed with the reviewer that there is an urgent need to control the high concentrations of PFAS present in industrial wastewaters. This manuscript addresses the challenge of treating a significant PFAS source, i.e., fluorochemical wastewater. This study represents therefore a significant advance in the promotion of Zn-based EC for practical applications.

Fluorochemical wastewater contains hundreds of PFAS compounds with diverse structures. However, there are very limited studies examining the removal of PFAS from fluorochemical wastewater for all PFASs. To date, only one recent study has reported on the removal of PFAS in full-scale fluorochemical wastewater treatment processes. Zhang et al. (2024) found that the current wastewater treatment processes were ineffective in removing PFAS discharged from fluorochemical manufacturers. The study revealed that a large quantity of 48 PFAS (ranging from 14.7 to 5200 $\mu\text{g}\cdot\text{L}^{-1}$) in the effluents of 10 fluorochemical industrial parks (FIP) in China was discharged to surface waters without complete treatment. Specifically, the mass flows of PFAS increased by 233% after the activated sludge system but decreased by only 0–13% after activated carbon (AC) filtration. Despite the success of traditional adsorbents such as AC/anion exchange resin (AER) in removing PFAS from relatively clean waters with minimal low background matrix such as groundwater and surface water, it is unclear whether they can effectively remove hundreds of PFAS from the complex fluorochemical wastewater.

Our study indicates that SO_4^{2-} and NO_3^- in the fluorochemical wastewater participate in the Zn^{2+} hydrolysis reaction, leading to their removal as zinc hydroxide salts, i.e., zinc hydroxide

nitrate ($Zn_5(NO_3)_2(OH)_8$, ZnHN) and zinc hydroxide sulfate ($Zn_4SO_4(OH)_6$, ZnHS). This is a significant finding given that few technologies have been reported to be capable of adsorbing large quantities of PFAS as well as simultaneously removing competing constituents from complex waste streams. This would greatly benefit the subsequent conventional adsorption processes. We therefore proposed a treatment-train process combining Zn-based EC with the existing AC/AER adsorption devices to achieve broad-spectrum removal of PFAS from the complex fluorochemical wastewater. It is promising that the treatment-train process achieves remarkable removal of all 107 PFAS, much better than the AC/AER alone.

Effective technologies for removing PFAS from complex industrial wastewater are currently urgently needed. A treatment train based on currently available processes is the best option for future applications. Both EC and AC/AER adsorption processes have been successfully applied on a large scale. We are confident that the coupling of Zn-based EC and AC/AER adsorption processes in one treatment train is an innovative technology, as it is ready for scale-up with a high level of technological maturity.

This study is novel because it: (1) examines a much wider range of PFAS than most studies; (2) advances the currently available technologies for PFAS treatment for industrial waste streams; (3) proposes a concept of "zero-carbon" adsorbent for PFAS removal; and (4) provides insights into the design of environmentally-friendly fluorochemicals.

Zhang et al. Emerging and legacy per- and polyfluoroalkyl substances (PFAS) in fluorochemical wastewater along full-scale treatment processes: source, fate, and ecological risk. J. Hazard. Mater. 2024, 465, 133270.

Major comments:

Comment 1. No rationale was provided why Zn-based EC was chosen for this study. What were the hypotheses or reasons for employing an electrocoagulation process among various PFAS treatment processes? There are previous studies that examined Zn-based EC for PFAS treatment, and none of them were mentioned or cited in the manuscript.

Response: We agree with the reviewer that references should be added, and the relevant contents have been added in the revised manuscript (**Pages 4-5, Lines 80-99**).

Added: One of the defining characteristics of PFAS is its exceptional surface activity, which allows it to be adsorbed in multiple layers through semi-micellar or micellar adsorption. This significantly boosts its adsorption capacity. An example of this is mineral flotation, where trapping agents are adsorbed onto hydrophobic mineral particles through semi-micellar adsorption, enabling selective mineral capture. Inspired by this, we propose a reverse mineral flotation process using hydrophobic "mineral particles" as adsorbents to selectively extract PFAS from water via semi-micellar adsorption. We found that zinc hydroxide flocs, formed in situ by electrolysis, exhibit properties of hydrophobic "mineral particles," thus enabling them to rapidly adsorb hydrophobic long-chain PFAS from water with extremely high adsorption capacities^{34, 35}. For example, these zinc hydroxide flocs achieved an equilibrium adsorbed amount (q_e) of up to 5.74/7.69 mmol g⁻¹ (Zn) for PFOA/PFOS within minutes at an initial concentration of 0.5 mM³⁴. We therefore hypothesize that Zn-based electrocoagulation (EC) can effectively capture long-chain PFAS in the fluorochemical wastewater, while the existing AC and/or AER adsorption devices can then remove the

remaining low concentrations of short-chain PFAS. This approach is founded on two primary observations: 1) Despite the presence of hundreds of PFAS types in fluorochemical wastewater, long-chain PFAS, particularly PFOA, dominate the total PFAS concentration Zn-based EC also removes dissolved organic matter (DOM) and traps colloidal particles from wastewater, significantly mitigating the impact of these competing constituents on subsequent adsorption processes. This strategy represents the first attempt to achieve broad-spectrum removal of hundreds of PFAS from real fluorochemical wastewaters, potentially offering new options for tackling PFAS in complex industrial waste streams.

Comment 2. Regarding Zn-based EC related results and conclusions, most of the results here are not new, previous EC studies already reported very similar results. The first Zn-based EC study was reported in 2015 (<https://doi.org/10.1021/acs.est.5b02092>). In that study, better removal efficiencies were reported in EC with a Zn electrode with longer PFAS due to the hydrophobic interactions between a hydrophobic tail, tested PFAS with different C-F lengths, and reported better long-chain PFAS removal with a Zn electrode versus Al and Mg electrodes. That previous study systemically analyzed the Zn hydroxide floc to identify the main removal mechanism in the Zn-based EC system for long-chain PFAS removal. The produced Zn hydroxide flocs were characterized and their adsorption capacities were also examined in that study. Most of the results in this work regarding Zn-based EC were overlapped with the results in previous studies and they were not mentioned at all.

Response: We appreciate your valuable comments. As you stated, it was us, Dr. Lin and Dr. Huang, who firstly proposed the utilization of Zn-based EC for the removal of PFAS in 2015 (<https://doi.org/10.1021/acs.est.5b02092>), and dozens of research groups followed us with further studies. Nevertheless, none of these studies explored complex industrial wastewater with PFAS. It is an urgent need to control the high concentrations of PFAS present in industrial wastewaters. This manuscript addresses the most significant source of PFAS to the environment, i.e., fluorochemical wastewater. This study represents therefore a significant advance in the promotion of Zn-based EC for practical applications.

We concur with the proposition that the mechanism of PFAS removal by Zn-based EC, as proposed in this study, is based on our previous study. Particularly, hydrophobic interaction is the dominant force governing the adsorptive removal of PFAS. However, it should be highlighted that this study was not merely an investigation of the adsorption kinetics and adsorption capacities of typical PFAS in simulated solutions. More importantly, the study established a correlation between PFAS structure and adsorbability, based on the calculated selective adsorption coefficient (K_d) of 107 PFAS in the fluorochemical wastewater. For the first time, it was observed that substituting iodine for fluorine significantly alters the properties of PFAS that favors their adsorptive removal and degradation, offering valuable insights into the design of environmentally-friendly fluorochemicals.

Considering your comments, we have added the relevant contents related to the progress of the Zn-based EC in the revised manuscript (See Comment 1).

Comment 3. So, both Zn-based EC and anion exchange resin processes are well-known strategies to control either short-chain or long-chain PFAS. Then what are the new perspectives on implementing them together in one system? The rationale for using the two-stage process is very weak and no scientific merits were found. This work is close to the optimization of two existing processes rather than exploring new significant findings or suggesting new directions for this research field.

Response: Your comments are appreciated. In **Lines 145-157** of the original manuscript, we set forth the rationale behind the proposed coupling process of Zn-based EC with AC/AER. The absence of a robust specific affinity between AER and PFAS renders it highly susceptible to interference from coexisting competing constituents, thereby markedly reducing its efficacy in the removal of PFAS in practical complex wastewater matrices. This study demonstrated that SO_4^{2-} and NO_3^- in the fluorochemical wastewater participate in the Zn^{2+} hydrolysis reaction, leading to their removal as zinc hydroxide salts (see figure below), i.e. zinc hydroxide nitrate ($\text{Zn}_5(\text{NO}_3)_2(\text{OH})_8$, ZnHN) and zinc hydroxide sulfate ($\text{Zn}_4\text{SO}_4(\text{OH})_6$, ZnHS). This is a notable finding, as few technologies have been reported to be capable of adsorbing large quantities of PFAS as well as simultaneously removing competing constituents from complex waste streams. This would greatly benefit subsequent conventional adsorption processes. In light of this, we proposed a treatment-train process that combines Zn-based EC with the existing AC/AER adsorption devices to achieve broad-spectrum removal of PFAS from complex fluorochemical wastewater. The results of the RSSCT experiments provided support to this hypothesis.

To respond this comment, we have added the relevant content to explain the proposed treatment-train strategy in the **Introduction** section (**Page 5, Lines 90-99**) and the results of XRD characterization (**Page 9, Lines 189-194**) in the revised manuscript.

Added: We therefore hypothesize that Zn-based electrocoagulation (EC) can effectively capture long-chain PFAS in the fluorochemical wastewater, while the existing AC and/or AER adsorption devices can then remove the remaining low concentrations of short-chain PFAS. This approach is founded on two primary observations: 1) Despite the presence of hundreds of PFAS types in fluorochemical wastewater, long-chain PFAS, particularly PFOA, dominate the total PFAS concentration Zn-based EC also removes dissolved organic matter (DOM) and traps colloidal particles from wastewater, significantly mitigating the impact of these competing constituents on subsequent adsorption processes. This strategy represents the first attempt to achieve broad-spectrum removal of hundreds of PFAS from real fluorochemical wastewaters, potentially offering new options for tackling PFAS in complex industrial waste streams.

Added: Our observations showed that the Zn-based EC process also significantly reduced TOC as well as F^- , SO_4^{2-} and NO_3^- (almost complete removal) in the fluorochemical wastewater (Supplementary Fig. 7). Results from XRD analysis (Supplementary Fig. 8) showed that SO_4^{2-} and NO_3^- participate in the Zn^{2+} hydrolysis reaction, leading to their removal as zinc hydroxide salts, i.e. zinc hydroxide nitrate ($Zn_5(NO_3)_2(OH)_8 \cdot 2H_2O$, ZnHN) and zinc hydroxide sulfate ($Zn_4SO_4(OH)_6 \cdot 2H_2O$, ZnHS).

Supplementary Fig. 8 XRD characterization of the formed zinc hydroxide flocs in the fluorochemical wastewater.

Comment 4. Descriptions of methods should be elaborated. For example, there is limited information on Zn-based EC reactor design and how Zn-based EC reactors were operated (semi-batch? Continuous flow? With aeration? Applied voltages? Etc.). In previous Zn-based EC studies, they employed aeration in the EC system to save electrical energy consumption (<https://doi.org/10.1021/acs.est.5b02092>) and one recent study also reported that aeration can be very critical for PFAS removal.

Response: Thank you for the valuable comment. The solution was stirred and purged with pure N_2 (1 L(N_2)/L(solution)/min). In the original manuscript of **line 435-441**, we have written “*Electrocoagulation, Batch Adsorption and RSSCT Experiments*. The electrocoagulation (EC) reactor was composed of a cylindrical EC cell (8 cm diameter and 15 cm height) with a 500 mL volume, as shown in Supplementary Fig. 1a. A metal (Zn, Al or Fe) sheet of 100 cm² surface area was used as anode, while a 304 stainless steel rod of 0.3 cm diameter was used as the cathode, with a distance of 3 cm between the electrodes. In each run, 300 mL of wastewater or simulated solution was added, and powered by a DC power (DH1718G-4, Dahua, China) under a constant current mode. During the EC treatment, the solution pH value was not adjusted or controlled.”

The paper you referenced should be <https://doi.org/10.1016/j.chemosphere.2021.130956>. We acknowledged that aeration may be helpful for the removal of low concentrations of PFAS, as it permits a greater accumulation of PFAS in the upper solution, thereby facilitating their

adsorption by the zinc flocs. To test this hypothesis, comparative experiments were conducted, which demonstrated that aeration did not significantly affect the removal of PFOA at concentration of mg/L level, but significantly facilitated adsorptive removal of PFOA at $\mu\text{g/L}$ levels (see figure below).

Figure. The effect of stirring on the removal of PFOA (10 mg/L) by Zn-based EC.

Wu T., Park, M, Kim, K. Energy-efficient removal of PFOA and PFOS in water using electrocoagulation with an air cathode. *Chemosphere*, **2021**, 281, 130956.

Considering the comment, we have modified the experiment section in the revised manuscript (**Page 24, Lines 544-554**).

Modified: Electrocoagulation, Batch Adsorption and RSSCT Experiments. The electrocoagulation (EC) reactor was composed of a cylindrical EC cell (8 cm diameter and 15 cm height, Supplementary Fig. 1a) with a 500 mL volume, and was equipped with a gas stirring device (0.3 L N₂ per min), as shown in Supplementary Fig. 1a. A metal (Zn, Al or Fe) sheet of 100 cm² surface area was used as anode, while a 304 stainless steel rod of 0.3 cm diameter was used as the cathode, with a distance of 3 cm between the electrodes. In each experimental run, 300 mL of wastewater or simulated solution was added and operated in a batch mode with a DC power supply (DH1718G-4, Dahua, China) under a constant current mode. During the EC treatment, the solution pH value was not adjusted or controlled. During the experimental procedure, once a specific time has been reached, the aeration and electrolysis will be terminated. The solution is left to stand for five minutes to ensure that the residual PFAS is evenly distributed.

Comment 5. To assess the environmental impact of the proposed process, potential Zn ion concentration in the effluent should be analyzed. Since Zn (heavy metal) is released from the Zn-based EC reactor, it could potentially cause significant environmental issues in the environment.

Response: We have measured the concentration of the residue zinc ion in the effluent of the Zn-based EC and RSSCT column using an ICP-MS (Thermo Fisher iCAP RQ). The residue zinc ion concentration in the fluorochemical wastewater was $0.84 \text{ mg}\cdot\text{L}^{-1}$ after Zn-based electrocoagulation with a solution pH value of 8.6. The concentration of the effluent from the RSSCT column was less than $0.5 \text{ mg}\cdot\text{L}^{-1}$. The relevant content and discussion have been added in the revised manuscript (Page 19, Lines 429-434; Page 22, Lines 506-509).

Added: Furthermore, the concentrations of residual zinc in the effluent of the Zn-based EC and RSSCT column were also determined. The residue zinc concentration in the treated fluorochemical wastewater was $0.84 \text{ mg}\cdot\text{L}^{-1}$ after Zn-based EC and less than $0.5 \text{ mg}\cdot\text{L}^{-1}$ after the RSSCT column (Supplementary Fig. 21). Given that zinc is an essential semi-trace element and the drinking water limits set by the World Health Organization (WHO) and European standards are $5 \text{ mg}\cdot\text{L}^{-1}$ and $10 \text{ mg}\cdot\text{L}^{-1}$ respectively, using Zn-based EC for water treatment is considered safe

Added: It is important to note that zinc is a toxic heavy metal element. The safe disposal and utilization of zinc hydroxide floc sludge are vital considerations for the potential application of the Zn-based EC process. Pyrolysis, including incineration and low-temperature alkaline hydrothermal processes, may be applicable to recover Zn.

Supplementary Fig. 21 the concentrations of residue residual zinc in the effluent of the Zn-based EC and RSSCT column

Comment 6. L26-27, please clarify how these adsorption capacities were assessed. Were they measured in a separate reactor with Zn hydroxide flocs only, or in the Zn-based EC reactor with aeration?

Response: Thanks for your comment. The adsorption capacities of PFAS were measured in the Zn-based EC reactor. You may be concerned that aeration may affect the concentration distribution of PFAS in solution, and we share this concern. During the experimental

procedure, once a specific time has been reached, the aeration and electrolysis will be terminated. The solution is left to stand for five minutes to ensure that the residual PFAS is evenly distributed.

It is crucial to emphasize that the reported adsorption capacities in this study represent the actual measured PFAS adsorbed amounts (q_t), rather than the theoretical maximum adsorption capacities derived through model fitting. For details, the adsorbed amount (q_t , $\text{mmol}\cdot\text{g}^{-1}$) of PFAS on zinc hydroxide flocs was calculated using the following equation:

$$q_t = 1000 \frac{C_0 - C_t}{\text{Zinc dosage}} \quad (1)$$

where C_0 and C_t are the PFAS concentration in solution at initial and reaction time, respectively. The dissolved zinc dosage ($\text{mg}\cdot\text{L}^{-1}$) was calculated as a function of electrocoagulation time using the following equation:

$$\text{Zinc dosage} = \frac{1000}{V} \times \frac{I \times t_{\text{EC}}}{nF} \times M \times \eta \quad (2)$$

where I and t_{EC} refer to the applied current and time during electrocoagulation, respectively; F is the Faraday's constant; n is the number of electrons in $\text{Zn} - 2e \rightarrow \text{Zn}^{2+}$; M ($65 \text{ g}\cdot\text{mol}^{-1}$) refers to the relative molar mass of zinc; η refers the current efficiency of $\text{Zn} - 2e \rightarrow \text{Zn}^{2+}$, which is determined to 0.91 in the fluorochemical wastewater used in this study.

Considering your comments, we have modified the related contents in the revised manuscript (**Page 24, Lines 549-552; Pages 25-26, Lines 585-588**).

Added: During the EC treatment, the solution pH value was not adjusted or controlled. During the experimental procedure, once a specific time has been reached, the aeration and electrolysis will be terminated. The solution is left to stand for five minutes to ensure that the residual PFAS is evenly distributed.

Added: The adsorbed amount (q_t , $\text{mmol}\cdot\text{g}^{-1}$) of PFAS on zinc hydroxide flocs was calculated using the following equation:

$$q_t = 1000 \frac{C_0 - C_t}{\text{Zinc dosage}} \quad (5)$$

where C_0 and C_t are the PFAS concentration in solution at initial and reaction time, respectively.

Comment 7. It was mentioned that 107 PFAS were detected in the wastewater sample but their concentrations and impacts on the removal process are not clear. PFOA was found in the highest concentration among those PFAS, and treatment results were mostly focused on PFOA control and potential mechanism, not other species with different physicochemical properties. Also, the impacts of the wastewater complexity (due to other components) were not well addressed in the manuscript. The impacts of other components are often very significant to PFAS treatment in the Zn-based EC as published in the previous study (<https://doi.org/10.1016/j.scitotenv.2016.03.114>).

Response: We thank the reviewer for this comment. Our previous study (<https://doi.org/10.1016/j.scitotenv.2016.03.114>) showed that the presence of $\text{CO}_3^{2-}/\text{HCO}_3^-$ significantly reduced the adsorption capacity of hydrolysis products of Zn^{2+} on PFAS, and Cl^- , NO_3^- and SO_4^{2-} (main ions in the fluorochemical wastewater) have less effect on the PFAS sorption. However, the concentration of $\text{CO}_3^{2-}/\text{HCO}_3^-$ in the fluorochemical wastewater was very low, only ~ 26 mgTIC/L (see figure below). The negative effect of $\text{CO}_3^{2-}/\text{HCO}_3^-$ could be avoided by simply adjusting the solution to weak acidity ($\text{pH} < 5.6$, see figure below).

Figure. TC and TIC of the fluorochemical wastewater.

PFOA is the most significant PFAS in the fluorochemical wastewater. Therefore, it is reasonable to utilize PFOA as an indicator to assess the efficacy of PFAS removal. On the other hand, the concentrations and physicochemical properties of 107 PFAS have been well investigated in the original manuscript. Firstly, the selective adsorption coefficient (K_d) values of 107 PFAS were calculated as illustrated in **Figure 3a**, and the results are promising: (1) the AER adsorbs all 107 PFAS despite their extensive variation in chemical structure and concentration, and (2) the Zn-based EC selectively adsorbs hydrophobic PFAS with $\log K_{ow} > 4$ except hydrophilic PFAS. Secondly, the impact of the chemical structure of 107 PFAS on their adsorbability has been extensively discussed, and the detailed discussion was in the section entitled “*Effect of the Structure of PFASs on their Adsorbability*” in the original manuscript (**Lines 250-350**). The proposed Zn-based EC adsorption mechanism is applicable to all PFAS, and not solely to PFOA (**Lines 166-249**).

Considering the comments, the following changes have been made: (1) the TIC value has been included in **Supplementary Table S2**, and (2) a comparison of the adsorption capacities of more PFAS has been made and discussed in the revised manuscript (**Pages 11-12, Lines 230-256**).

Modified: To elucidate the selective adsorption mechanism in the Zn-based EC process, we further investigated the adsorption kinetics of eight PFAS (for which reliable $\log K_{ow}$ values were available) with varying chain-lengths and structural characteristics in a simulated solution. Zinc hydroxide flocs presented extremely rapid adsorption of all eight PFAS with equilibrium time (t_{eq}) less than of 2 min (Supplementary Fig. 9), whereas AC and AER widely used in current applications had t_{eq} of tens of hours or more (Supplementary Tables 4-6). The observed maximum adsorption amount (q_m , mmol PFAS·g⁻¹ zinc hydroxide flocs) was monotonically correlated with their hydrophobicity and chain-lengths (Fig. 3b, Supplementary Table 3 and Supplementary Fig. 9), suggesting a pivotal role of hydrophobic

interaction. The weakly hydrophobic PFBA (C4, $\log K_{ow} = 2.14$) had a q_m value of $< 0.1 \text{ mmol}\cdot\text{g}^{-1}$, the moderately hydrophobic PFHxA (C6, $\log K_{ow} = 3.48$) had an elevated q_m of $0.6 \pm 0.3 \text{ mmol}\cdot\text{g}^{-1}$, and the highly hydrophobic PFDA (C10, $\log K_{ow} = 6.15$) achieved an ultra-high q_m of $>23 \text{ mmol}\cdot\text{g}^{-1}$ ($>10 \text{ g}\cdot\text{g}^{-1}$). For the GenX (C6, $\log K_{ow} = 3.36$) with less hydrophobicity, the achieved q_m was $1.6 \pm 0.7 \text{ mmol}\cdot\text{g}^{-1}$, lower than that of AER but also several times higher than that of the AC and other reported adsorbents (Fig. 3c). However, the most discussed long-chain PFAS, i.e., PFOA (C8, $\log K_{ow}=4.81$) and PFOS (C8, $\log K_{ow}=4.88$), their q_m values were estimated to be as high as $6.4 \pm 0.4 \text{ mmol}\cdot\text{g}^{-1}$ ($2.6 \text{ g}\cdot\text{g}^{-1}$) and $7.5 \pm 0.04 \text{ mmol}\cdot\text{g}^{-1}$ ($3.8 \text{ g}\cdot\text{g}^{-1}$), respectively. To the best of our knowledge, these achieved q_m for hydrophobic PFAS such as PFOA and PFOS are the highest of all values reported in the literature, which are over an order-of-magnitude higher than the theoretical maximum adsorption capacity derived from the adsorption model fitting that of the data for the benchmark AC and several times higher than that of the AER (Fig. 3c and Supplementary Tables 4-6). Furthermore, the dynamic adsorption capacity (q_{dyn} , $\text{mmol}\cdot\text{g}^{-1}\cdot\text{h}^{-1}$) was more than 1~4 orders of magnitude higher than that of the benchmark AC and AER and other state-of-art adsorbents reported in the literature (Fig. 3c). The q_{dyn} ($\text{mmol}\cdot\text{g}^{-1}\cdot\text{h}^{-1}$) was calculated according to the equation, $q_{dyn} = q_m/t_{eq}$, in which the t_{eq} represents the adsorption equilibration time⁴³. The kinetics of PFAS adsorption by conventional porous adsorbents are primarily constrained by the diffusion process within the particles. In contrast, the adsorption of PFAS on the zinc hydroxide flocs predominantly occurs through a particle-surface adsorption process, as depicted in Figure 3f, resulting in significantly faster adsorption rates.

Figure 3. (b) The achieved maximum adsorption amount (q_m , $\text{mmol}\cdot\text{g}^{-1}$) of 8 PFAS (C4~C10) in simulated solution by Zn-based EC vs. their $\log K_{ow}$ values, and (c) comparison of the q_m and q_{dyn} values for PFOA, PFOS and GenX by Zn-based EC with various adsorbents in reported literature (listed in Supplementary Tables 4-6).

Comment 8. L306, for the full-scale simulation, only anion exchange resin was examined, nothing was conducted for Zn-based EC for the full scale simulation. This study examined those processes in one single process train but only one part of the process was assessed for the pilot-scale operation. It seems not appropriate.

Response: Thanks for your comment. One of the significant advantages of electrochemical techniques is their capacity for linear scale up. Under identical reaction conditions (current density, surface/volume ratios, treatment time, etc.), the outcomes from small trials of Zn-based EC can be linearly scaled up to pilot or even full scale. A 20 L scale-up test was conducted, yielding results that were consistent with those of the bench-scale test. Considering the comment, we have added additional explanations in the revised manuscript (**Page 17, Lines 379-381**).

Added: A scale-up test of the Zn-based EC was conducted with a volume of 20 L (Supplementary Fig. 16), yielding results that were largely consistent with those of the bench-scale test.

Supplementary Fig. 16 The images of the 20 L pilot Zn-based EC reactor.

Minor comments:

Comment 9. L63, the cited reference was published in 2007, now we have many more studies regarding treating PFAS in complex water, it is not limited.

Response: Thanks for the suggestion. We have updated the references in the revised manuscript.

Comment 10. L76, no rationale was provided to justify the use of Zn-based EC for this study. How did you know the hydrophobicity of Zn hydroxide flocs?

Response: Thanks for the suggestion. We agree with you, and the relevant contents have been added to the revised manuscript (**See Comment 1**).

How did you know the hydrophobicity of Zn hydroxide flocs? We observed that the flocs initially produced by electrolysis exhibited a tendency to float on the surface of the solution like an oil, displaying notable hydrophobic characteristics that differed significantly from those of conventional Al and Fe flocs. To response of your concern, we have also measured the contact angles of water on the three flocs. The results demonstrated that the Zn flocs exhibited a considerably greater contact angle ($\sim 90^\circ$), compared with the Al ($\sim 60^\circ$) and Fe ($\sim 33^\circ$) flocs. The relevant contents have been included in the revised manuscript (**Pages 12, Lines 269-272**).

Added: The characterization results indicated that the zinc hydroxide flocs have a contact angle of $\sim 90^\circ$ (Supplementary Fig. 11), which is significantly greater than the contact angles of the aluminum ($\sim 60^\circ$) and ferric hydroxide flocs ($\sim 33^\circ$).

Supplementary Fig. 11 The contact angles of different flocs.

Comment 11. L135, what does mean the bubble diameter here? Were there any aeration bubbles during EC?

Response: Figure 2d and Supplementary Figures 3-6 in the original manuscript are bubble charts, and the bubble diameters represent the concentrations of PFAS. The solution was stirred and purged with pure N₂ (1 L(N₂)/L(solution)/min) (see **Comment 4**).

Comment 12. L178-L179, this is not interesting since the same result was already reported in the previous study published in 2015 (<https://doi.org/10.1021/acs.est.5b02092>).

Response: We agree that our previous study (<https://doi.org/10.1021/acs.est.5b02092>) has discovered that the Zn-based EC could preferentially adsorb the more hydrophobic PFAS. However, the aforementioned study solely examined the adsorption capacities of Zn-based EC for six PFAS (PFDA, PFNA, PFOS, PFOA, PFHpA and PFBA) in a simulated solution. In the present study, we introduced the selective adsorption coefficient (K_d), which considers both the physicochemical properties and the concentration of PFAS, to characterize the adsorbability of Zn-based EC for 107 PFAS in a complex water matrix of the fluorochemical wastewater. Our present study is distinguished from our previous one by its methodology and the much greater number of PFAS with diverse structures and the complex water matrix.

Comment 13. L191, the kinetic study depending on PFAS chain lengths also reported by the previous study (<https://doi.org/10.1021/acs.est.5b02092>).

Response: We agree that our previous study has investigated the adsorption capacities of Zn-based EC for six PFAS (PFDA, PFNA, PFOS, PFOA, PFHpA and PFBA) in a simulated solution. In the revised manuscript, we also conducted tests on the adsorption capacities of additional structural PFAS, including novel ether PFAS (**Pages 11-12, Lines 230-256**). We would like to retain this section, as it will facilitate readers to better understand the mechanisms of PFAS removal by Zn-based EC.

Modified: To elucidate the selective adsorption mechanism in the Zn-based EC process, we further investigated the adsorption kinetics of eight PFAS (for which reliable $\log K_{ow}$ values were available) with varying chain-lengths and structural characteristics in a simulated solution. Zinc hydroxide flocs presented extremely rapid adsorption of all eight PFAS with

equilibrium time (t_{eq}) less than of 2 min (Supplementary Fig. 9), whereas AC and AER widely used in current applications had t_{eq} of tens of hours or more (Supplementary Tables 4-6). The observed maximum adsorption amount (q_m , mmol PFAS·g⁻¹ zinc hydroxide flocs) was monotonically correlated with their hydrophobicity and chain-lengths (Fig. 3b, Supplementary Table 3 and Supplementary Fig. 9), suggesting a pivotal role of hydrophobic interaction. The weakly hydrophobic PFBA (C4, log K_{ow} = 2.14) had a q_m value of < 0.1 mmol·g⁻¹, the moderately hydrophobic PFH_xA (C6, log K_{ow} = 3.48) had an elevated q_m of 0.6 ± 0.3 mmol·g⁻¹, and the highly hydrophobic PFDA (C10, log K_{ow} = 6.15) achieved an ultra-high q_m of >23 mmol·g⁻¹ (>10 g·g⁻¹). For the GenX (C6, log K_{ow} = 3.36) with less hydrophobicity, the achieved q_m was 1.6 ± 0.7 mmol·g⁻¹, lower than that of AER but also several times higher than that of the AC and other reported adsorbents (Fig. 3c). However, the most discussed long-chain PFAS, i.e., PFOA (C8, log K_{ow} =4.81) and PFOS (C8, log K_{ow} =4.88), their q_m values were estimated to be as high as 6.4 ± 0.4 mmol·g⁻¹ (2.6 g·g⁻¹) and 7.5 ± 0.04 mmol·g⁻¹ (3.8 g·g⁻¹), respectively. To the best of our knowledge, these achieved q_m for hydrophobic PFAS such as PFOA and PFOS are the highest of all values reported in the literature, which are over an order-of-magnitude higher than the theoretical maximum adsorption capacity derived from the adsorption model fitting that of the data for the benchmark AC and several times higher than that of the AER (Fig. 3c and Supplementary Tables 4-6). Furthermore, the dynamic adsorption capacity (q_{dyn} , mmol·g⁻¹·h⁻¹) was more than 1~4 orders of magnitude higher than that of the benchmark AC and AER and other state-of-art adsorbents reported in the literature (Fig. 3c). The q_{dyn} (mmol·g⁻¹·h⁻¹) was calculated according to the equation, $q_{dyn} = q_m/t_{eq}$, in which the t_{eq} represents the adsorption equilibration time⁴³. The kinetics of PFAS adsorption by conventional porous adsorbents are primarily constrained by the diffusion process within the particles. In contrast, the adsorption of PFAS on the zinc hydroxide flocs predominantly occurs through a particle-surface adsorption process, as depicted in Figure 3f, resulting in significantly faster adsorption rates.

Figure 3. (b) The achieved maximum adsorption amount (q_m , mmol·g⁻¹) of 8 PFAS (C4~C10) in simulated solution by Zn-based EC vs. their $\log K_{ow}$ values, and (c) comparison of the q_m and q_{dyn} values for PFOA, PFOS and GenX by Zn-based EC with various adsorbents in reported literature (listed in Supplementary Tables 4-6).

Comment 14. L305, however, PFAS removal by EC can be significantly hindered by other cations or anions in the water. See (<https://doi.org/10.1016/j.scitotenv.2016.03.114>).

Response: Thanks for your comment. As shown in response to **Comment 7**, our previous study showed that the presence of $\text{CO}_3^{2-}/\text{HCO}_3^-$ significantly reduced the adsorption capacity of hydrolysis products of Zn^{2+} on PFAS, while Cl^- , NO_3^- and SO_4^{2-} (the main ions in the fluorochemical wastewater) featured less effect on the PFAS sorption. The concentration of $\text{CO}_3^{2-}/\text{HCO}_3^-$ in the fluorochemical wastewater was very low, only ~ 26 mgTIC/L. The negative effect of $\text{CO}_3^{2-}/\text{HCO}_3^-$ could be avoided by simply adjusting the solution to weak acid (see **Comment 7**).

We also agree with you that certain cations or anions in other water samples may have a significant impact on the PFAS removal by Zn-based EC. These ions may affect the formation of Zn flocs, reducing the adsorption of PFAS. We have tried to identify the composition of Zn flocs by high-resolution mass spectrometry, and the results showed that the hydrolysis process of Zn^{2+} was significantly different from that of Al^{3+} , so it is not confirmed yet which structure of Zn flocs is responsibility for the adsorption of PFAS. Hence, it is hard to know what ion will affect PFAS removal. Further, experimentally assessment of other types of wastewaters with certain ion contamination is not the focus of this study, which is beyond the scope of this manuscript and will be evaluated in future work. We have added the relevant content in the revised manuscript (**Page 22, Lines 509-513**).

Added: Additionally, our studies show that zinc hydroxide flocs generated in a simulated solution with NaCl as the supporting electrolyte exhibited superior PFAS adsorption capabilities. Therefore, a comprehensive investigation into the hydrolysis behavior of Zn^{2+} and the identification of the key hydrolysis products that are responsible for the adsorption of PFAS is warranted, which is crucial for optimizing the Zn-based EC process.

Reviewer #2

In this manuscript, the authors report a bench-scale application of zinc-based electrocoagulation and anion exchange resin to remove PFAS from concentrated industrial wastewater. The combination approach was found to be much more efficient at removing hydrophobic PFAS than anion exchange treatment alone. This type of work is timely and important; needs for efficient treatment methods for PFAS-laden industrial discharges are growing as PFAS become increasingly regulated. The data and methodology supporting this finding appear sound. However, the quality of the manuscript could be significantly improved if some minor technical and presentation issues were addressed.

Response: We truly appreciate the very constructive comments that help to clarify and improve our manuscript. We have carefully revised the manuscript according to the comments, as detailed below.

Comment 1. The focus of this study is clearly on PFAS-rich industrial effluents, but readers might be interested in applying the treatment train in other contexts. Consider adding some discussion of the feasibility of this treatment train to applications such as site remediation or landfill leachate treatment.

Response: That's a good suggestion. Considering your suggestions, more discussions have been included in the revised manuscript (**Page 14, Lines 307-311; Page 22, Lines 499-506**).

Modified: In principle, this unique mechanism would also enable the Zn-based EC to avoid adverse effects from other coexisting contaminants and DOMs in solution, as these competing constituents tend not to be highly hydrophobic. Therefore, the Zn-based EC could be an effective technique for treating highly complex waste streams such as aqueous fire forming foams (AFFFs) solution, landfill leachate, and still bottoms liquid waste containing high concentrations of PFAS from adsorbent regeneration.

Added: For example, aqueous film-forming foams contain substantial PFAS; however, the efficacy of direct treatment by advanced redox processes is limited due to the coexistence of numerous additional components, including sodium dodecyl sulfate (SDS), a hydrocarbon surfactant, and diethylene glycol butyl ether (DGBE), an organic solvent. Given the poor hydrophobicity of SDS ($\log K_{ow}=1.69$) and DGBE ($\log K_{ow}=0.29$), our preliminary results indicate that the Zn-based EC can achieve selective enrichment of PFAS from aqueous film-forming foams, which can then be effectively degraded by advanced redox processes.

Comment 2. The Zn-based EC process appears in the introduction at L76 but isn't given a proper introduction. Where has this technique been used before? Why was it selected for this study over other metal-based EC processes? Consider including a citation for SI ref 14 and other relevant

literature here. Adding references or reviews in the introduction will help readers more quickly understand the novelty and importance of this paper.

Response: I am most grateful for your kind suggestion! We offer our sincerest apologies for failing to identify that the most critical reference (<https://doi.org/10.1021/acs.est.5b02092>, our previous study) was not cited in the manuscript. In light of the length of the manuscript, the section on research in progress in the Zn-based EC has been omitted. The revised manuscript now includes relevant content related to the progress of Zn-based EC (**Pages 4-5, Lines 80-99**).

Added: One of the defining characteristics of PFAS is its exceptional surface activity, which allows it to be adsorbed in multiple layers through semi-micellar or micellar adsorption. This significantly boosts its adsorption capacity. An example of this is mineral flotation, where trapping agents are adsorbed onto hydrophobic mineral particles through semi-micellar adsorption, enabling selective mineral capture. Inspired by this, we propose a reverse mineral flotation process using hydrophobic "mineral particles" as adsorbents to selectively extract PFAS from water via semi-micellar adsorption. We found that zinc hydroxide flocs, formed in situ by electrolysis, exhibit properties of hydrophobic "mineral particles," thus enabling them to rapidly adsorb hydrophobic long-chain PFAS from water with extremely high adsorption capacities^{34, 35}. For example, these zinc hydroxide flocs achieved an equilibrium adsorbed amount (q_e) of up to 5.74/7.69 mmol g⁻¹ (Zn) for PFOA/PFOS within minutes at an initial concentration of 0.5 mM³⁴. We therefore hypothesize that Zn-based electrocoagulation (EC) can effectively capture long-chain PFAS in the fluorochemical wastewater, while the existing AC and/or AER adsorption devices can then remove the remaining low concentrations of short-chain PFAS. This approach is founded on two primary observations: 1) Despite the presence of hundreds of PFAS types in fluorochemical wastewater, long-chain PFAS, particularly PFOA, dominate the total PFAS concentration Zn-based EC also removes dissolved organic matter (DOM) and traps colloidal particles from wastewater, significantly mitigating the impact of these competing constituents on subsequent adsorption processes. This strategy represents the first attempt to achieve broad-spectrum removal of hundreds of PFAS from real fluorochemical wastewaters, potentially offering new options for tackling PFAS in complex industrial waste streams.

Comment 3. The constant diffusivity scheme for the RSSCT was selected without commentary. Some discussion as to whether this approach can be reasonably expected to produce correspondence for pilot or full scale-systems would be beneficial. Some recent work on the subject might be worth discussion or citation: Liu et al., 2024 <https://doi.org/10.1016/j.watres.2024.121661> and Cheng et al., 2024 <https://doi.org/10.1016/j.watres.2023.120956> seem relevant.

Response: Your comments are appreciated. We examined the PFAS adsorption capacities of the as-received PFA694E and crushed PFA694E. Considering your comments, we have added more discussions and cited the related refs in the revised manuscript (**Pages 17, Lines 381-386**).

Added: The breakthrough of PFAS in RSSCT is dependent on the particle size of the adsorbent, and hence the constant diffusion model is appropriate for designing RSSCT experiments⁵⁰. The crushed PFA694E exhibited a comparable total anion exchange capacity to the as-received (6.6 vs. 5.9 mmol Cl⁻ per mL resin, Supplementary Table 8). The RSSCT design parameters and the simulated full-scale PFA694E bed parameters are presented in the Method section and in Supplementary Table 9.

Comment 4. L27: The statement “resulting in the highest 26 adsorption capacities among all reported adsorbents” might need some clarification or qualification because, for instance, the adsorption capacity for PFBA is about zero.

Response: Your comments are appreciated. We note that this statement may mislead readers. In the revised version (**Pages 2, Lines 24-27**), a more rigorous presentation has been provided.

Original: The “zero-carbon” adsorbent, zinc hydroxide flocs, generated in-situ by Zn-based EC bulk removes PFAS with $\log K_{ow} > 4$ through a semi-micellar adsorption mechanism similar to mineral flotation, resulting in the highest adsorption capacities all reported adsorbents.

Modified: The “zero-carbon” adsorbent, zinc hydroxide flocs, generated in-situ by Zn-based EC bulk removes hydrophobic PFAS with $\log K_{ow} > 4$ through a semi-micellar adsorption mechanism similar to mineral flotation, resulting in adsorption capacities at the optimal level of all reported adsorbents.

Comment 5. L55: It is not clear from the references or from the text what is meant by “special helical structure” and how it is relevant to this work. Are any of the treated PFAS helical?

Response: Your comments are appreciated. The spatial helical structure of PFAS gives it better chemical stability. In the revised version (**Pages 3, Lines 53-54**), the sentence has been rewritten.

Figure. The molecular structures of octane zigzag configuration (top) and perfluorooctane helical structure (bottom)

Original: Eliminating PFAS from waste streams remains a significant challenge despite intensive efforts, as traditional chemical and biological treatment processes are ineffective due to the strong carbon-fluorine bond and the special helical structure.

Modified: Eliminating PFAS from waste streams poses a significant challenge due to their highly stable spiral structure and strong carbon-fluorine bonds (531.5 kJ·mol⁻¹).

Comment 6. L58: delete “necessary.” Some membrane techniques are used to concentration PFAS solutions for destructive techniques.

Response: Revised as suggested.

Comment 7. L123: The choice of the Langmuir isotherm model needs discussion. At least some discussion of traditional ion exchange isotherms would be helpful here. The empirical application single-component Langmuir model here does make some sense because the wastewater matrix is so complex and more rigorous treatment of the IX isotherms (Wahman et al., 2023 <https://doi.org/10.1021/acsestwater.3c00396>; Smith et al., 2023 <https://doi.org/10.1021/acsestwater.2c00572>) probably aren’t feasible, but there are still opportunities to learn more about the mechanisms by at least thinking about them. For instance, how does the estimated maximum adsorption capacity estimated in this work compare to the anion exchange capacity of the resin? What does that imply about the mechanism of PFAS removal from the wastewater in this case?

Response: Thanks for your suggestion, we have added additional discussions in the revised manuscript (**Pages 7-8, Lines 155-164**).

Added: It was noted that the quantity of Cl⁻ (1416.4 ± 246.3 μmol·g⁻¹) exchanged from the PFA694E significantly surpassed the amount of PFOA adsorbed (46.4 ± 2.6 μmol·g⁻¹) (Supplementary Fig. 7a). This observation suggests that the elevated concentration of background inorganic anions in fluorochemical wastewater markedly competes with PFAS for adsorption sites. Wahman et al.³⁶ have previously reported that the exchange adsorption of PFAS by strong base AERs is affected by the presence of nitrite, sulfate, and bicarbonate. In this study, NO₃⁻ (382.4 ± 9.9 μmol·g⁻¹ adsorbed) and SO₄²⁻ (389.4 ± 40.7 μmol·g⁻¹ adsorbed) emerged as the predominant competing anions (Supplementary Fig. 7a), accounting for over 89% of the chloride ions exchanged out of PFA694E. Consequently, it is clear that effective pretreatment strategies are essential to reduce the interference of competing ions when utilizing AERs in the treatment of PFAS-laden industrial wastewaters.

Comment 8. Line 170: What is that selective adsorption coefficient? It almost looks like a separation factor. But it’s just all the PFAS? [What is in this fluorochemical wastewater? Are most of the ions PFAS?] Interesting that most PFAS on the IX resins have K_d of about 1. What does that mean? Would it make more sense to use an equilibrium separation factor with a reference ion (even if it’s one of the PFAS?).

Response: I am grateful for your suggestion. The selective adsorption coefficient (K_d) represents the tendency of the interface to favor the adsorption of one specific PFAS over others. The

K_d values of PFA694E for all 107 PFAS were around 1 (Fig. 3a), indicating that it did not show significant preferential adsorption for any of the 107 PFAS. It may be beneficial to employ an equilibrium separation factor for the reference ion, but we believe that the K_d values could also provide the reader with a comprehensive understanding. Thank you again for your suggestions. Considering the comment, we have provided a clearer definition of K_d in the revised manuscript (**Pages 10, Lines 206-213**).

Modified: To examine the adsorption behavior of PFAS with different structures, the selective adsorption coefficient (K_d)⁴⁰ was introduced as a measure of PFAS adsorbability. The K_d value represents the tendency of the interface to favor the adsorption of one specific PFAS over others, and a K_d value greater than 1 for PFAS indicates that it will be preferentially adsorbed. The coefficient is calculated using the following equation 1:

$$K_d = \frac{\omega_b}{\omega_a} / \frac{(1-\omega_b)}{(1-\omega_a)} \quad (1)$$

where ω_b and ω_a represent the molar concentration fraction of a specific PFAS to the total 107 PFAS in the fluorochemical wastewater before and after adsorption, respectively.

Comment 9. Line 193: What is the particle size of the Zn EC flocs? This might matter for discussions of process kinetics. On a related note, what does a full-scale Zn EC system look like? Would the floc particle size be similar to this work?

Response: Your comments are appreciated. We determined the particle size of Zn flocs in solution using a laser particle-size analyzer (Mastersizer 3000 Malvern, UK). The results show that Zn flocs adsorbed with PFOA appear to aggregate and grow, which is consistent with the SEM observation.

Figure particle size distribution of the zinc hydroxide flocs generated in-situ by Zn-based EC in simulated solution with or without PFOA

One of the significant advantages of electrochemical techniques is their capacity for linear scaling. Under identical reaction conditions (current density, surface/volume ratios, treatment time,

etc.), the outcomes from small trials of Zn-based EC can be linearly scaled up to pilot or even engineering scale. A 20 L scale-up test was conducted, yielding results that were largely consistent with those of the bench-scale test. We also measured the particle size of the Zn flocs in the scale-up system. The results show that the particle size of Zn flocs in the scale-up system ($DV_{50}=73.6\ \mu\text{m}$) is similar to that in the bench-scale test ($DV_{50}=79.8\ \mu\text{m}$).

Considering the comments, we have added additional explanations in the revised manuscript (**Page 12, Lines 260-264; Page 17, Lines 379-381**).

Added: PFAS seemed to act as a binder to tightly aggregate and completely cover the dispersed Zn hydroxide flocs. As shown in Fig. 3e, the particle size of PFOA adsorbed zinc hydroxide flocs ($DV_{50}=7.32\ \mu\text{m}$) had a much greater diameter than that of the fresh zinc hydroxide flocs ($DV_{50}=79.76\ \mu\text{m}$). This phenomenon is interesting, as it suggests that the adsorption of PFOA and the aggregation and growth of zinc hydroxide flocs occur simultaneously.

Added: A scale-up test of the Zn-based EC was conducted with a volume of 20 L (Supplementary Fig. 16), yielding results that were largely consistent with those of the bench-scale test.

Supplementary Fig. 16 The images of the 20 L pilot Zn-based EC reactor.

Comment 10. Line 206: What is the physical significance of q_{dyn} (Called q_{uni} in SI Table S3)? Is it an empirical parameter for comparing kinetics across processes? If that is the case, some remarks on the generalizability of this comparison are needed. Kinetics of adsorption on ion exchange resins and activated carbons are usually determined by diffusive processes and depend on particle size and adsorbent dose---among other things---while Zn-EC likely follows a different mechanism. Thus, q_{dyn} may depend differently on changes in conditions across techniques.

Response: Your comments are appreciated. We are very sorry for giving the incorrect definition of q_{dyn} . The q_{dyn} ($\text{mmol}\cdot\text{g}^{-1}\cdot\text{h}^{-1}$), which is defined as the ratio of the maximum adsorption capacity (q_{m} , $\text{mmol}\cdot\text{g}^{-1}$) to the adsorption equilibrium time (t_{eq} , h), is employed in many reported studies (such as *Chem. Eng. J* 2022, 433, 133271) to compare the adsorption performance of different adsorbents for PFAS. Considering your suggestions, we have added more explanations in the revised manuscript (**Pages 11-12, Lines 249-256**).

Added: Furthermore, the dynamic adsorption capacity (q_{dyn} , $\text{mmol}\cdot\text{g}^{-1}\cdot\text{h}^{-1}$) was more than 1~4 orders of magnitude higher than that of the benchmark AC and AER and other state-of-art adsorbents reported in the literature (Fig. 3c). The q_{dyn} ($\text{mmol}\cdot\text{g}^{-1}\cdot\text{h}^{-1}$) was calculated

according to the equation, $q_{dyn} = q_m / t_{eq}$, in which the t_{eq} represents the adsorption equilibration time⁴³. The kinetics of PFAS adsorption by conventional porous adsorbents are primarily constrained by the diffusion process within the particles. In contrast, the adsorption of PFAS on the zinc hydroxide flocs predominantly occurs through a particle-surface adsorption process, as depicted in Figure 3f, resulting in significantly faster adsorption rates.

Figure 3. (b) The achieved maximum adsorption amount (q_m , $\text{mmol}\cdot\text{g}^{-1}$) of 8 PFAS (C4~C10) in simulated solution by Zn-based EC vs. their $\log K_{ow}$ values, and (c) comparison of the q_m and q_{dyn} values for PFOA, PFOS and GenX by Zn-based EC with various adsorbents in reported literature (listed in Supplementary Tables 4-6).

Wang H.; Mills R.; Qu K., Hower J.C.; Mottaleb M., A.; Bhattacharyya D., Xu, Z. Rapid removal of PFOA and PFOS via modified industrial solid waste: mechanisms and influences of water matrices. *Chem. Eng. J* 2022, 433, 133271.

Comment 11. L346 – 349: Argument is hard to follow from the petal plots. Consider changing to more traditional bar plots to make these points. Was the adsorbed PFOA concentration “dramatically” different?

Response: Your comments are appreciated. The absorbed PFOA amounts were 33.4 and 19.2 $\text{mmol}\cdot\text{g}^{-1}$ for the PFA694E by feeding the untreated raw wastewater and the Zn-based EC treated wastewater, respectively. We have provided clearer petal plots and have also used bar charts to show the loading amounts of 23 PFAS and included them in the revised manuscript (Pages 19, Lines 422-425).

Modified: Expanded enlarged petal plots in Figs. 5f,5g (bar chart see Supplementary Fig. 20) showed significant increases in loadings for most of the 23 PFAS, except for several highly hydrophobic PFAS (such as PFOA, PFNA, 3:3 I-FTHxA and 4:4 I-FTOA) whose concentrations were dramatically or nearly completely removed by the front-end Zn-based EC treatment.

Fig. 5 e Total concentrations of 23 representative PFAS and their percentage contribution in fluorochemical wastewater: untreated (outer ring) vs. Zn-based EC treated (inter ring). **f, g** Finite loading of 23 representative PFAS onto the PFA694E bed fed with untreated(**f**) or Zn-based EC treated (**g**) fluorochemical wastewater.

Supplementary Fig. 20 Finite loading of 23 representative PFAS onto the PFA694E bed fed with untreated (yellow bar) or Zn-based EC treated (green bar) fluorochemical wastewater.

Comment 12. L399—401: It is not clear how the evaluations in 6c and Table S6* [actually S7] were determined. The categories and scores seem arbitrary. Is this part of some more formal decision-making framework or developed ad hoc for this manuscript?

Response: Thank you for pointing out the mistake! Table S7 was specifically customized for this study. To compare the three treatment options in as much detail as possible, we defined six criteria and gave ratings including PFAS removal efficiency and broad spectrum, environmental friendliness, solid waste production, economic and automatic operability. Similar scoring methods have been employed by many studies (such as *Nature Water* 2024, 2, 52-61) to compare performance of disparate technologies or materials.

Chen Y.; Yang S.; Wang Z.; Elimelech M. Transforming membrane distillation to a membraneless fabric distillation for desalination. *Nat. Water* 2024, 2, 52-61.

Comment 13. L431: (And other locations) There is an off-by-one error in the numbering of supplementary tables.

Response: We have examined the full manuscript and made corrections.

Comment 14. Line 445 mentions isotherms fits using the Freundlich model, but Freundlich fits do not appear to have been done.

Response: Thanks for pointing out the mistake. In the revised manuscript (**Page 24, Line 558**), we have clarified the statement.

Original: All isotherm data were fitted by the Langmuir and/or Freundlich model.

Modified: All isotherm data were fitted by the Langmuir model.

Comment 15. L451. How was the resin crushed? This could be important because it is not yet obvious if or how crushing method affects ion exchange RSSCT correspondence.

Response: That is a good question! The PFA694E resin used in this study was crushed with a grinder and then sieved to obtain a specific size. A recent study by Cheng & Knappe (2024) suggested that crushing resin did not substantially change the bed density and total anion exchange capacity. Considering the comment, we have tested the adsorption capacities of as-received and crushed PFA694E, and the results demonstrated that the crushed PFA694E exhibited a comparable PFOA adsorption capacity to the as-received. Therefore, crushed PFA694E meets an important RSSCT requirement, i.e., the capacity and bed density of the sorbent before and after crushing is well matched. Considering your comment, we have added more explanations in the revised manuscript (**Page 17, Lines 383-385**).

Added: The crushed PFA694E exhibited a comparable total anion exchange capacity to the as-received (6.6 vs. 5.9 mmol Cl⁻ per mL resin, Supplementary Table 8).

Cheng L., Knappe D.R.U. Removal of per- and polyfluoroalkyl substances by anion exchange resins: scale-up of rapid small-scale column test data. *Water Res.* 2024, 249, 120956.

Comment 16. Ref 1 needs to also cite federal register for the PFAS drinking water rule. USEPA, (2024). PFAS National Primary Drinking Water Regulation Rulemaking Fed. Regist., 89(82): p. 32532-32757

Response: Revised as suggested.

Comment 17. Figure 2: Check that panel letter labels in the caption are correct.

Response: We have double-checked.

Comment 18. Figures 2d, S3--6: What are the subplots in the two panels? Specifically, what are the 3 columns?

Response: The concentration differences between 107 PFAS were more than 3 orders of magnitude, and thus the data were plotted in three concentration intervals, i.e., <100 µg/L (the left column), 101-1000 µg/L (the middle column), and >1000 µg/L (the right column). Considering your comment, we re-plotted for a more accurate representation in the revised manuscript.

Modified:

Comment 19. Figure 3d: Panels are hard to see/interpret --- especially the spectra.

Response: Higher resolution figures are provided in the revised manuscript. The EDX spectra has been removed to the Supplementary Information.

Modified:

Supplementary Fig. 10 (a) Characterization of the zinc hydroxide flocs generated in-situ by Zn-based EC in simulated solution with or without PFAS: (a) XPS spectra, (b) high resolution of C1s XPS spectra, and (c) SEM-EDX.

Comment 20. Figure 5b: Legend on (b)? Markers and line styles are different from (a).

Response: We have corrected the colors of the curves in Figure 5b.

Modified:

Comment 21. Figure 5 e,f,g: The nested petal plots and ring plots are difficult to read and interpret. Consider breaking out separate pie and bar charts to highlight important trends more clearly.

Response: Thanks for your good suggestions. We have provided clearer petal plots and have also used bar charts to show the loading amounts of 23 PFAS and included them in the revised manuscript (see Comment 11).

Comment 22. Figure S9: Rightmost panels are too condensed to be interpretable.

Response: Higher resolution figures are provided in the revised manuscript, see Comment 19.

Comment 23. Figure S13 (and others): Text is too small and compressed to read easily, even zoomed in digitally.

Response: Higher resolution figures are provided in the revised manuscript.

Modified:

Supplementary Fig. 17 Concentrations of 107 PFAS in the RSSCT influent at 1.03×10^3 BV: (a) 9 PFCA, (b) 14 Cl/I-PFAA, (c) 52 Ether-PFAA, and (d) 32 H-PFAA.

Comment 24. SI Title: “realities” should be “realizes.”

Response: Revised as suggested.

Reviewer #3

This manuscript makes a major contribution to advancing treatment for industrial wastewaters. The authors tested very high PFAS concentrations (mg/L), as opposed to lower ng/L levels that drinking water standards require, but the authors did excellent work on their system and proposed mechanisms to explain findings. They studied a much wider range of PFAS than most studies, and the graphics are phenomenal. The following are minor suggestions to improve clarity.

Response: Your praise for our work is much appreciated. We truly appreciate the very constructive comments that help to clarify and improve our manuscript. We have carefully revised the manuscript according to the comments, as detailed below.

Comment 1. Line 23: “PFASs” does not need the second s because PFAS is plural as defined in line 20

Response: Your comments are appreciated. All “PFASs” has been changed to “PFAS” in the revised manuscript.

Comment 2. “iodinated PFAS, in which the fluorine atom is replaced by an iodine atom...” if the F is replaced is it still PFAS? This line is confusing. Please edit for clarity. It sounds like PFAS are becoming PIAS.

Response: Thanks for your comment. It should be noted that not all fluorine atoms in the PFAS are replaced by iodine. In fact, the most iodinated-PFAS identified in the fluorochemical wastewater only contains an iodine atom (see figure below). In US EPA, PFAS have been identified as a group of per- and polyfluoroalkyl substances, incompletely F-substituted compounds, their precursors, and their transformation products. Specifically, chlorinated and brominated PFAS are also included in the list for PFAS contaminants. Further, we have published two of our previously published papers on the discovery of iodinated-PFAS.

Considering the comment, we have modified the sentence in the revised manuscript (**Page 2, Lines 29-30**).

Original: It was also observed that iodinated PFAS, in which the fluorine atom is replaced by an iodine atom...

Modified: It was also observed that iodinated-PFAS, with some fluorine atoms are replaced by iodine atoms...

1. Tang C.; Zhu Y.; Liang Y.; Zeng Y.; Peng X.; Mai B.; Xu J.; Huang Q.; Lin H. First discovery of iodinated polyfluoroalkyl acid by nontarget mass-spectrometric analysis and iodine-specific screening algorithm. *Environ. Sci. Technol.* 2023, 57, 1378-1390.
2. Tang C.; Zheng R.; Zhu Y.; Liang Y.; Liang Y.; Liang, S.; Xu J.; Zeng Y.; Luo X.; Lin H.; Huang Q.; Mai B. Nontarget Analysis and Comprehensive Characterization of Iodinated Polyfluoroalkyl Acids in Wastewater

Figure The identified I-PFAA in the fluorochemical wastewater

Comment 3. Line 36: “a few days ago, the Biden-Harris administration...” I recommend not being so specific on timing because it can make the paper look outdated quickly.

Response: That is a good suggestion. The sentence has been modified in the revised manuscript (Page 3, Lines 39-42).

Original: A few days ago, the Biden-Harris Administration issued the legally binding U.S. National Drinking Water Standard for per- and polyfluoroalkyl substances (PFAS) to ensure that everyone has access to clean, safe drinking water, which will ultimately reduce PFAS exposure for more than 100 million Americans.

Modified: In response to this public health crisis, the Biden-Harris Administration issued the legally binding U.S. National Drinking Water Standard for PFAS to ensure that everyone has access to clean and safe drinking water, which will ultimately reduce PFAS exposure for more than 100 million Americans¹.

Comment 4. Introduction: Studying 107 PFAS is impressive. The authors could add a bit more information on the range of chemical properties such as Know, solubility, MW or other to help show how broad the PFAS were that they studied. For example, did they study short-chain and long chain? Cationic, anionic? Sulfonates and carboxylic acids? It would be worth mentioning these key groups PFAS are classified into somewhere in the introduction around line 74.

Response: That is a good suggestion. The sentence has been modified in the revised manuscript **(Page 5, Lines 100-102)**.

Original: Here, we report the extensive capture of 107 PFASs (Fig. 1, ranging from C2 to C16) from polymer fluoropolymer production ...

Modified: Here, we report the extensive capture of 107 PFAS (Fig. 1, C2 to C16, including 82 carboxylic acids and 25 sulfonic acids), comprising 60 long-chain PFAS (C_≥7) ad 47 short-chain PFAS, from polymer fluoropolymer production ...

Results

Comment 5. Line 87: I disagree that 107 is same as “hundreds”. Hundreds would imply at least two one-hundreds, and the authors did not measure 200 PFAS.

Response: Your comments are appreciated. We have changed “hundreds of” to “107” in the revised manuscript **(Page 6, Lines 115-116)**.

Original: **Capturing of hundreds of PFAS in Complex Fluorochemical Wastewaters with Zn-based EC and Conventional Adsorbents: Efficiency and Broad-Spectrum**

Modified: **Capturing 107 PFAS in Complex Fluorochemical Wastewaters with Zn-based EC and Conventional Adsorbents: Efficiency and Broad-Spectrum**

Tang C.; Liang Y.; Wang K.; Liao J.; Zeng Y.; Luo X.; Peng X.; Mai B.; Huang Q.; Lin H. Comprehensive characterization of per- and polyfluoroalkyl substances in wastewater by liquid chromatography-mass spectrometry and screening algorithms. NPJ Clean Water 2023, 6, 6.

Comment 6. In general, the figures are spectacular. Very well done!

Response: Your praise for our work is much appreciated.

Comment 7. The mechanism discussion on line 218 is very interesting.

Response: Your praise for our work is much appreciated.

Comment 8. Line 240 users “PFASs” and “PFAS”, both should be PFAS.

Response: Revised as suggested. All “PFASs” has been changed to “PFAS” in the revised manuscript.

Comment 8. Line 305: Do authors have a reference to a toxicity study that shows iodinated PFAS are indeed less toxic or is this speculation?

Response: Your comments are appreciated. In our previous study, we calculated some physicochemical properties, environmental behaviors, and toxicities of iodinated-PFAS (I-PFAS) and their parent PFAS using computational toxicology and environmental simulation software (EPI suite 4.11, ECOSAR 2.0 and U.S. EPA, Toxicity Estimation Software Tool, Version 5.1.1). The results are shown in the figure below, and the details can be found in this paper *Environ. Sci. Technol.* 2023, 57, 1378-1390. The biodegradation data indicates that the I-PFAS are less persistent in the sewage treatment plant than their parent PFAS. The fish chronic and acute toxicities suggest that the I-PFAS may be slightly toxic. It should be noted that the software may lack the necessary training sets for fluorinated substances, which could result in inaccurate predictions. For example, as shown in Figure 3a in the manuscript, the short-chain I-PFAS (C2~C6) are preferentially adsorbed with K_d values greater than 3, suggesting that all of them should be highly hydrophobic with $\log K_{ow}$ value > 4 . However, their $\log K_{ow}$ values predicted by the EPI Suite software for all of them are less than 4. It is therefore unclear whether I-PFAS is actually less toxic.

Similarly, the software also suggests that Cl-PFAS are more toxic than their parent PFAS. However, a recent study by Jin et al. (2023) showed that the substitution of F with Cl significantly improved the biodegradability and reduced the toxicity of PFAS. Replacing F with I could further enhance this effect due to the larger radius of the iodine atom. Therefore, we prefer to believe that I-PFAS is less toxic.

The experimental results of electro-oxidative treatment of the fluorochemical wastewater demonstrated that the degradation rate of PFAS with the same chain length followed: I-PFAA \gg Cl-PFAA $>$ ether-PFAA, H-PFAA and PFCA (Supplementary Fig. 15). Furthermore, it was also observed that the total I-PFAA concentrations in a sealed white PE plastic drum (25 L) stored at ambient temperature decreased by 52% over a 13-month period (*Environ. Sci. Technol.* 2023, 57, 1378-1390), suggesting that I-PFAA is susceptible to degradation under natural conditions. Therefore, it can be confirmed that I-PFAS are more easily degraded than their parents.

1. Tang C.; Zhu Y.; Liang Y.; Zeng Y.; Peng X.; Mai B.; Xu J.; Huang Q.; Lin H. First discovery of iodinated polyfluoroalkyl acid by nontarget mass-spectrometric analysis and iodine-specific screening algorithm. *Environ. Sci. Technol.* 2023, 57, 1378-1390.
2. Jin B.; Liu H.; Che S.; Gao J.; Yu Y.; Liu J.; Men Y. Substantial defluorination of polychlorofluorocarboxylic acids triggered by anaerobic microbial hydrolytic dechlorination. *Nature Water* 2023, 1, 451-461.

Figure. Predicted environmental behaviors and toxicities of I-PFAAs and their I→F exchanged PFAA analogues. The data were calculated with U.S. EPA, Estimation Program Interface (EPI) Suite, Version 4.11 (<https://www.epa.gov/tsc-screening-tools/download-epi-suite-estimation-program-interface-v411>), ECOSAR, Version 2.0 (<https://www.epa.gov/tsc-screening-tools/ecological-structure-activity-relationships-ecosar-predictive-model>), and Toxicity Estimation Software Tool (TEST), Version 5.1.1 (<https://www.epa.gov/chemical-research/toxicity-estimation-software-tool-test>). The compound pair Nos. were listed in Table S11. AOPWIN: a model calculating the atmospheric degradation half-life of chemical compounds; Fish ChV: chronic toxicities of chemicals to fish; BCF: bioconcentration factor; LC₅₀: median lethal concentration.

Table. Calculated toxicities of I-PFAAs and their corresponding I→F exchanged PFAA analogues (calculated by TEST 5.1.1).

No.	Compound/ Molecular formula	Fathead minnow LC ₅₀ (96 h, mg/L)	Daphnia magna LC ₅₀ (48 h, mg/L)	T. pyriformis IGC ₅₀ (48 h, mg/L)	Oral rat LD ₅₀ (mg/kg)	Developmental toxicity	Mutagenicity
	I-PFAAs						
1	C ₄ H ₂ O ₂ F ₂ I	58.60	0.73	NA	NA	Toxicant	NA
2	C ₂ H ₃ O ₃ F ₂ IS	162.84	NA	NA	NA	Toxicant	NA

No.	Compound/ Molecular formula	Fathead minnow LC ₅₀ (96 h, mg/L)	Daphnia magna LC ₅₀ (48 h, mg/L)	T. pyriformis IGC ₅₀ (48 h, mg/L)	Oral rat LD ₅₀ (mg/kg)	Developmental toxicity	Mutagenicity
3	C ₂ H ₃ O ₄ F ₂ IS	25.69	NA	NA	NA	Non-toxicant	NA
4	C ₄ H ₃ O ₃ F ₄ IS	57.10	NA	NA	NA	Toxicant	NA
5	C ₅ HO ₂ F ₄ I	33.91	0.33	NA	NA	Toxicant	Negative
6	C ₆ H ₂ O ₂ F ₇ I	NA	0.13	NA	NA	Toxicant	Negative
7	C ₆ H ₃ O ₂ F ₆ I	21.46	0.16	NA	NA	Toxicant	Negative
8	C ₄ H ₃ O ₄ F ₄ IS	75.75	NA	NA	NA	Toxicant	NA
9	C ₆ H ₇ O ₃ F ₆ IS	14.07	NA	NA	NA	Toxicant	NA
10	C ₈ H ₇ O ₂ F ₈ I	3.24	3.21E-02	NA	NA	Toxicant	NA
11	C ₆ H ₇ O ₄ F ₆ IS	37.39	NA	NA	NA	Toxicant	NA
12	C ₈ H ₉ O ₃ F ₈ IS	6.58	NA	NA	NA	NA	NA
13	C ₁₀ H ₉ O ₂ F ₁₀ I	1.33	1.71E-02	NA	NA	NA	NA
14	C ₈ H ₉ O ₄ F ₈ IS	3.41	NA	NA	NA	NA	NA
15	C ₆ H ₅ O ₃ F ₈ IS	4.84	NA	NA	NA	NA	NA
16	C ₇ HO ₂ F ₁₀ I	NA	7.79E-02	NA	NA	NA	NA
17	C ₁₀ H ₁₁ O ₄ F ₁₀ IS	2.94	NA	NA	NA	NA	NA
18	C ₂ HO ₃ F ₄ IS	51.93	NA	NA	NA	Toxicant	NA
	I→F PFAAs						
1	C ₄ HO ₂ F ₃	176.69	149.17	NA	NA	Toxicant	NA
2	C ₃ H ₃ O ₃ F ₃ S	107.94	NA	NA	NA	Toxicant	NA
3	C ₂ H ₃ O ₄ F ₃ S	95.31	NA	NA	NA	Toxicant	NA
4	C ₄ H ₃ O ₃ F ₅ S	69.53	NA	NA	828.31	Toxicant	Negative
5	C ₃ HO ₂ F ₅	107.81	43.47	NA	NA	Toxicant	Negative
6	C ₆ H ₂ O ₂ F ₈	NA	64.36	NA	NA	NA	Negative
7	C ₆ H ₃ O ₂ F ₇	38.44	64.98	NA	2999.72	Toxicant	Negative
8	C ₄ H ₄ O ₄ F ₄ IS	94.93	NA	NA	898.71	Non-toxicant	Positive
9	C ₆ H ₇ O ₃ F ₇ S	13.67	NA	NA	909.76	Toxicant	Negative
10	C ₈ H ₇ O ₂ F ₉	10.65	47.93	NA	1606.27	NA	NA
11	C ₆ H ₇ O ₄ F ₇ S	36.68	NA	NA	661.12	Toxicant	Negative
12	C ₈ H ₉ O ₃ F ₉ S	6.98	NA	NA	308.43	NA	NA
13	C ₁₀ H ₉ O ₂ F ₁₁	2.38	18.34	NA	1014.78	NA	NA
14	C ₈ H ₉ O ₄ F ₉ S	3.95	NA	NA	255.90	NA	NA
15	C ₆ H ₃ O ₃ F ₉ S	16.15	NA	NA	803.32	NA	NA
16	C ₇ HO ₂ F ₁₁	NA	22.18	NA	329.68	NA	NA
17	C ₁₀ H ₁₁ O ₄ F ₁₁ S	3.22	NA	NA	NA	NA	NA
18	C ₂ HO ₃ F ₅ S	57.53	NA	NA	601.42	Toxicant	Negative

Note, LC₅₀: median lethal concentration; IGC₅₀: median inhibition growth concentration; LD₅₀: median lethal dose; NA: not available; U.S. EPA, Toxicity Estimation Software Tool (TEST), Version 5.1.1 (<https://www.epa.gov/chemical-research/toxicity-estimation-software-tool-test>).

Considering your comment, we have deleted “less toxic” in the revised manuscript (**Pages 16-17, Lines 366-375**).

Original: Therefore, it may be possible to design alternative iodinated PFAS that are readily degradable and less toxic.

Modified: The experimental results of electro-oxidative treatment of the fluorochemical wastewater demonstrated that the degradation rate of PFAS with the same chain length followed: I-PFAA >> Cl-PFAA > ether-PFAA, H-PFAA and PFCA (Supplementary Fig. 15). Additionally, it was observed that the total I-PFAA concentrations in a sealed white polyethylene plastic drum stored at ambient temperature decreased by 52% over a 13-month period⁴⁹, suggesting that I-PFAA is susceptible to degradation under natural conditions. The presence of I-PFAA in the fluorochemical wastewater may be attributed to

the unintentional by-products of iodine transfer polymerization/copolymerization processes during the synthesis of fluoropolymers, where iodofluoroalkanes are used as chain transfer agents⁴⁹. However, their specialized properties provide new insights into the design of alternative iodinated PFAS that are readily degradable.

Supplementary Fig.15 The electrooxidation degradation kinetics of PFAS in the fluorochemical wastewater.

Comment 10. Implications: do authors think this treatment strategy could work on more dilute streams such as municipal wastewater or drinking water? The opening line in the paper is about drinking water standards. Could this work for a drinking water utility, or is it more applicable to the higher mg/L levels tested?

Response: Your comments are appreciated. From the point of view of adsorption mechanism, we believe that Zn-based EC is more efficacious for the treatment of complex and highly concentrated PFAS wastewater, such as industrial wastewater and AFFFs wastewater. Further, Zn-based EC has been employed in groundwater remediation (~250,000 gallons) at a US Air Force base, with an average removal rate of approximately 72% for concentrations ranging from hundreds of ppt to several ppb of PFOA (585 ppt) and PFOS (4800~8400 ppt). It can therefore be concluded that Zn-based EC is also suitable for low concentration PFAS treatment, but its efficacy may be significantly lower than that of the treatment of high concentration of PFAS effluents.

Methods:

Comment 11. Somewhere in the first section can authors note the total concentration of PFAS in the industrial wastewater?

Response: Thanks for your good suggestion, we have added the total concentration of PFAS in the revised manuscript (**Page 23, Lines 532-534**).

Original: Wastewater samples were taken from the reverse osmosis (RO) concentrate...

Modified: Wastewater samples with a concentration of Σ PFAS of 36 mg·L⁻¹ were taken from the reverse osmosis (RO) concentrate...

Reviewer #4

This article presents an innovative treatment strategy that achieves efficient and broad-spectrum capture of 107 PFASs by combining Zn-based electrocoagulation (EC) with anion-exchange resin (AER) beds. The study demonstrates that the "zero-carbon" adsorbent, zinc hydroxide flocs generated by Zn-based EC, significantly enhances adsorption capacity through a semi-micellar adsorption mechanism similar to mineral flotation. The technical-economic analysis and life-cycle environmental impact assessment reveal that this method not only reduces costs by an order of magnitude but also decreases the carbon footprint by 70%. Additionally, the study observes significantly improved adsorption selectivity for iodinated PFAS, providing new insights into designing environmentally friendly fluorochemicals. Overall, this research offers a valuable solution to the PFAS challenge. However, the following issues need to be appropriately addressed.

Response: We truly appreciate the very constructive comments that help to clarify and improve our manuscript. We have carefully revised the manuscript according to the comments, as detailed below. Thanks once again for your positive comment.

Comment 1. The abstract mentions that the PFAS removed by Zn-based electrocoagulation achieved the highest adsorption capacities among all reported adsorbents. On what basis was this result generated? The authors need to clarify whether this result applies to all PFAS or specific PFAS.

Response: Your comments are appreciated. We note that this statement may mislead readers. In the revised version (**Pages 2, Lines 24-27**), a more rigorous presentation has been provided.

Original: The "zero-carbon" adsorbent, zinc hydroxide flocs, generated in-situ by Zn-based EC bulk removes PFAS with $\log K_{ow} > 4$ through a semi-micellar adsorption mechanism similar to mineral flotation, resulting in the highest adsorption capacities all reported adsorbents.

Modified: The "zero-carbon" adsorbent, zinc hydroxide flocs, generated in-situ by Zn-based EC bulk removes hydrophobic PFAS with $\log K_{ow} > 4$ through a semi-micellar adsorption mechanism similar to mineral flotation, resulting in adsorption capacities at the optimal level of all reported adsorbents.

Comment 2. In the first paragraph of the introduction, it is recommended to specify the hazards of PFAS.

Response: Thanks for your good suggestion, we have added the relevant content in the revised manuscript (**Pages 3, Lines 38-39**).

Added: Regrettably, PFAS are linked to severe health issues, including cancer and reproductive toxicity.

Comment 3. In introduction part, the authors need to summarize the current PFASs removal method in FIP. How about the removal efficiencies of these 107 PFASs? What is the biggest challenge for the current used method?

Response: Thanks for your comments. Fluorochemical wastewater contains hundreds of PFAS with extreme structural diversity. However, there are very limited studies examining the removal of PFAS from fluorochemical wastewater. To date, only one recent study has reported on the removal of PFAS in full-scale fluorochemical wastewater treatment processes. Zhang et al. (2024) reported that current wastewater treatment processes are ineffective in removing PFAS discharged by fluorochemical manufacturers. The study revealed that 48 PFAS (ranging from 14.7 to 5200 $\mu\text{g}\cdot\text{L}^{-1}$) from the effluents of 10 FIPs in China were discharged into surface water. Specifically, the mass flows of PFAS increased by at most 233% after the activated sludge system but decreased by only 0–13% after the AC filtration. Despite the great success of traditional adsorbents such as AC/AER in applications for the removal of PFAS from relatively clean waters with minimal low background matrix such as groundwater and drinking water, it is unclear whether they can effectively remove hundreds of PFAS from the complex fluorochemical wastewater.

Considering the comment, we have added the relevant content in the revised manuscript (**Page 4, Lines 70-75**).

Added: In a recent study, Zhang and colleagues³² examined the mass flows of 48 PFAS, ranging from 14.7 to 5200 $\mu\text{g}\cdot\text{L}^{-1}$, in the effluents of 10 FIPs in China and showed that the current wastewater treatment processes are ineffective at removing these PFAS from discharges by fluorochemical manufacturers, resulting in significant PFAS contamination of the receiving waters. Specifically, the mass flows of PFAS increased from -20% to 233% with activated sludge system but decreased by only 0–13% after the AC filtration.

Zhang et al. Emerging and legacy per- and polyfluoroalkyl substances (PFAS) in fluorochemical wastewater along full-scale treatment processes: source, fate, and ecological risk. J. Hazard. Mater. 2024, 465, 133270.

Comment 4. Lines 53-55 suggest providing examples of chemical and biological processes.

Response: Thanks for your comments, we have added the relevant content in the revised manuscript (**Page 3, Lines 56-58**).

Added: Microbial treatment, such as anaerobic biochemical process, has shown potential for addressing certain polyfluorinated chemicals. For instance, a recent study demonstrated the capability of this process is capable of defluorinating chlorinated polyfluorocarboxylic acids (Cl-PFCA)¹⁹.

Comment 5. Line 74: why do the authors choose these 107 PFASs, are cation and zwitter PFASs included?

Response: Your comments are appreciated. A total of 175 PFAS (>350 formulas) were identified in the fluorochemical effluents (*Tang & Lin et al. NPJ Clean Water, 2023, 6, 6*), and the removal of all PFAS was tracked. However, it is challenging to ascertain the accuracy of results for some PFAS because their concentrations are too low. Consequently, the data for 107 PFAS were retained for further analysis.

Most of the PFAS (>95%) present in the fluorochemical effluents are anionic. Nevertheless, several nitrogen-containing cationic or zwitterionic PFAS, such as $C_7H_4O_4N_2F_9S$, $C_8H_4O_5N_2F_{11}S$ and $C_9H_4O_4N_2F_{11}S$, were identified, but their concentrations were very low. Therefore, these PFAS were not investigated in this study.

Tang C.; Liang Y.; Wang K.; Liao J.; Zeng Y.; Luo X.; Peng X.; Mai B.; Huang Q.; Lin H. Comprehensive characterization of per- and polyfluoroalkyl substances in wastewater by liquid chromatography-mass spectrometry and screening algorithms. NPJ Clean Water 2023, 6, 6.

Comment 6. In lines 74-76, the description of the Zn-based electrocoagulation process is brief. It is recommended to elaborate on its advantages compared to traditional methods.

Response: Your comments are appreciated. We agree with you and the relevant contents have been added in the revised manuscript (**Pages 4-5, Lines 80-99**).

Added: One of the defining characteristics of PFAS is its exceptional surface activity, which allows it to be adsorbed in multiple layers through semi-micellar or micellar adsorption. This significantly boosts its adsorption capacity. An example of this is mineral flotation, where trapping agents are adsorbed onto hydrophobic mineral particles through semi-micellar adsorption, enabling selective mineral capture. Inspired by this, we propose a reverse mineral flotation process using hydrophobic "mineral particles" as adsorbents to selectively extract PFAS from water via semi-micellar adsorption. We found that zinc hydroxide flocs, formed in situ by electrolysis, exhibit properties of hydrophobic "mineral particles," thus enabling them to rapidly adsorb hydrophobic long-chain PFAS from water with extremely high adsorption capacities^{34, 35}. For example, these zinc hydroxide flocs achieved an equilibrium adsorbed amount (q_e) of up to 5.74/7.69 mmol g⁻¹ (Zn) for PFOA/PFOS within minutes at an initial concentration of 0.5 mM³⁴. We therefore hypothesize that Zn-based electrocoagulation (EC) can effectively capture long-chain PFAS in the fluorochemical wastewater, while the existing AC and/or AER adsorption devices can then remove the remaining low concentrations of short-chain PFAS. This approach is founded on two primary observations: 1) Despite the presence of hundreds of PFAS types in fluorochemical wastewater, long-chain PFAS, particularly PFOA, dominate the total PFAS concentration Zn-based EC also removes dissolved organic matter (DOM) and traps colloidal particles from

wastewater, significantly mitigating the impact of these competing constituents on subsequent adsorption processes. This strategy represents the first attempt to achieve broad-spectrum removal of hundreds of PFAS from real fluorochemical wastewaters, potentially offering new options for tackling PFAS in complex industrial waste streams.

Comment 7. The concept of Zn-based electrocoagulation is not proposed for the first time. Please summarize previous reports on this topic in the introduction.

Response: We agree with you, and the relevant contents have been added in the revised manuscript (see Comment 6).

Comment 8. Line 123: Why was the Langmuir model used to study adsorption capacity instead of other models?

Response: Your comments are appreciated. The Langmuir model was able to fit the adsorption data well, so we did not consider other models. The objective here is just to illustrate that the AC and AER used in this study have high theoretical PFOA adsorption capacities, as reported in the literature.

Comment 9. Line 192: Please explain the reason for the selection of these 6 PFCAs? no other PFASs?

Response: Your comments are appreciated. The design of this experiment was based on the following reasons: (1) the PFAS to be tested must have an accurate $\log K_{ow}$ value (not from software calculations); and (2) due to the control of PFAS chemicals in China, it is no longer possible to purchase PFAS such as PFOS/PFOA, from regular sources (e.g. Sigma-Aldrich), and these PFCAs are the only ones available in our laboratory. In light of your comments, three additional PFAS (sourced from other laboratories), i.e., PFOS and GenX, have been included in the revised manuscript (Pages 11-12, Lines 230-256).

Modified: To elucidate the selective adsorption mechanism in the Zn-based EC process, we further investigated the adsorption kinetics of eight PFAS (for which reliable $\log K_{ow}$ values were available) with varying chain-lengths and structural characteristics in a simulated solution. Zinc hydroxide flocs presented extremely rapid adsorption of all eight PFAS with equilibrium time (t_{eq}) less than of 2 min (Supplementary Fig. 9), whereas AC and AER widely used in current applications had t_{eq} of tens of hours or more (Supplementary Tables 4-6). The observed maximum adsorption amount (q_m , mmol PFAS \cdot g $^{-1}$ zinc hydroxide flocs) was monotonically correlated with their hydrophobicity and chain-lengths (Fig. 3b, Supplementary Table 3 and Supplementary Fig. 9), suggesting a pivotal role of hydrophobic

interaction. The weakly hydrophobic PFBA (C4, $\log K_{ow} = 2.14$) had a q_m value of $< 0.1 \text{ mmol}\cdot\text{g}^{-1}$, the moderately hydrophobic PFHxA (C6, $\log K_{ow} = 3.48$) had an elevated q_m of $0.6 \pm 0.3 \text{ mmol}\cdot\text{g}^{-1}$, and the highly hydrophobic PFDA (C10, $\log K_{ow} = 6.15$) achieved an ultra-high q_m of $>23 \text{ mmol}\cdot\text{g}^{-1}$ ($>10 \text{ g}\cdot\text{g}^{-1}$). For the GenX (C6, $\log K_{ow} = 3.36$) with less hydrophobicity, the achieved q_m was $1.6 \pm 0.7 \text{ mmol}\cdot\text{g}^{-1}$, lower than that of AER but also several times higher than that of the AC and other reported adsorbents (Fig. 3c). However, the most discussed long-chain PFAS, i.e., PFOA (C8, $\log K_{ow}=4.81$) and PFOS (C8, $\log K_{ow}=4.88$), their q_m values were estimated to be as high as $6.4 \pm 0.4 \text{ mmol}\cdot\text{g}^{-1}$ ($2.6 \text{ g}\cdot\text{g}^{-1}$) and $7.5 \pm 0.04 \text{ mmol}\cdot\text{g}^{-1}$ ($3.8 \text{ g}\cdot\text{g}^{-1}$), respectively. To the best of our knowledge, these achieved q_m for hydrophobic PFAS such as PFOA and PFOS are the highest of all values reported in the literature, which are over an order-of-magnitude higher than the theoretical maximum adsorption capacity derived from the adsorption model fitting that of the data for the benchmark AC and several times higher than that of the AER (Fig. 3c and Supplementary Tables 4-6). Furthermore, the dynamic adsorption capacity (q_{dyn} , $\text{mmol}\cdot\text{g}^{-1}\cdot\text{h}^{-1}$) was more than 1~4 orders of magnitude higher than that of the benchmark AC and AER and other state-of-art adsorbents reported in the literature (Fig. 3c). The q_{dyn} ($\text{mmol}\cdot\text{g}^{-1}\cdot\text{h}^{-1}$) was calculated according to the equation, $q_{dyn} = q_m/t_{eq}$, in which the t_{eq} represents the adsorption equilibration time⁴³. The kinetics of PFAS adsorption by conventional porous adsorbents are primarily constrained by the diffusion process within the particles. In contrast, the adsorption of PFAS on the zinc hydroxide flocs predominantly occurs through a particle-surface adsorption process, as depicted in Figure 3f, resulting in significantly faster adsorption rates.

Figure 3. (b) The achieved maximum adsorption amount (q_m , $\text{mmol}\cdot\text{g}^{-1}$) of 8 PFAS (C4~C10) in simulated solution by Zn-based EC vs. their $\log K_{ow}$ values, and (c) comparison of the q_m and q_{dyn} values for PFOA, PFOS and GenX by Zn-based EC with various adsorbents in reported literature (listed in Supplementary Tables 4-6).

Comment 10. In line 276, it is stated: "Notably, the behavior of the novel Ix-PFAAs was markedly different." It is recommended to further explore the reasons behind this phenomenon. Consider using theoretical calculations (e.g., DFT) or other methods to elucidate the mechanism.

Response: That is a good suggestion. In our previous study (*Environ. Sci. Technol.* 2023, 57, 1378-1390), we calculated some physicochemical properties, environmental behaviors, and toxicities of iodinated-PFAS (I-PFAS) and their parent PFAS using computational toxicology and environmental simulation software (EPI suite 4.11, ECOSAR 2.0 and U.S. EPA, Toxicity Estimation Software Tool, Version 5.1.1). It should be noted that the software may lack the necessary training sets for fluorinated substances, which could result in inaccurate predictions. For example, as shown in Figure 3a in the manuscript, the short-chain I-PFAS (C2~C6) are preferentially adsorbed with K_d values greater than 3, suggesting that all of them should be highly hydrophobic with $\log K_{ow}$ value > 4 . However, their $\log K_{ow}$ values predicted by the EPI Suite software for all of them are less than 4. The main results are shown in the figure below, and the detail could be found in else (*Environ. Sci. Technol.* 2023, 57, 1378-1390).

DFT calculations may yield more accurate results, and we will consider your suggestion in future studies.

Tang C.; Zhu Y.; Liang Y.; Zeng Y.; Peng X.; Mai B.; Xu J.; Huang Q.; Lin H. First discovery of iodinated polyfluoroalkyl acid by nontarget mass-spectrometric analysis and iodine-specific screening algorithm. *Environ. Sci. Technol.* 2023, 57, 1378-1390.

Figure. Predicted environmental behaviors and toxicities of I-PFAAs and their I→F exchanged PFAA analogues. The data were calculated with U.S. EPA, Estimation Program Interface (EPI) Suite, Version 4.11 (<https://www.epa.gov/tsca-screening-tools/download-epi-suite-estimation-program-interface-v411>), ECOSAR, Version 2.0 (<https://www.epa.gov/tsca-screening-tools/ecological-structure-activity-relationships-ecosar-predictive-model>), and Toxicity Estimation Software Tool (TEST), Version 5.1.1 (<https://www.epa.gov/chemical-research/toxicity-estimation-software-tool-test>). The compound pair Nos. were listed in Table S11. AOPWIN: a model calculating the atmospheric degradation half-life of chemical compounds; Fish ChV: chronic toxicities of chemicals to fish; BCF: bioconcentration factor; LC₅₀: median lethal concentration.

Comment 11. Lines 293-295 mention that iodine substitution significantly affects the chemical properties of PFAS, while chlorine substitution does not. It is recommended to expand this discussion and compare the environmental behavior (e.g., degradability) of iodinated and chlorinated PFAS in detail.

Response: Your comments are appreciated. We investigated electrooxidation for the treatment of fluorochemical wastewater and showed that the degradation rate of I-PFAS was significantly faster than that of Cl-PFAS (see figure below). In light of your comment, the results have been included in the revised manuscript (**Pages 16-17, Lines 366-375**).

Modified: The experimental results of electro-oxidative treatment of the fluorochemical wastewater demonstrated that the degradation rate of PFAS with the same chain length followed: I-PFAA >> Cl-PFAA > ether-PFAA, H-PFAA and PFCA (Supplementary Fig. 15). Additionally, it was observed that the total I-PFAA concentrations in a sealed white polyethylene plastic drum stored at ambient temperature decreased by 52% over a 13-month period⁴⁹, suggesting that I-PFAA is susceptible to degradation under natural conditions. The presence of I-PFAA in the fluorochemical wastewater may be attributed to the unintentional by-products of iodine transfer polymerization/copolymerization processes during the synthesis of fluoropolymers, where iodofluoroalkanes are used as chain transfer agents⁴⁹. However, their specialized properties provide new insights into the design of alternative iodinated PFAS that are readily degradable.

Supplementary Fig.15 The electrooxidation degradation kinetics of PFAS in the fluorochemical wastewater.

Comment 12. Is the software-predicted $\log K_{ow}$ of PFAS reasonable?

Response: Your comments are appreciated. Although software is widely used in the literature to predict the $\log K_{ow}$ of PFAS, our experimental findings suggest that the resulting predictions for the PFAS with novel structural (e.g., Ether-PFAS and I-PFAS) may be unreliable due to the paucity of data for training.

Comment 13. 417: The degradation results of iodinated PFAS under natural conditions should be presented in detail.

Response: Your comments are appreciated. The data now have been included in the revised manuscript (**Pages 16-17, Lines 368-371**).

Added: Additionally, it was observed that the total I-PFAA concentrations in a sealed white polyethylene plastic drum stored at ambient temperature decreased by 52% over a 13-month period⁴⁹, suggesting that I-PFAA is susceptible to degradation under natural conditions.

Comment 14. Line 428: Is PFOS/PFOA linear, branched, or an isomer?

Response: The mass spectrometry results (see figures below) indicated that the chemicals were predominantly linear PFOA/PFOS. Both L-PFOA and B-PFOA are present in the fluorochemical effluents, with L-PFOA accounting for approximately 78.5%.

Figure. The TIC of the mass spectra for the PFOA/PFOS chemicals (up) and in the fluorochemical wastewater (down).

Comment 15. Line 431: Please double check the table and figure across the whole manuscript. In this sentence, Table S7 should be replaced by Table S8. There are other similar mistakes in the MS.

Response: Thanks very much for pointing out that the table numbers were incorrectly labelled. We have double-checked the whole manuscript and corrected them.

Comment 16. Line 443-444: Why is the isotherm only for PFOA? Besides, the concentrations were also quite high?"

Response: Your comments are appreciated. To ascertain the maximum adsorption capacity of AC and AER, high-concentration PFOA adsorption experiments are necessary.

Considering your comment, the theoretical adsorption capacities of AC/AER for GenX have been included in the revised manuscript.

Added:

Supplementary Fig. 2 Sorption isotherms of PFOA and GenX on the PFA694E and AC at 25 °C and modeling using the Langmuir equations.

Comment 17. Line 450: Is there any possibility that a certain amount of PFASs were adsorbed on the surface of the glass? Did you check it?

Response: Your concerns are reasonable. Previous studies have demonstrated that glass may adsorb trace amounts of PFAS. However, the concentration of PFAS in this study was up to mg/L, which resulted in an insignificant effect.

Comment 18. Line 455: What do you mean “of a rate a mL·min⁻¹”

Response: That is a mistake, we have revised to “of a rate 2 mL·min⁻¹” in the revised manuscript.

Comment 19. What was the unadjusted pH range during the EC treatment process?

Response: We did not control the pH during the treatment process. The final pH of the treated fluorochemical wastewater was around 8~9.

Comment 20. There are errors in the shapes of Figures 2b and 2c. Please check carefully.

Response: Your comments are appreciated. All figures and tables have been double-checked and corrected.

Modified:

Comment 21. Considering the manuscript, it is recommended that the authors expand on the practical application prospects and potential challenges of combining Zn-based electrochemical and adsorption processes.

Response: That is a good suggestion, the relevant contents have been added in the revised manuscript (**Pages 21-22, Lines 490-513**).

Added: Electrocoagulation is a mature and simple pretreatment process that has been widely used in water treatment for decades. The Zn-based electrocoagulation (EC) process is notably versatile, allowing for integration with various other techniques to remove PFAS from complex wastewater streams. Its most promising application involves coupling with existing conventional adsorption processes and membrane separation technologies. This not only efficiently removes hydrophobic PFAS from wastewater but also reduces competing constituents in tandem adsorption processes or mitigates membrane fouling. Because zinc hydroxide flocs are readily soluble in acidic or alkaline solutions, the Zn-based EC process therefore could also be employed as a pre-enrichment approach for PFAS. This approach can be combined with various destruction technologies, such as advanced redox processes and pyrolysis, to achieve cost-effective results. For example, aqueous film-forming foams contain substantial PFAS; however, the efficacy of direct treatment by advanced redox processes is limited due to the coexistence of numerous additional components, including sodium dodecyl sulfate (SDS), a hydrocarbon surfactant, and diethylene glycol butyl ether (DGBE), an organic solvent. Given the poor hydrophobicity of SDS ($\log K_{ow}=1.69$) and DGBE ($\log K_{ow}=0.29$), our preliminary results indicate that the Zn-based EC can achieve selective enrichment of PFAS from aqueous film-forming foams, which can then be effectively degraded by advanced redox processes. It is important to note that zinc is a toxic heavy metal element. The safe disposal and utilization of zinc hydroxide floc sludge are vital considerations for the potential application of the Zn-based EC process. Pyrolysis, including incineration and low-temperature alkaline hydrothermal processes, may be applicable to recover Zn. Additionally, our studies show that zinc hydroxide flocs generated in a simulated solution with NaCl as the supporting electrolyte exhibited superior

PFAS adsorption capabilities. Therefore, a comprehensive investigation into the hydrolysis behavior of Zn²⁺ and the identification of the key hydrolysis products that are responsible for the adsorption of PFAS is warranted, which is crucial for optimizing the Zn-based EC process.

Reviewer #5

This manuscript describes a comprehensive study of PFAS treatment comparing established and novel approaches. The study advances the current options available for PFAS treatment for industrial waste streams. The authors are encouraged to consider the following comments in a revised version of the manuscript.

Response: We truly appreciate your very constructive comments that help to clarify and improve our manuscript. We have carefully revised the manuscript according to the comments, as detailed below.

Comment 1. For the results in Figure 2, the authors should provide more discussion on the performance of AC and AER given that adsorption is considered the best available treatment for PFAS. For instance, was the relatively low removal of PFAS by AC and AER due to high concentration of PFAS, diverse PFAS structures, or high concentration of TOC or inorganic chemicals?

Response: Your comments are appreciated. Further analysis of the AER revealed that co-existing ions represent a significant factor influencing the adsorption capacity of the AER. Considering your comment, the relevant contents have been included in the revised manuscript (**Pages 7-8, Lines 155-164**).

Added: It was noted that the quantity of Cl^- ($1416.4 \pm 246.3 \mu\text{mol}\cdot\text{g}^{-1}$) exchanged from the PFA694E significantly surpassed the amount of PFOA adsorbed ($46.4 \pm 2.6 \mu\text{mol}\cdot\text{g}^{-1}$) (Supplementary Fig. 7a). This observation suggests that the elevated concentration of background inorganic anions in fluorochemical wastewater markedly competes with PFAS for adsorption sites. Wahman et al.³⁶ have previously reported that the exchange adsorption of PFAS by strong base AERs is affected by the presence of nitrite, sulfate, and bicarbonate. In this study, NO_3^- ($382.4 \pm 9.9 \mu\text{mol}\cdot\text{g}^{-1}$ adsorbed) and SO_4^{2-} ($389.4 \pm 40.7 \mu\text{mol}\cdot\text{g}^{-1}$ adsorbed) emerged as the predominant competing anions (Supplementary Fig. 7a), accounting for over 89% of the chloride ions exchanged out of PFA694E. Consequently, it is clear that effective pretreatment strategies are essential to reduce the interference of competing ions when utilizing AERs in the treatment of PFAS-laden industrial wastewaters.

Comment 2. Given that Zn EC effectively removed diverse PFAS, TOC, F^- , NO_3^- , and SO_4^{2-} , additional discussion on the mechanisms of Zn EC are needed to generalize the results to other waste streams and operating conditions. For example, the authors developed relationship between PFAS K_{ow} and K_d for Zn EC removal, but what were the molecular interactions and what accounts for the removal of TOC and inorganic chemicals? What is unique about Zn hydroxide flocs relative to Al or Fe hydroxide flocs? Also, what is unique about the adsorption of PFAS to Zn hydroxide floc relative to AC and AER? PFAS removal by AC is due to hydrophobic + van der Waals (vsW)

interactions, and AER is combination of electrostatic and hydrophobic + vdW interactions, so the authors should be able to say something more specific about the interactions between PFAS/Zn hydroxide floc relative to AC and AER.

Response: Your comments are appreciated. Firstly, we performed XRD analysis of the formed Zn hydroxide flocs, which demonstrated that SO_4^{2-} and NO_3^- participate in the Zn^{2+} hydrolysis reaction, leading to their removal as zinc hydroxide salts, i.e. zinc hydroxide nitrate ($\text{Zn}_5(\text{NO}_3)_2(\text{OH})_8$, ZnHN) and zinc hydroxide sulfate ($\text{Zn}_4\text{SO}_4(\text{OH})_6$, ZnHS), rather than being adsorbed. The relevant contents have been included in the revised manuscript (**Pages 9, Lines 191-194**).

Added: Results from XRD analysis (Supplementary Fig. 8) showed that SO_4^{2-} and NO_3^- participate in the Zn^{2+} hydrolysis reaction, leading to their removal as zinc hydroxide salts, i.e. zinc hydroxide nitrate ($\text{Zn}_5(\text{NO}_3)_2(\text{OH})_8$, ZnHN) and zinc hydroxide sulfate ($\text{Zn}_4\text{SO}_4(\text{OH})_6$, ZnHS).

Supplementary Fig. 8 XRD characterization of the formed zinc hydroxide flocs in the fluorochemical wastewater.

Secondly, the most significant distinction between Zn flocs and Al/Fe flocs is that Zn flocs exhibit hydrophobic properties. The commercially available iron flakes are galvanized iron (to prevent corrosion). In our studies of Fe-based EC, we observed that the flocs initially produced by electrolysis exhibited a tendency to float on the surface of the solution like an oil, displaying notable hydrophobic characteristics that differed significantly from those of conventional Al and Fe flocs. In light of the aforementioned comment, the contact angles of water on the three flocs were also measured. The results demonstrated that the Zn flocs exhibited a considerably larger contact angle ($\sim 90^\circ$), in comparison to the Al ($\sim 60^\circ$) and Fe ($\sim 33^\circ$) flocs. The relevant contents have been included in the revised manuscript (**Pages 12, Lines 269-272**).

Added: The characterization results indicated that the zinc hydroxide flocs have a contact angle of $\sim 90^\circ$ (Supplementary Fig. 11), which is significantly greater than the contact angles of the aluminum ($\sim 60^\circ$) and ferric hydroxide flocs ($\sim 33^\circ$).

Supplementary Fig. 11 The contact angles of different floccs.

In addition, an additional discussion has been included to elucidate the distinctions between the mechanisms of Zn-based EC and AC/AER in PFAS adsorption mechanisms in the revised manuscript (**Pages 13, Lines 280-285**).

Added: The adsorption of hydrophobic PFAS by adsorbents such as AC and AER also involves hydrophobic and van der Waals interactions, so hydrophobic PFAS could also form semi-micelles or micelles on the surfaces of these adsorbents. However, intraparticle diffusion determines their adsorption of PFAS, the presence of semi-micelles or micelles on the external surface would impede the diffusion of PFAS into the internal micropores of these adsorbents and may even lead to a decrease in overall adsorption capacity⁴⁶.

Comment 3. Following from comment 2, the authors describe unique F/Zn interaction, but this description is still general. What is unique about F/Zn interaction relative to other metals.

Response: Your comments are appreciated. As mentioned in comment 2, the most significant distinction between Zn floccs and traditional Al/Fe floccs is that Zn floccs exhibit hydrophobic properties. Considering your comment, we have characterized their contact angles (see Comment 2).

Comment 4. Following from comment 3, what is the expected nature of the PFAS/Zn hydroxide floc? Will substantial de-watering be required before the floc can be treated or disposed of?

Response: Your comments are appreciated. The final Zn-floc sludge (see photos below) obtained is similar to the conventional Fe/Al-floc sludge and can therefore be dewatered in similar manners, such as centrifugal dewatering.

Comment 5. The results for I-PFAA were interesting. Do the authors have additional insights on the production or use of I-PFAAs? Is the higher adsorption of I-PFAAs simply explained by more hydrophobic chemical or are other size or shape factors part of the explanation?

Response: Your comments are appreciated. The replacement of one or a few iodine atoms should not result in a significant alteration to the dimensions and conformation of the PFAS molecule. We therefore think that changes in physicochemical properties, such as markedly enhanced hydrophobicity, are responsible for the enhanced adsorption capacity of I-PFAS by Zn-based EC. This is, of course, mere speculation. To reveal the mystery of I-PFAS and to determine whether I-PFAS can meet the performance specifications of PFAS for fluorine chemical applications, we are currently synthesizing high-purity I-PFAS in collaboration with organic synthetic chemists for systematic experiments and characterizations.

I-PFAS may be unintentional byproducts in production of PFAS, as telechelic fluorinated dioxides, PFIs, and FTIs are reactants commonly used in telomerization processes for industrial synthesis of PFAS. I-PFAS may also be hydrolysis products of iodinated alkyl acyl fluorides and alkyl sulfonyl fluorides, which are intermediates in synthesis of PFASs. In addition, the presence of I-PFAS may be in connection with iodine transfer polymerization/copolymerization of vinylidene fluoride in synthesis of polyvinylidene fluoride (a very important fluoropolymer) and copolymers, in which iodoalkanes are often used as chain transfer agents and polyvinylidene fluoride products (oligomers) with an iodo end are generated.

It is very interesting that the physicochemical properties of I-PFAS diverge significantly from those of the parent PFAS, as well as from those of currently available alternatives, including Cl-PFAS, H-PFAS and ether-PFAS. The experimental results of electro-oxidative treatment of the fluorochemical wastewater demonstrated that the degradation rate of PFAS with the same chain length followed: I-PFAA \gg Cl-PFAA $>$ ether-PFAA, H-PFAA and PFCA. It was also observed that the total I-PFAA concentrations in a sealed white PE plastic drum (25 L) stored at ambient temperature decreased by 52% over a 13-month period, suggesting that I-PFAA is susceptible to degradation under natural conditions. Therefore, it may be possible to design alternative I-PFAS that are readily degradable.

Considering your comment, this contents have been included in the revised manuscript (**Pages 16-17, Lines 366-375**).

Modified: The experimental results of electro-oxidative treatment of the fluorochemical wastewater demonstrated that the degradation rate of PFAS with the same chain length followed: I-PFAA \gg Cl-PFAA $>$ ether-PFAA, H-PFAA and PFCA (Supplementary Fig. 15). Additionally, it was observed that the total I-PFAA concentrations in a sealed white polyethylene plastic drum stored at ambient temperature decreased by 52% over a 13-month period⁴⁹, suggesting that I-PFAA is susceptible to degradation under natural conditions. The presence of I-PFAA in the fluorochemical wastewater may be attributed to

the unintentional by-products of iodine transfer polymerization/copolymerization processes during the synthesis of fluoropolymers, where iodofluoroalkanes are used as chain transfer agents⁴⁹. However, their specialized properties provide new insights into the design of alternative iodinated PFAS that are readily degradable.

Supplementary Fig. 15 The electrooxidation degradation kinetics of PFAS in the fluorochemical wastewater.

Comment 6. The coupled system of Zn EC followed by AER makes sense, especially for high-strength industrial PFAS wastewater. Coagulation followed by AER is usually more effective than either process alone for TOC removal from natural water or leachate.

Response: We couldn't agree with you more. In the revised manuscript (**Page 5, Lines 90-99**), we have added more discussions in the introduction section to illustrate the treatment-train of the combination if the Zn-based EC with existing AC and/or AER devices.

Added: We therefore hypothesize that Zn-based electrocoagulation (EC) can effectively capture long-chain PFAS in the fluorochemical wastewater, while the existing AC and/or AER adsorption devices can then remove the remaining low concentrations of short-chain PFAS. This approach is founded on two primary observations: 1) Despite the presence of hundreds of PFAS types in fluorochemical wastewater, long-chain PFAS, particularly PFOA, dominate the total PFAS concentration Zn-based EC also removes dissolved organic matter (DOM) and traps colloidal particles from wastewater, significantly mitigating the impact of these competing constituents on subsequent adsorption processes. This strategy represents the first attempt to achieve broad-spectrum removal of hundreds of PFAS from real

fluorochemical wastewaters, potentially offering new options for tackling PFAS in complex industrial waste streams.

Comment 7. For economic and environmental impact, the authors should qualify/support the 10% water content for the PFAS/Zn floc. Given the proof of concept nature of the research, the authors should also highlight other assumptions that are likely to have a strong impact on the economic and environmental results. These assumptions (sensitive inputs) provide direction for future research.

Response: Your comments are appreciated. A moisture content of 10% was assumed for the Zn floc sludge, because a low moisture content is expected for incineration treatment. To avoid misunderstandings among readers, we have chosen to exclude the water content from our analysis in the revised manuscript (**Pages 19, Lines 439-442**), given that the incineration cost represents only a minor component of the overall treatment cost.

Original: As shown in Fig. 6a and Supplementary Table 5, the operational cost of the Zn-based EC process was approximately \$1.43 per m³ treated under a treatment time of 20 min, consisting of \$1.14 for zinc metal cost, \$0.1 for the electricity, and \$0.19 for the incineration of zinc hydroxide flocs (assuming 10% water content).

Modified: As shown in Fig. 6a and Supplementary Table 10, the operational cost of the Zn-based EC process was approximately \$1.43 per m³ treated under a treatment time of 20 min, consisting of \$1.14 for zinc metal cost, \$0.1 for the electricity, and \$0.17 for the incineration of zinc hydroxide flocs (Moisture content not considered).

The dosage of metal Zn determines the treatment cost of the Zn-based EC process. Consequently, reducing the quantity of Zn metal used is the key for reducing the treatment cost and environmental impact of the Zn-based EC process. In addition, resource utilization of the produced Zn floc sludge is also an avenue for further investigation. Considering your comment, we have included the relevant contents in the revised manuscript (**Pages 20, Lines 459-465**).

Added: We would also like to highlight that the amount of dissolved metallic zinc determines the cost and environmental impact of the Zn-based EC treatment. Therefore, developing methods to produce zinc hydroxide flocs with enhanced PFAS adsorption capabilities is an important area for future research. Additionally, the resourceful reuse of waste zinc hydroxide flocs should be considered. For example, recycling the incineration byproduct, ZnO, as a resource could significantly reduce the cost of Zn-based EC treatment to just \$0.08 per m³.

Comment 8. Do the authors expect the results for PFAS removal by Zn EC to be applicable to lower PFAS concentrations encountered in surface water or groundwater? Additional insights on factors that support or inhibit PFAS removal by Zn EC would benefit the PFAS research community in prioritizing future research.

Response: Your comments are appreciated. From the point of view of adsorption mechanism, we believe that Zn-based EC is more efficacious for the treatment of complex and highly concentrated PFAS wastewater, such as industrial wastewater and AFFFs wastewater. Nevertheless, Zn-based EC has been employed in groundwater remediation (~250,000 gallons) at a US Air Force base, with an average removal rate of approximately 72% for concentrations ranging from hundreds of ppt to several ppb of PFOA (585 ppt) and PFOS (4800~8400 ppt). It can therefore be concluded that Zn-based EC is also suitable for low concentration PFAS treatment, but its efficacy may be significantly lower than that of the treatment of high concentration of PFAS effluents.

In addition, the relevant contents related to further research have been included in the revised manuscript (**Pages 21-22, Lines 490-513**).

Added: Electrocoagulation is a mature and simple pretreatment process that has been widely used in water treatment for decades. The Zn-based electrocoagulation (EC) process is notably versatile, allowing for integration with various other techniques to remove PFAS from complex wastewater streams. Its most promising application involves coupling with existing conventional adsorption processes and membrane separation technologies. This not only efficiently removes hydrophobic PFAS from wastewater but also reduces competing constituents in tandem adsorption processes or mitigates membrane fouling. Because zinc hydroxide flocs are readily soluble in acidic or alkaline solutions, the Zn-based EC process therefore could also be employed as a pre-enrichment approach for PFAS. This approach can be combined with various destruction technologies, such as advanced redox processes and pyrolysis, to achieve cost-effective results. For example, aqueous film-forming foams contain substantial PFAS; however, the efficacy of direct treatment by advanced redox processes is limited due to the coexistence of numerous additional components, including sodium dodecyl sulfate (SDS), a hydrocarbon surfactant, and diethylene glycol butyl ether (DGBE), an organic solvent. Given the poor hydrophobicity of SDS ($\log K_{ow}=1.69$) and DGBE ($\log K_{ow}=0.29$), our preliminary results indicate that the Zn-based EC can achieve selective enrichment of PFAS from aqueous film-forming foams, which can then be effectively degraded by advanced redox processes. It is important to note that zinc is a toxic heavy metal element. The safe disposal and utilization of zinc hydroxide floc sludge are vital considerations for the potential application of the Zn-based EC process. Pyrolysis, including incineration and low-temperature alkaline hydrothermal processes, may be applicable to recover Zn. Additionally, our studies show that zinc hydroxide flocs generated in a simulated solution with NaCl as the supporting electrolyte exhibited superior PFAS adsorption capabilities. Therefore, a comprehensive

investigation into the hydrolysis behavior of Zn²⁺ and the identification of the key hydrolysis products that are responsible for the adsorption of PFAS is warranted, which is crucial for optimizing the Zn-based EC process.

Detailed Response to the Reviewers' Comments

Dec. 24, 2024

Journal: Nature Commun. (Manuscript ID: NCOMMS-24-25626A)

Title: "Treatment-train strategy realizes broad-spectrum capture of hundreds of per- and polyfluoroalkyl substances from fluorochemical wastewater"

Yiyang Liang^{†#}, Lihui Yang^{†#}, Caiming Tang^{†#}, Ying Yang[†], Shangtao Liang[§], Anqi Wang[†], Jiale Xu[†], Qingguo Huang[§], Hui Lin^{†*}

We sincerely thank all reviewers for their valuable comments and suggestions, which are certainly helpful in improving the quality of our work. We have carefully and systematically responded to all the points raised. We have also highlighted the revised text in **red** in the main text. Provided below are our detailed responses to each point.

Reviewer #2

The manuscript has been significantly improved by the revisions the authors made in response to reviewer comments. However, there are still some minor technical revisions which would be useful to make before publication.

Response: We truly appreciate the very constructive comments that help to clarify and improve our manuscript. We have carefully revised the manuscript according to the comments, as detailed below.

Comment 1. The justification for selecting the constant diffusivity approach for the RSSCT (revised lines 381-383) is still not clear in the text. I suggest deleting the text "The breakthrough of PFAS in RSSCT is dependent³⁸² on the particle size of the adsorbent, and hence the constant diffusion model is appropriate for designing RSSCT experiments" and replacing it with something like "A constant diffusivity design was chosen for the RSSCTs. A recent study has shown that, with proper interpretation, this approach can lead to useful approximations of pilot- or full-scale system performance.[50]"

Response: That's a good suggestion. Revised as your suggestions (**Page 17, Lines 381-383**).

Modified: A constant diffusivity design was chosen for the RSSCTs. A recent study⁵⁰ has shown that, with proper interpretation, this approach can lead to useful approximations of pilot- or full-scale system performance.

Comment 2. L563: Please state what kind of grinder was used to crush the resin. List any relevant operational settings.

Response: Your comments are appreciated. Considering your suggestions, the relevant contents have been included in the revised manuscript (**Page 24, Lines 564-569**).

Original: The PFA694E was crushed with a grinder and sieved to 0.282~0.25 mm, which allowed the column diameter/PFA694E particle diameter ratio to be >8~10 to eliminate wall or channel effects.

Modified: The PFA694E resins were crushed with a ball mill (Nanbei Instrument Limited, China) and sieved to 0.282~0.25 mm, which allowed the column diameter/PFA694E particle diameter ratio to be >8~10 to eliminate wall or channel effects. The volume of the ball-milling jar was 50 mL, and the diameter of the balls was 20 mm. The ball milling process was performed at a speed of 50 Hz with a 30 seconds pause after 30 seconds of milling, for a total of five cycles.

At last, Merry Christmas! As the festive season unfolds, my heart is filled with warm wishes for you.

Reviewer #4

The authors have made a commendable effort to address and revise the majority of the comments. However, there remain two minor points of clarification:

Response: We truly appreciate the very constructive comments that help to clarify and improve our manuscript. We have carefully revised the manuscript according to the comments, as detailed below. Thanks once again for your positive comment.

Comment 1. In response to Comment 1, the authors have not yet clearly specified whether the results pertain to all PFAS or are limited to certain specific compounds. While hydrophobic PFAS are mentioned in the authors' reply, the definition of hydrophobic versus hydrophilic PFAS is still unclear. Are all PFAS with a log $K_{ow} > 4$ considered hydrophobic?

Response: Your comments are appreciated. The K_{ow} value is an important parameter for the quantitative characterization of the hydrophobicity of organic compounds. Compounds with high K_{ow} values are considered to be highly hydrophobic, whereas hydrophilic compounds generally have a log K_{ow} not greater than 1. In light of your concerns, we have deleted the word “hydrophobic” in the revised version (**Pages 2, Lines 24-27**). We just use the log K_{ow} value of PFAS as a constraint.

Original: The “zero-carbon” adsorbent, zinc hydroxide flocs, generated in-situ by Zn-based EC bulk removes hydrophobic PFAS with $\log K_{ow} > 4$ through a semi-micellar adsorption mechanism similar to mineral flotation, resulting in adsorption capacities at the optimal level of all reported adsorbents.

Modified: The “zero-carbon” adsorbent, zinc hydroxide flocs, generated in-situ by Zn-based EC bulk removes PFAS with $\log K_{ow} > 4$ through a semi-micellar adsorption mechanism similar to mineral flotation, resulting in adsorption capacities at the optimal level of all reported adsorbents.

Comment 2. Regarding Comment 20, what do the red squares in Figure 2c represent? Is it possible that the red squares should be replaced with red triangles?

Response: Thank you for pointing out our error, Figure 2c has been corrected in the revised manuscript.

Modified:

At last, Merry Christmas! As the festive season unfolds, my heart is filled with warm wishes for you.